# Federated Compositional Deep AUC Maximization

**Xinwen Zhang** *
Temple University
Philadelphia, PA, USA
ellenz@temple.edu

**Yihan Zhang** *
Temple University
Philadelphia, PA, USA
yihan.zhang0002@temple.edu

**Tianbao Yang**
Texas A&M University
College Station, TX, USA
ianbao-yang@tamu.edu

**Richard Souvenir**
Temple University
Philadelphia, PA, USA
souvenir@temple.edu

**Hongchang Gao**
Temple University
Philadelphia, PA, USA
hongchang.gao@temple.edu

## Abstract

Federated learning has attracted increasing attention due to the promise of balancing privacy and large-scale learning; numerous approaches have been proposed. However, most existing approaches focus on problems with balanced data, and prediction performance is far from satisfactory for many real-world applications where the number of samples in different classes is highly imbalanced. To address this challenging problem, we developed a novel federated learning method for imbalanced data by directly optimizing the area under curve (AUC) score. In particular, we formulate the AUC maximization problem as a federated compositional minimax optimization problem, develop a local stochastic compositional gradient descent ascent with momentum algorithm, and provide bounds on the computational and communication complexities of our algorithm. To the best of our knowledge, this is the first work to achieve such favorable theoretical results. Finally, extensive experimental results confirm the efficacy of our method.

## 1 Introduction

Federated learning [19, 34] is a paradigm for training a machine learning model across multiple devices without sharing the raw data from each device. Practically, models are trained on each device, and, periodically, model parameters are exchanged between these devices. By not sharing the data itself, federated learning allows private information in the raw data to be preserved to some extent. This property has allowed federated learning to be proposed for numerous real-world computer vision and machine learning tasks.

Currently, one main drawback of existing federated learning methodologies is the assumption of balanced data, where the number of samples across classes is essentially the same. Most real-world data is imbalanced, even highly imbalanced. For example, in the healthcare domain, it is common to encounter problems where the amount of data from one class (e.g., patients with a rare disease) is significantly lower than the other class(es), leading to a distribution that is highly imbalanced. Traditional federated learning methods do not handle such imbalanced data scenarios very well. Specifically, training the classifier typically requires minimizing a classification-error induced loss function (e.g., cross-entropy). As a result, the resulting classifier may excel at classifying the majority, while failing to classify the minority.

To handle imbalanced data classification, the most common approach is to train the classifier by optimizing metrics designed for imbalanced data distributions. For instance, under the single-machine

---

*Equal contributions

37th Conference on Neural Information Processing Systems (NeurIPS 2023).

setting, Ying et al. [35] proposed to train the classifier by maximizing the Area under the ROC curve (AUC) score. Since the AUC score can be affected by performance on both the majority and minority classes, the classifier is less prone to favoring one class above the rest. Later, [8, 40] extended this approach to federated learning. However, optimizing the AUC score introduces some new challenges, since maximizing the AUC score requires solving a minimax optimization problem, which is more challenging to optimize than conventional minimization problems. More specifically, when the classifier is a deep neural network, recent work [39] has demonstrated empirically that training a deep classifier from scratch with the AUC objective function cannot learn discriminative features; the resulting classifier sometimes fails to achieve satisfactory performance. To address this issue, [39] developed a compositional deep AUC maximization model under the single-machine setting, which combines the AUC loss function and the traditional cross-entropy loss function, leading to a stochastic compositional minimax optimization problem. This compositional deep AUC maximization model can learn discriminative features, achieving superior performance over traditional models consistently.

Considering its remarkable performance under the single-machine setting, a natural question is: **How can a compositional deep AUC maximization model be applied to federated learning?** The challenge is that the loss function of the compositional model involves *two levels of distributed functions*. Moreover, the stochastic compositional gradient is a *biased* estimation of the full gradient. Therefore, on the algorithmic design side, it is unclear what variables should be communicated when estimating the stochastic compositional gradient. On the theoretical analysis side, it is unclear if the convergence rate can achieve the linear speedup with respect to the number of devices in the presence of a *biased stochastic compositional gradient*, *two levels of distributed functions*, and the *minimax structure of the loss function*.

To address the aforementioned challenges, in this paper, we developed a novel local stochastic compositional gradient descent ascent with momentum (LocalSCGDAM) algorithm for federated compositional deep AUC maximization. In particular, we demonstrated which variables should be communicated to address the issue of two levels of distributed functions. Moreover, for this nonconvex-strongly-concave problem, we established the convergence rate of our algorithm, disclosing how the communication period and the number of devices affect the computation and communication complexities. Specifically, with theoretical guarantees, the communication period can be as large as $O(T^{1/4}/K^{3/4})$ so that our algorithm can achieve $O(1/\sqrt{KT})$ convergence rate and $O(T^{3/4}/K^{3/4})$ communication complexity, where $K$ is the number of devices and $T$ is the number of iterations. To the best of our knowledge, this is the first work to achieve such favorable theoretical results for the federated compositional minimax problem. Finally, we conduct extensive experiments on multiple image classification benchmark datasets, and the experimental results confirm the efficacy of our algorithm.

In summary, we made the following important contributions in our work.

- We developed a novel federated optimization algorithm, which enables compositional deep AUC maximization for federated learning.
- We established the theoretical convergence rate of our algorithm, demonstrating how it is affected by the communication period and the number of devices.
- We conducted extensive experiments on multiple imbalanced benchmark datasets, confirming the efficacy of our algorithm.

## 2   Related Work

**Imbalanced Data Classification.**   In the field of machine learning, there has been a fair amount of work addressing imbalanced data classification. Instead of using conventional cross-entropy loss functions, which are not suitable for imbalanced datasets, optimizing the AUC score has been proposed. For instance, Ying et al. [35] have proposed the minimax loss function to optimize the AUC score for learning linear classifiers. Liu et al. [18] extended this minimax method to deep neural networks and developed the nonconvex-strongly-concave loss function. Yuan et al. [39] have proposed a compositional training framework for end-to-end deep AUC maximization, which minimizes a compositional loss function, where the outer-level function is an AUC loss, and the inner-level function substitutes a gradient descent step for minimizing a traditional loss. Based on empirical results, this approach improved the classification performance by a large degree.

To address stochastic minimax optimization problems, there have been a number of diverse efforts launched in recent years. In particular, numerous stochastic gradient descent ascent (SGDA) algo-

rithms [42, 17, 21, 33] have been proposed. However, most of them focus on non-compositional optimization problems. On the other hand, to solve compositional optimization problems, existing work [31, 41, 38, 7] tends to only focus on the minimization problem. Only two recent works [5, 39] studied how to optimize the compositional minimax optimization problem, but they focused on the single-machine setting.

**Federated Learning.** In recent years, federated learning has shown promise with several empirical studies in the field of large-scale deep learning [19, 20, 26]. The FedAvg [19] algorithm has spawned a number of variants [25, 37, 36] designed to address the minimization problem. For instance, by maintaining a local momentum, Yu et al. [36] have provided rigorous theoretical studies for the convergence of the local stochastic gradient descent with momentum (LocalSGDM) algorithm. These algorithms are often applied to balanced datasets, and their performance in the imbalanced imbalanced regime is lacking.

To address minimax optimization for federated learning, Deng et al.[2] proposed local stochastic gradient descent ascent (LocalSGDA) algorithms to provably optimize federated minimax problems. However, their theoretical convergence rate was suboptimal and later improved by [27]. However, neither method could achieve a linear speedup with respect to the number of devices. Recently, Sharma et al. [23] developed the local stochastic gradient descent ascent with momentum (LocalSGDAM) algorithm, whose convergence rate is able to achieve a linear speedup for nonconvex-strongly-concave optimization problems. Guo et al. [8] proposed and analyzed a communication-efficient distributed optimization algorithm (CoDA) for the minimax AUC loss function under the assumption of PL-condition, which can also achieve a linear speedup, in theory. Yuan et al. [40] extended CoDA to hetereogneous data distributions and established its convergence rate. Shen et al. [24] proposed to handle the imbalance issue via introducing a constrained optimization problem and then formulated it as an unconstrained minimax problem. While these algorithms are designed for federated minimax problems, none can deal with the federated *compositional* minimax problems.

To handle the compositional optimization problem under the distributed setting, Gao et al. [3] developed the *first* parallel stochastic compositional gradient descent algorithm and established its convergence rate for nonconvex problems, inspiring many federated learning methods [12, 28, 4, 27, 7] in the past few years. For instance, [12] directly used a biased stochastic gradient to do local updates, suffering from large sample and communication complexities. [4] employed the stochastic compositional gradient and the momentum technique, which can achieve much better sample and communication complexities than [12]. [7] considered the setting where the inner-level function is distributed on different devices, which shares similar sample and communication complexities as [4]. However, all these works restrict their focus on the compositional *minimization* problem.

It is worth noting that our method is significantly different from the heterogeneous federated learning approaches [14, 16, 29]. Specifically, most existing heterogeneous federated learning approaches consider a setting where *the local distribution is imbalanced but the global distribution is balanced*. For example, Scaffold [14] method uses the global gradient to correct the local gradient because it assumes the global gradient is computed on a balanced distribution. On the contrary, our work considers a setting where *both the local and global distributions are imbalanced*, which is much more challenging than existing heterogeneous federated learning methods.

## 3 Preliminaries

In this section, we first introduce the compositional deep AUC maximization model under the single-machine setting and then provide the problem setup in federated learning.

### 3.1 Compositional Deep AUC Maximization

Training classifiers by optimizing AUC ([9, 11]) is an effective way to handle highly imbalanced datasets. However, traditional AUC maximization models typically depend on pairwise sample input, limiting the application to large-scale data. Recently, Ying et al. [35] formulated AUC maximization model as a minimax optimization problem, defined as follows:

$$
\min_{\mathbf{w},\tilde{w}_1,\tilde{w}_2} \max_{\tilde{w}_3} \mathcal{L}_{AUC}(\mathbf{w},\tilde{w}_1,\tilde{w}_2,\tilde{w}_3;a,b) \triangleq (1-p)(h(\mathbf{w};a)-\tilde{w}_1)^2 \mathbb{I}_{[b=1]} - p(1-p)\tilde{w}_3^2
$$
$$
+ p(h(\mathbf{w};a)-\tilde{w}_2)^2 \mathbb{I}_{[b=-1]} + 2(1+\tilde{w}_3)(ph(\mathbf{w};a)\mathbb{I}_{[b=-1]} - (1-p)h(\mathbf{w};a)\mathbb{I}_{[b=1]}) ,
$$
(1)

where $h$ denotes the classifier parameterized by $\mathbf{w} \in \mathbb{R}^d$, $\tilde{w}_1 \in \mathbb{R}$, $\tilde{w}_2 \in \mathbb{R}$, $\tilde{w}_3 \in \mathbb{R}$ are the parameters for measuring AUC score, $(a, b)$ substitutes the sample's feature and label, $p$ is the prior probability of the positive class, and $\mathbb{I}$ is an indicator function that takes value 1 if the argument is true and 0 otherwise. Such a minimax objective function decouples the dependence of pairwise samples so that it can be applied to large-scale data.

Since training a deep classifier from scratch with $\mathcal{L}_{AUC}$ loss function did not yield satisfactory performance, Yuan et al. [39] developed the compositional deep AUC maximization model, which is defined as follows:

$$\min_{\tilde{\mathbf{w}}, \tilde{w}_1, \tilde{w}_2} \max_{\tilde{w}_3} \mathcal{L}_{AUC}(\tilde{\mathbf{w}}, \tilde{w}_1, \tilde{w}_2, \tilde{w}_3; a, b) \qquad s.t. \quad \tilde{\mathbf{w}} = \mathbf{w} - \rho \nabla_{\mathbf{w}} \mathcal{L}_{CE}(\mathbf{w}; a, b) . \tag{2}$$

Here, $\mathcal{L}_{CE}$ denotes the cross-entropy loss function, $\mathbf{w} - \rho \nabla_{\mathbf{w}} \mathcal{L}_{CE}$ indicates using the gradient descent method to minimize the cross-entropy loss function, where $\rho > 0$ is the learning rate. Then, for the obtained model parameter $\tilde{\mathbf{w}}$, one can optimize it through optimizing the AUC loss function.

By denoting $g(\mathbf{x}) = \mathbf{x} - \rho\Delta(\mathbf{x})$ and $\mathbf{y} = \tilde{w}_3$, where $\mathbf{x} = [\mathbf{w}^T, \tilde{w}_1, \tilde{w}_2]^T$, $\Delta(\mathbf{x}) = [\nabla_{\mathbf{w}} \mathcal{L}_{CE}(\mathbf{w}; a, b)^T, 0, 0]^T$, and $f = \mathcal{L}_{AUC}$, Eq. (2) can be represented as a generic compositional minimax optimization problem as follows:

$$\min_{\mathbf{x} \in \mathbb{R}^{d_1}} \max_{\mathbf{y} \in \mathbb{R}^{d_2}} f(g(\mathbf{x}), \mathbf{y}) , \tag{3}$$

where $g$ is the inner-level function and $f$ is the outer-level function. It is worth noting that when $f$ is a nonlinear function, the stochastic gradient regarding $\mathbf{x}$ is a biased estimation of the full gradient. As such, the stochastic compositional gradient [31] is typically used to optimize this kind of problem. We will demonstrate how to adapt this compositional minimax optimization problem to federated learning and address the unique challenges.

## 3.2 Problem Setup

In this paper, to optimize the deep compositional AUC maximization problem under the cross-silo federated learning setting, we will concentrate on developing an efficient optimization algorithm to solve the following generic federated stochastic compositional minimax optimization problem:

$$\min_{\mathbf{x} \in \mathbb{R}^{d_1}} \max_{\mathbf{y} \in \mathbb{R}^{d_2}} \frac{1}{K} \sum_{k=1}^{K} f^{(k)} \Big( \frac{1}{K} \sum_{k'=1}^{K} g^{(k')}(\mathbf{x}), \mathbf{y} \Big) , \tag{4}$$

where $K$ is the number of devices, $g^{(k)}(\cdot) = \mathbb{E}_{\xi \sim \mathcal{D}_g^{(k)}}[g^{(k)}(\cdot; \xi)] \in \mathbb{R}^{d_g}$ denotes the inner-level function for the data distribution $\mathcal{D}_g^{(k)}$ of the $k$-th device, $f^{(k)}(\cdot, \cdot) = \mathbb{E}_{\zeta \sim \mathcal{D}_f^{(k)}}[f^{(k)}(\cdot, \cdot; \zeta)]$ represents the outer-level function for the data distribution $\mathcal{D}_f^{(k)}$ of the $k$-th device. It is worth noting that *both the inner-level function and the outer-level function are distributed on different devices*, which is significantly different from traditional federated learning models. Therefore, we need to design a new federated optimization algorithm to address this unique challenge.

Here, we introduce the commonly-used assumptions from existing work [6, 41, 39, 5] for investigating the convergence rate of our algorithm.

**Assumption 1.** *The gradient of the outer-level function $f^{(k)}(\cdot, \cdot)$ is $L_f$-Lipschitz continuous where $L_f > 0$, i.e.,*

$$\|\nabla_g f^{(k)}(\mathbf{z}_1, \mathbf{y}_1) - \nabla_g f^{(k)}(\mathbf{z}_2, \mathbf{y}_2)\|^2 \le L_f^2 \|(\mathbf{z}_1, \mathbf{y}_1) - (\mathbf{z}_2, \mathbf{y}_2)\|^2 ,$$
$$\|\nabla_{\mathbf{y}} f^{(k)}(\mathbf{z}_1, \mathbf{y}_1) - \nabla_{\mathbf{y}} f^{(k)}(\mathbf{z}_2, \mathbf{y}_2)\|^2 \le L_f^2 \|(\mathbf{z}_1, \mathbf{y}_1) - (\mathbf{z}_2, \mathbf{y}_2)\|^2 , \tag{5}$$

*hold for $\forall (\mathbf{z}_1, \mathbf{y}_1), (\mathbf{z}_2, \mathbf{y}_2) \in \mathbb{R}^{d_g} \times \mathbb{R}^{d_2}$. The gradient of the inner-level function $g^{(k)}(\cdot)$ is $L_g$-Lipschitz continuous where $L_g > 0$, i.e.,*

$$\|\nabla g^{(k)}(\mathbf{x}_1) - \nabla g^{(k)}(\mathbf{x}_2)\|^2 \le L_g^2 \|\mathbf{x}_1 - \mathbf{x}_2\|^2 , \tag{6}$$

*holds for $\forall \mathbf{x}_1, \mathbf{x}_2 \in \mathbb{R}^{d_1}$.*

**Assumption 2.** *The second moment of $\nabla_g f^{(k)}(\mathbf{z}, \mathbf{y}; \zeta)$ and $\nabla g^{(k)}(\mathbf{x}; \xi)$ satisfies:*

$$\mathbb{E}[\|\nabla_g f^{(k)}(\mathbf{z}, \mathbf{y}; \zeta)\|^2] \le C_f^2 , \mathbb{E}[\|\nabla g^{(k)}(\mathbf{x}; \xi)\|^2] \le C_g^2 , \tag{7}$$

*for $\forall (\mathbf{z}, \mathbf{y}) \in \mathbb{R}^{d_g} \times \mathbb{R}^{d_2}$ and $\forall \mathbf{x} \in \mathbb{R}^{d_1}$, where $C_f > 0$ and $C_g > 0$. Meanwhile, the second moment of the full gradient is assumed to have the same upper bound.*

**Assumption 3.** *The variance of the stochastic gradient of the outer-level function* $f^{(k)}(\cdot, \cdot)$ *satisfies:*

$$\mathbb{E}[\|\nabla_g f^{(k)}(\mathbf{z}, \mathbf{y}; \zeta) - \nabla_g f^{(k)}(\mathbf{z}, \mathbf{y})\|^2] \le \sigma_f^2, \mathbb{E}[\|\nabla_{\mathbf{y}} f^{(k)}(\mathbf{z}, \mathbf{y}; \zeta) - \nabla_{\mathbf{y}} f^{(k)}(\mathbf{z}, \mathbf{y})\|^2] \le \sigma_f^2 , \quad (8)$$

*for* $\forall (\mathbf{z}, \mathbf{y}) \in \mathbb{R}^{d_g} \times \mathbb{R}^{d_2}$, *where* $\sigma_f > 0$. *Additionally, the variance of the stochastic gradient and the stochastic function value of* $g^{(k)}(\cdot)$ *satisfies:*

$$\mathbb{E}[\|\nabla g^{(k)}(\mathbf{x}; \xi) - \nabla g^{(k)}(\mathbf{x})\|^2] \le \sigma_{g'}^2 , \mathbb{E}[\|g^{(k)}(\mathbf{x}; \xi) - g^{(k)}(\mathbf{x})\|^2] \le \sigma_g^2 , \quad (9)$$

*for* $\forall \mathbf{x} \in \mathbb{R}^{d_1}$, *where* $\sigma_g > 0$ *and* $\sigma_{g'} > 0$.

**Assumption 4.** *The outer-level function* $f^{(k)}(\mathbf{z}, \mathbf{y})$ *is* $\mu$-*strongly-concave with respect to* $\mathbf{y}$ *for any fixed* $\mathbf{z} \in \mathbb{R}^{d_g}$, *where* $\mu > 0$, *i.e.,*

$$f^{(k)}(\mathbf{z}, \mathbf{y_1}) \le f^{(k)}(\mathbf{z}, \mathbf{y_2}) + \langle \nabla_y f^{(k)}(\mathbf{z}, \mathbf{y_2}), \mathbf{y_1} - \mathbf{y_2} \rangle - \frac{\mu}{2}\|\mathbf{y_1} - \mathbf{y_2}\|^2 . \quad (10)$$

**Notation:** Throughout this paper, $\mathbf{a}_t^{(k)}$ denotes the variable of the $k$-th device in the $t$-th iteration and $\bar{\mathbf{a}}_t = \frac{1}{K}\sum_{k=1}^{K} \mathbf{a}_t^{(k)}$ denotes the averaged variable across all devices, where $a$ denotes any variables used in this paper. $\mathbf{x}_*$ denotes the optimal solution.

## 4 Methodology

In this section, we present the details of our algorithm for the federated compositional deep AUC maximization problem defined in Eq. (4).

---

**Algorithm 1** LocalSCGDAM

---

**Input:** $\mathbf{x}_0, \mathbf{y}_0, \eta \in (0, 1), \gamma_x > 0, \gamma_y > 0, \beta_x > 0, \beta_y > 0, \alpha > 0, \alpha\eta \in (0, 1), \beta_x\eta \in (0, 1), \beta_y\eta \in (0, 1)$.

    All workers conduct the steps below to update $\mathbf{x}, \mathbf{y}$. $\mathbf{x}_0^{(k)} = \mathbf{x}_0$ , $\mathbf{y}_0^{(k)} = \mathbf{y}_0$,

    $\mathbf{h}_0^{(k)} = g^{(k)}(\mathbf{x}_0^{(k)}; \xi_0^{(k)})$ ,

    $\mathbf{u}_0^{(k)} = \nabla g^{(k)}(\mathbf{x}_0^{(k)}; \xi_0^{(k)})^T \nabla_g f^{(k)}(\mathbf{h}_0^{(k)}, \mathbf{y}_0^{(k)}; \zeta_0^{(k)}), \quad \mathbf{v}_0^{(k)} = \nabla_y f^{(k)}(\mathbf{h}_0^{(k)}, \mathbf{y}_0^{(k)}; \zeta_0^{(k)})$,

1: **for** $t = 0, \cdots, T - 1$ **do**
2:     Update $\mathbf{x}$ and $\mathbf{y}$:
        $\mathbf{x}_{t+1}^{(k)} = \mathbf{x}_t^{(k)} - \gamma_x \eta \mathbf{u}_t^{(k)}$ ,    $\mathbf{y}_{t+1}^{(k)} = \mathbf{y}_t^{(k)} + \gamma_y \eta \mathbf{v}_t^{(k)}$ ,
3:     Estimate the inner-level function:
        $\mathbf{h}_{t+1}^{(k)} = (1 - \alpha\eta)\mathbf{h}_t^{(k)} + \alpha\eta g^{(k)}(\mathbf{x}_{t+1}^{(k)}; \xi_{t+1}^{(k)})$,
4:     Update momentum:
        $\mathbf{u}_{t+1}^{(k)} = (1 - \beta_x\eta)\mathbf{u}_t^{(k)} + \beta_x\eta\nabla g^{(k)}(\mathbf{x}_{t+1}^{(k)}; \xi_{t+1}^{(k)})^T \nabla_g f^{(k)}(\mathbf{h}_{t+1}^{(k)}, \mathbf{y}_{t+1}^{(k)}; \zeta_{t+1}^{(k)})$,
        $\mathbf{v}_{t+1}^{(k)} = (1 - \beta_y\eta)\mathbf{v}_t^{(k)} + \beta_y\eta\nabla_y f^{(k)}(\mathbf{h}_{t+1}^{(k)}, \mathbf{y}_{t+1}^{(k)}; \zeta_{t+1}^{(k)})$,
5:     **if** $\text{mod}(t + 1, p) == 0$ **then**
6:         $\mathbf{h}_{t+1}^{(k)} = \bar{\mathbf{h}}_{t+1} \triangleq \frac{1}{K}\sum_{k'=1}^{K} \mathbf{h}_{t+1}^{(k')}$ ,
        $\mathbf{u}_{t+1}^{(k)} = \bar{\mathbf{u}}_{t+1} \triangleq \frac{1}{K}\sum_{k'=1}^{K} \mathbf{u}_{t+1}^{(k')}$ ,    $\mathbf{v}_{t+1}^{(k)} = \bar{\mathbf{v}}_{t+1} \triangleq \frac{1}{K}\sum_{k'=1}^{K} \mathbf{v}_{t+1}^{(k')}$ ,
        $\mathbf{x}_{t+1}^{(k)} = \bar{\mathbf{x}}_{t+1} \triangleq \frac{1}{K}\sum_{k'=1}^{K} \mathbf{x}_{t+1}^{(k')}$ ,    $\mathbf{y}_{t+1}^{(k)} = \bar{\mathbf{y}}_{t+1} \triangleq \frac{1}{K}\sum_{k'=1}^{K} \mathbf{y}_{t+1}^{(k')}$ ,
7:     **end if**
8: **end for**

---

To optimize Eq. (4), we developed a novel local stochastic compositional gradient descent ascent with momentum algorithm, shown in Algorithm 1. Generally speaking, in the $t$-th iteration, we employ the local stochastic (compositional) gradient with momentum to update the local model parameters $\mathbf{x}_t^{(k)}$ and $\mathbf{y}_t^{(k)}$ on the $k$-th device. There exists an unique challenge when computing the local stochastic compositional gradient compared to traditional federated learning models. Specifically, as shown in Eq. (4), *the objective function depends on the global inner-level function*. However, it is not feasible to communicate the inner-level function in every iteration. To address this challenge, we propose to employ the *local* inner-level function to compute the stochastic compositional gradient at each iteration and then communicate the estimation of this function periodically to obtain the global inner-level function.

In detail, since the objective function in Eq. (4) is a compositional function whose stochastic gradient regarding $\mathbf{x}$, i.e., $\nabla g^{(k)}(\mathbf{x}_t^{(k)}; \xi_t^{(k)})^T \nabla_g f^{(k)}(g^{(k)}(\mathbf{x}_t^{(k)}; \xi_t^{(k)}), \mathbf{y}_t^{(k)}; \zeta_t^{(k)})$, is a biased estimation for the full gradient, we employ the stochastic compositional gradient $\nabla g^{(k)}(\mathbf{x}_t^{(k)}; \xi_t^{(k)})^T \nabla_g f^{(k)}(\mathbf{h}_t^{(k)}, \mathbf{y}_t^{(k)}; \zeta_t^{(k)})$ to update the model parameter $\mathbf{x}$, where $\mathbf{h}_t^{(k)}$ is the moving-average estimation of the inner-level function $g^{(k)}(\mathbf{x}_t^{(k)}; \xi_t^{(k)})$ on the $k$-th device, which is defined as follows:

$$\mathbf{h}_t^{(k)} = (1 - \alpha\eta)\mathbf{h}_{t-1}^{(k)} + \alpha\eta g^{(k)}(\mathbf{x}_t^{(k)}; \xi_t^{(k)}) , \tag{11}$$

where $\alpha > 0$ and $\eta > 0$ are two hyperparameters, and $\alpha\eta \in (0, 1)$. The objective function in Eq. (4) is not compositional regarding $\mathbf{y}$, thus we can directly leverage its stochastic gradient to perform an update. Then, based on the obtained stochastic (compositional) gradient, we compute the momentum as follows:

$$\begin{aligned}
\mathbf{u}_t^{(k)} &= (1 - \beta_x\eta)\mathbf{u}_{t-1}^{(k)} + \beta_x\eta\nabla g^{(k)}(\mathbf{x}_t^{(k)}; \xi_t^{(k)})^T \nabla_g f^{(k)}(\mathbf{h}_t^{(k)}, \mathbf{y}_t^{(k)}; \zeta_t^{(k)}) , \\
\mathbf{v}_t^{(k)} &= (1 - \beta_y\eta)\mathbf{v}_{t-1}^{(k)} + \beta_y\eta\nabla_y f^{(k)}(\mathbf{h}_t^{(k)}, \mathbf{y}_t^{(k)}; \zeta_t^{(k)}) ,
\end{aligned} \tag{12}$$

where $\beta_x > 0$ and $\beta_y > 0$ are two hyperparameters, $\beta_x\eta \in (0, 1)$, and $\beta_y\eta \in (0, 1)$. Based on the obtained momentum, each device updates its local model parameters as follows:

$$\mathbf{x}_{t+1}^{(k)} = \mathbf{x}_t^{(k)} - \gamma_x\eta\mathbf{u}_t^{(k)} , \mathbf{y}_{t+1}^{(k)} = \mathbf{y}_t^{(k)} + \gamma_y\eta\mathbf{v}_t^{(k)} , \tag{13}$$

where $\gamma_x > 0$ and $\gamma_y > 0$.

As we mentioned before, to obtain the global inner-level function, our algorithm periodically communicates the moving-average estimation of the inner-level function, i.e., $\mathbf{h}_{t+1}^{(k)}$. In particular, at every $p$ iterations, i.e., $\mathrm{mod}(t + 1, p) == 0$ where $p > 1$ is the *communication period*, each device uploads $\mathbf{h}_{t+1}^{(k)}$ to the central server and the central server computes the average of all received variables, which will be further broadcast to all devices as follows:

$$\mathbf{h}_{t+1}^{(k)} = \bar{\mathbf{h}}_{t+1} \triangleq \frac{1}{K}\sum_{k'=1}^{K}\mathbf{h}_{t+1}^{(k')} . \tag{14}$$

In this way, each device is able to obtain the estimate of the global inner-level function periodically. As for the model parameters and momentum, we employ the same strategy as traditional federated learning methods [23, 36] to communicate them periodically with the central server, which is shown in Step 6 in Algorithm 1.

In summary, we developed a novel local stochastic compositional gradient descent ascent with momentum algorithm for the compositional minimax problem, which shows how to deal with two distributed functions in federated learning. With our algorithm, we can enable federated learning for the compositional deep AUC maximization model, benefiting imbalanced data classification tasks.

## 5 Theoretical Analysis

In this section, we provide the convergence rate of our algorithm to show how it is affected by the number of devices and communication period.

To investigate the convergence rate of our algorithm, we introduce the following auxiliary functions:

$$\mathbf{y}_*(\mathbf{x}) = \arg\max_{\mathbf{y}\in\mathbb{R}^{d_2}} \frac{1}{K}\sum_{k=1}^{K} f^{(k)}\Big(\frac{1}{K}\sum_{k'=1}^{K} g^{(k')}(\mathbf{x}), \mathbf{y}\Big) , \quad \Phi(\mathbf{x}) = \frac{1}{K}\sum_{k=1}^{K} f^{(k)}\Big(\frac{1}{K}\sum_{k'=1}^{K} g^{(k')}(\mathbf{x}), \mathbf{y}_*(\mathbf{x})\Big) . \tag{15}$$

Then, based on Assumptions 1-4, we can obtain that $\Phi^{(k)}$ is $L_\Phi$-smooth, where $L_\Phi = \frac{2C_g^2 L_f^2}{\mu} + C_f L_g$. The proof can be found in Lemma 2 of Appendix A. In terms of these auxiliary functions, we establish the convergence rate of our algorithm.

**Theorem 1.** *Given Assumption 1-4, by setting* $\alpha > 0$, $\beta_x > 0$, $\beta_y > 0$, $\eta \leq \min\{\frac{1}{2\gamma_x L_\phi}, \frac{1}{10p^2\gamma_y L_f}, \frac{1}{\alpha}, \frac{1}{\beta_x}, \frac{1}{\beta_y}, 1\}$, $\gamma_y \leq \min\Big\{\frac{1}{6L_f}, \frac{3\mu\beta_y^2}{400L_f^2}, \frac{3\beta_x^2}{16\mu}\Big\}$, *and* $\gamma_x \leq \min\Big\{\frac{\alpha\mu}{100C_g^2 L_f\sqrt{1+6L_f^2}}, \frac{\beta_x}{32\sqrt{C_g^4 L_f^2 + C_f^2 L_g^2}}, \frac{\beta_y\mu}{144C_g^2 L_f^2}, \frac{\sqrt{\alpha}\mu}{24C_g\sqrt{100C_g^2 L_f^4 + 2C_g^2\mu^2}}, \frac{\gamma_y\mu^2}{20C_g^2 L_f^2}\Big\}$,

*Algorithm 1 has the following convergence rate*

$$\frac{1}{T}\sum_{t=0}^{T-1}\mathbb{E}[\|\nabla\Phi(\bar{\mathbf{x}}_t)\|^2] \leq \frac{2(\Phi(\mathbf{x}_0) - \Phi(\mathbf{x}_*))}{\gamma_x\eta T} + \frac{24C_g^2 L_f^2}{\gamma_y\eta\mu T}\|\mathbf{y}_0 - \mathbf{y}^*(\mathbf{x}_0)\|^2 + O(\frac{\eta}{K}) + O(\frac{1}{\eta T})$$
$$+ O(p^2\eta^2) + O(p^4\eta^4) + O(p^6\eta^6) + O(p^8\eta^8) + O(p^{10}\eta^{10}) \, .$$

$$(16)$$

**Remark 1.** *In terms of Theorem 1, for sufficiently large $T$, by setting the learning rate $\eta = O(K^{1/2}/T^{1/2}), p = O(T^{1/4}/K^{3/4})$, Algorithm 1 can achieve $O(1/\sqrt{KT})$ convergence rate, which indicates a linear speedup with respect to the number of devices $K$. In addition, it is straightforward to show that the communication complexity of our algorithm is $T/p = O(K^{3/4}T^{3/4})$. Moreover, to achieve the $\epsilon$-accuracy solution, i.e., $\frac{1}{T}\sum_{t=0}^{T-1}\mathbb{E}[\|\nabla\Phi(\bar{\mathbf{x}}_t)\|^2] \leq \epsilon^2$, by setting $\eta = O(K\epsilon^2)$ and $p = O(1/(K\epsilon))$, then the sample complexity on each device is $O(1/(K\epsilon^4))$ and the communication complexity is $O(1/\epsilon^3)$.*

**Challenges.** The compositional structure in the loss function, especially when the inner-level functions are distributed on different devices, makes the convergence analysis challenging. In fact, the existing federated compositional *minimization* algorithms [4, 27, 12, 28] only consider a much simpler case where the inner-level functions are not distributed across devices. Therefore, our setting is much more challenging than existing works. On the other hand, all existing federated compositional *minimization* algorithms [4, 27, 12, 28] fail to achieve linear speedup with respect to the numbe of devices $K$. Thus, it is still unclear whether the linear speedup is achievable for federated compositional optimization algorithm. In this paper, we successfully addressed these challenges with novel theoretical analysis strategies and achieved the linear speedup for the first time for federated compositional minimax optimization algorithms. We believe our approaches, e.g., that for bounding consensus errors, can be applied to the minimization algorithms to achieve linear speedup.

## 6 Experiments

In this section, we present the experimental results to demonstrate the performance of our algorithm.

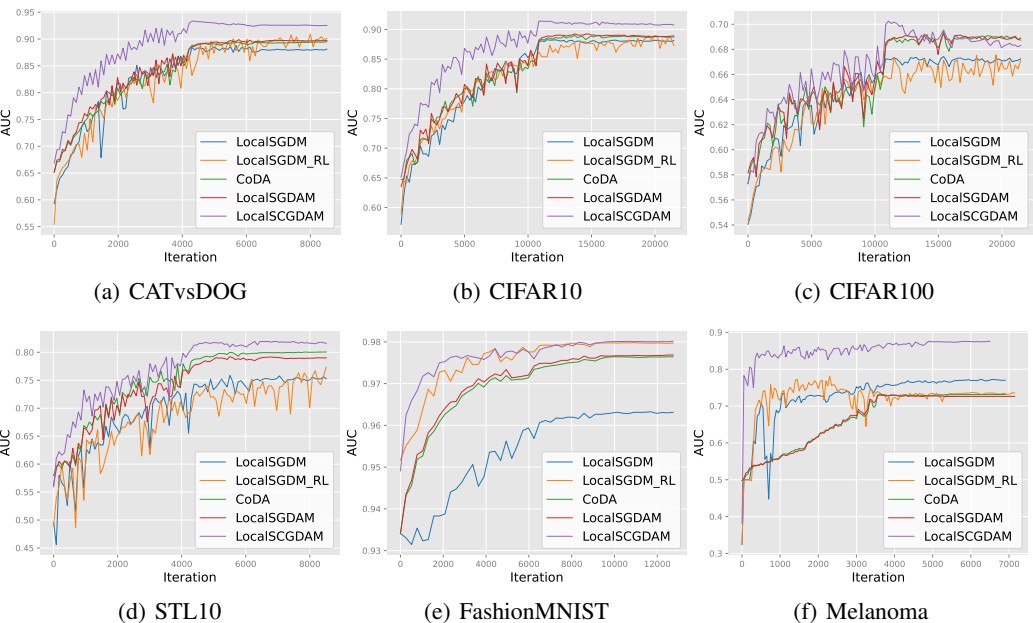

(a) CATvsDOG      (b) CIFAR10      (c) CIFAR100

(d) STL10      (e) FashionMNIST      (f) Melanoma

Figure 1: Testing performance with AUC score versus the number of iterations when the communication period $p = 4$.

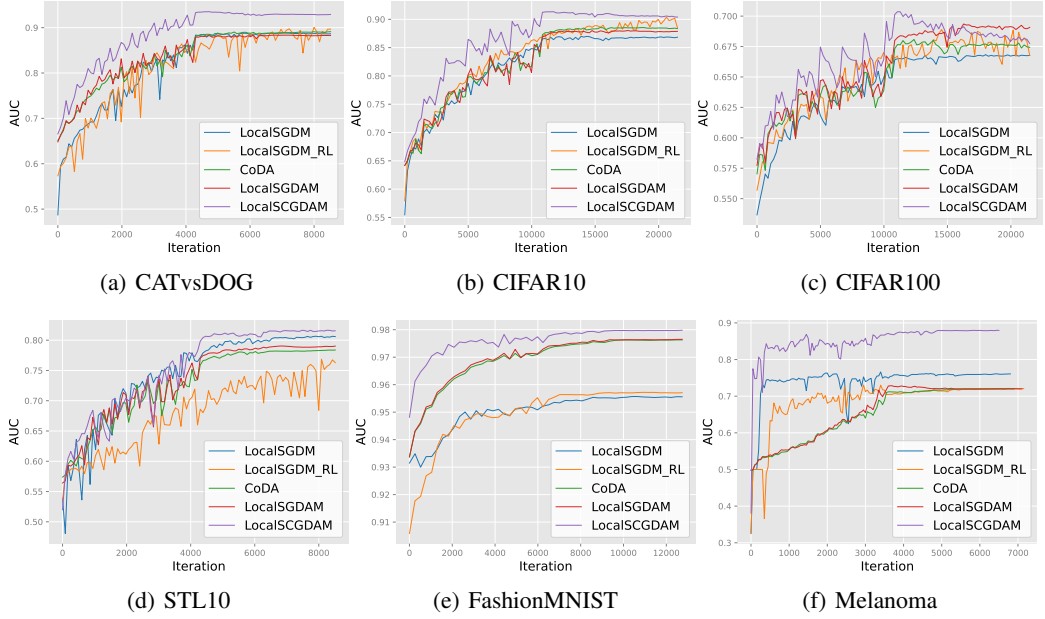

Figure 2: Testing performance with AUC score versus the number of iterations when the communication period $p = 8$.

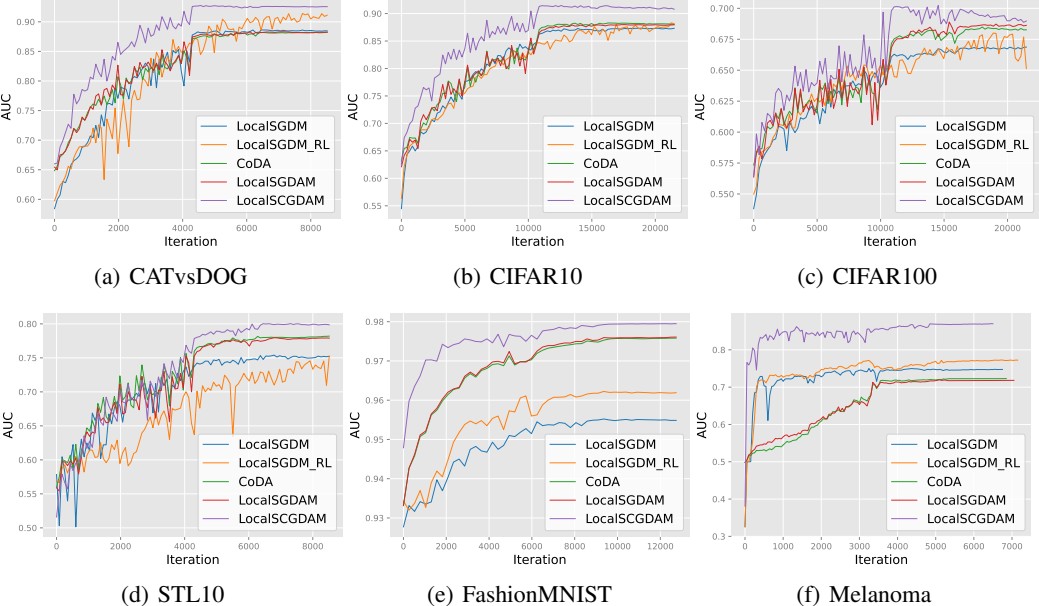

Figure 3: Testing performance with AUC score versus the number of iterations when the communication period $p = 16$.

## 6.1 Experimental Setup

**Datasets.** In our experiments, we employ six image classification datasets, including CIFAR10 [15], CIFAR100 [15], STL10 [1], FashionMNIST [32], CATvsDOG [2], and Melanoma [22]. For the first four datasets, following [39], we consider the first half of classes to be the positive class, and the second half as the negative class. Then, in order to construct highly imbalanced data, we randomly drop some samples of the positive class in the training set. Specifically, the ratio between positive samples and all samples is set to 0.1. For the two-class dataset, CATvsDOG, we employ the same

[2]https://www.kaggle.com/c/dogs-vs-cats

Table 1: The comparison between the test AUC score of different methods on all datasets. Here, $p$ denotes the communication period.

| Datasets | Methods | AUC | | |
|---|---|---|---|---|
| | | $p = 4$ | $p = 8$ | $p = 16$ |
| CATvsDOG | **LocalSCGDAM** | **0.933±0.000** | **0.936±0.000** | **0.928±0.000** |
| | CoDA | 0.895±0.000 | 0.892±0.000 | 0.883±0.001 |
| | LocalSGDAM | 0.899±0.000 | 0.884±0.000 | 0.884±0.001 |
| | LocalSGDM | 0.888±0.001 | 0.889±0.000 | 0.887±0.000 |
| | LocalSGDM_RL | 0.909±0.000 | 0.901±0.001 | 0.917±0.001 |
| CIFAR10 | **LocalSCGDAM** | **0.914±0.000** | **0.914±0.000** | **0.916±0.000** |
| | CoDA | 0.890±0.000 | 0.886±0.000 | 0.883±0.000 |
| | LocalSGDAM | 0.893±0.000 | 0.880±0.000 | 0.880±0.000 |
| | LocalSGDM | 0.883±0.001 | 0.871±0.000 | 0.874±0.000 |
| | LocalSGDM_RL | 0.890±0.000 | 0.904±0.001 | 0.883±0.001 |
| CIFAR100 | **LocalSCGDAM** | **0.702±0.000** | **0.704±0.001** | **0.703±0.001** |
| | CoDA | 0.694±0.001 | 0.681±0.001 | 0.685±0.000 |
| | LocalSGDAM | 0.692±0.000 | 0.694±0.000 | 0.689±0.001 |
| | LocalSGDM | 0.675±0.001 | 0.669±0.000 | 0.669±0.000 |
| | LocalSGDM_RL | 0.676±0.000 | 0.690±0.001 | 0.682±0.001 |
| STL10 | **LocalSCGDAM** | **0.820±0.001** | **0.817±0.000** | **0.801±0.000** |
| | CoDA | 0.801±0.000 | 0.784±0.000 | 0.783±0.000 |
| | LocalSGDAM | 0.792±0.000 | 0.790±0.000 | 0.780±0.000 |
| | LocalSGDM | 0.760±0.001 | 0.808±0.000 | 0.757±0.001 |
| | LocalSGDM_RL | 0.773±0.001 | 0.771±0.003 | 0.752±0.003 |
| FashionMNIST | **LocalSCGDAM** | **0.980±0.000** | **0.980±0.000** | **0.980±0.000** |
| | CoDA | 0.976±0.000 | 0.976±0.000 | 0.976±0.000 |
| | LocalSGDAM | 0.977±0.000 | 0.977±0.000 | 0.976±0.000 |
| | LocalSGDM | 0.963±0.000 | 0.956±0.000 | 0.955±0.000 |
| | LocalSGDM_RL | 0.980±0.000 | 0.957±0.000 | 0.962±0.000 |
| Melanoma | **LocalSCGDAM** | **0.876±0.000** | **0.880±0.000** | **0.870±0.000** |
| | CoDA | 0.734±0.002 | 0.721±0.000 | 0.725±0.003 |
| | LocalSGDAM | 0.730±0.000 | 0.729±0.000 | 0.721±0.003 |
| | LocalSGDM | 0.774±0.001 | 0.766±0.001 | 0.750±0.000 |
| | LocalSGDM_RL | 0.737±0.001 | 0.721±0.000 | 0.773±0.001 |

strategy to construct the imbalanced training data. For these synthetic imbalanced datasets, the testing set is balanced. Melanoma is an intrinsically imbalanced medical image classification dataset, which we do not modify. The details about these benchmark datasets are summarized in Table 4.

**Experimental Settings.** For Melanoma, we use DenseNet121 [13] where the dimensionality of the last layer is set to 1 for binary classification. The details for the classifier for FashionMNIST can be found in Appendix B. For the other datasets, we use ResNet20 [10], where the last layer is also set to 1. To demonstrate the performance of our algorithm, we compare it with three state-of-the-art methods: LocalSGDM [36], LocalSGDM_RL [30], CoDA [8], LocalSGDAM [23]. Specifically, LocalSGDM uses momentum SGD to optimize the standard cross-entropy loss function. LocalSGDM_RL employs momentum SGD to optimize a Ratio Loss function, which is to add a regularization term to the standard cross-entropy loss function to address the imbalance distribution issue. CoDA leverages SGDA to optimize AUC loss, while LocalSGDAM exploits momentum SGDA to optimize AUC loss. For a fair comparison, we use similar learning rates for all algorithms. The details can be found in Appendix B. We use 4 devices (i.e., GPUs) in our experiment. The batch size on each device is set to 8 for STL10, 16 for Melanoma, and 32 for the others.

## 6.2 Experimental Results

In Table 1, we report the AUC score of the test set for all methods, where we show the average and variance computed across all devices. Here, the communication period is set to 4, 8, and 16, respectively. It can be observed that our LocalSCGDAM algorithm outperforms all competing methods

for all cases. For instance, our LocalSCGDAM can beat baseline methods on CATvsDOG dataset with a large margin for all communication periods. These observations confirm the effectiveness of our algorithm. In addition, we plot the average AUC score of the test set versus the number of iterations in Figures 1, 2, 3. It can also be observed that our algorithm outperforms baseline methods consistently, which further confirms the efficacy of our algorithm.

To further demonstrate the performance of our algorithm, we apply these algorithms to the dataset with different imbalance ratios. Using the CATvsDOG dataset, we set the imbalance ratio to 0.01, 0.05, and 0.2 to construct three imbalanced training sets. The averaged testing AUC score of these three datasets versus the number of iterations is shown in Figure 4. It can be observed that our algorithm outperforms competing methods consistently and is robust to large imbalances in the training data. Especially when the training set is highly imbalanced, e.g., the imbalance ratio is 0.01, all AUC based methods outperform the cross-entropy loss based method significantly, and our LocalSCGDAM beats other AUC based methods with a large margin.

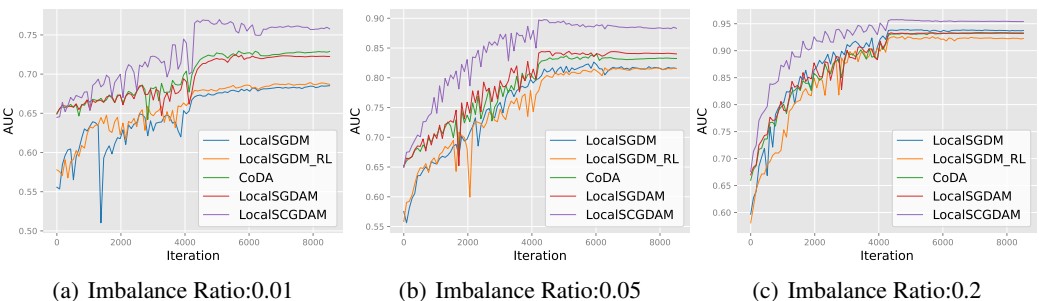

(a) Imbalance Ratio:0.01  (b) Imbalance Ratio:0.05  (c) Imbalance Ratio:0.2

Figure 4: The test AUC score versus the number of iterations when using different imbalance ratios for CATvsDOG.

Finally, we compare our algorithm with two additional baseline methods: SCAFFOLD [14] and FedProx [16]. These two methods assume the local data distribution is imbalanced but the global one is balanced. Then, they use the global gradient to correct the local one. However, this kind of methods do not work when the global data distribution is imbalanced. Specifically, when the global gradient itself is computed on the imbalanced data, rather than the balanced one, it cannot alleviate the imbalance issue in the local gradient. In Figure 5, we show the test AUC score of the STL10 dataset, where we use the same experimental setting as that of Figure 1, i.e., both the local and global data distributions are imbalanced. It can be observed that

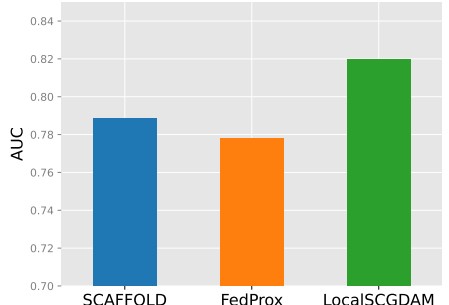

Figure 5: The test AUROC score for STL10.

our algorithm outperforms those two baselines with a large margin, which confirms the effectiveness of our algorithm in handling the global imbalanced data distribution.

## 7    Conclusion

In this paper, we developed a novel local stochastic compositional gradient descent ascent algorithm to solve the federated compositional deep AUC maximization problem. On the theoretical side, we established the convergence rate of our algorithm, which enjoys a linear speedup with respect to the number devices. On the empirical side, extensive experimental results on multiple imbalanced image classification tasks confirm the effectiveness of our algorithm.

## Acknowledcements

We thank anonymous reviewers for constructive comments. T. Yang was partially supported by NSF Career Award 2246753, NSF Grant 2246757.

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
