(17)

Based on these auxiliary functions, we provide the following lemmas to complete the proof.

**Lemma 1.** *Given Assumptions 1-4, the function $\mathbf{y}_*^{(k)}(\mathbf{x})$ is $L_{\mathbf{y}_*}$-Lipschitz continuous, where $L_{\mathbf{y}_*} = \frac{C_g L_f}{\mu}$.*

*Proof.* Since $\mathbf{y}_*^{(k)}(\mathbf{x}) = \arg\max_{\mathbf{y} \in \mathbb{R}^{d_2}} f^{(k)}\big(\frac{1}{K}\sum_{k'=1}^{K} g^{(k')}(\mathbf{x}), \mathbf{y}\big)$, according to the optimality condition, for any $\mathbf{x}_1, \mathbf{x}_2 \in \mathbb{R}^{d_1}$, we can get

$$\langle \nabla_{\mathbf{y}} f^{(k)}\big(\frac{1}{K}\sum_{k'=1}^{K} g^{(k')}(\mathbf{x}_1), \mathbf{y}_*^{(k)}(\mathbf{x}_1)\big), \mathbf{y}_*^{(k)}(\mathbf{x}_2) - \mathbf{y}_*^{(k)}(\mathbf{x}_1)\rangle \leq 0\,,$$

(18)

$$\langle \nabla_{\mathbf{y}} f^{(k)}\big(\frac{1}{K}\sum_{k'=1}^{K} g^{(k')}(\mathbf{x}_2), \mathbf{y}_*^{(k)}(\mathbf{x}_2)\big), \mathbf{y}_*^{(k)}(\mathbf{x}_1) - \mathbf{y}_*^{(k)}(\mathbf{x}_2)\rangle \leq 0\,,$$

As a result, we can get

$$\langle \mathbf{y}_*^{(k)}(\mathbf{x}_2) - \mathbf{y}_*^{(k)}(\mathbf{x}_1), \nabla_{\mathbf{y}} f^{(k)}\big(\frac{1}{K}\sum_{k'=1}^{K} g^{(k')}(\mathbf{x}_1), \mathbf{y}_*^{(k)}(\mathbf{x}_1)\big) - \nabla_{\mathbf{y}} f^{(k)}\big(\frac{1}{K}\sum_{k'=1}^{K} g^{(k')}(\mathbf{x}_2), \mathbf{y}_*^{(k)}(\mathbf{x}_2)\big)\rangle \leq 0\,.$$

(19)

Meanwhile, due to the strong monotonicity of the gradient with respect to $\mathbf{y}$, we can get

$$\langle \mathbf{y}_*^{(k)}(\mathbf{x}_2) - \mathbf{y}_*^{(k)}(\mathbf{x}_1), \nabla_{\mathbf{y}} f^{(k)}\big(\frac{1}{K}\sum_{k'=1}^{K} g^{(k')}(\mathbf{x}_1), \mathbf{y}_*^{(k)}(\mathbf{x}_2)\big) - \nabla_{\mathbf{y}} f^{(k)}\big(\frac{1}{K}\sum_{k'=1}^{K} g^{(k')}(\mathbf{x}_1), \mathbf{y}_*^{(k)}(\mathbf{x}_1)\big)\rangle$$
$$+ \mu\|\mathbf{y}_*^{(k)}(\mathbf{x}_2) - \mathbf{y}_*^{(k)}(\mathbf{x}_1)\|^2 \leq 0\,.$$

(20)

By adding above two inequalities together, we can get

$$\mu\|\mathbf{y}_*^{(k)}(\mathbf{x}_2) - \mathbf{y}_*^{(k)}(\mathbf{x}_1)\|^2$$

$$\leq \langle \mathbf{y}_*^{(k)}(\mathbf{x}_2) - \mathbf{y}_*^{(k)}(\mathbf{x}_1), \nabla_{\mathbf{y}} f^{(k)}\big(\frac{1}{K}\sum_{k'=1}^{K} g^{(k')}(\mathbf{x}_2), \mathbf{y}_*^{(k)}(\mathbf{x}_2)\big) - \nabla_{\mathbf{y}} f^{(k)}\big(\frac{1}{K}\sum_{k'=1}^{K} g^{(k')}(\mathbf{x}_1), \mathbf{y}_*^{(k)}(\mathbf{x}_2)\big)\rangle$$

$$\leq \|\mathbf{y}_*^{(k)}(\mathbf{x}_2) - \mathbf{y}_*^{(k)}(\mathbf{x}_1)\|\|\nabla_{\mathbf{y}} f^{(k)}\big(\frac{1}{K}\sum_{k'=1}^{K} g^{(k')}(\mathbf{x}_2), \mathbf{y}_*^{(k)}(\mathbf{x}_2)\big) - \nabla_{\mathbf{y}} f^{(k)}\big(\frac{1}{K}\sum_{k'=1}^{K} g^{(k')}(\mathbf{x}_1), \mathbf{y}_*^{(k)}(\mathbf{x}_2)\big)\|$$

$$\leq L_f\|\mathbf{y}_*^{(k)}(\mathbf{x}_2) - \mathbf{y}_*^{(k)}(\mathbf{x}_1)\|\|\frac{1}{K}\sum_{k'=1}^{K} g^{(k')}(\mathbf{x}_2) - \frac{1}{K}\sum_{k'=1}^{K} g^{(k')}(\mathbf{x}_1)\|$$

$$\leq C_g L_f\|\mathbf{y}_*^{(k)}(\mathbf{x}_2) - \mathbf{y}_*^{(k)}(\mathbf{x}_1)\|\|\mathbf{x}_2 - \mathbf{x}_1\|\,.$$

(21)

where the third step holds due to Assumption 1 and the last step holds due to Assumption 2. Therefore, we can get

$$\|\mathbf{y}_*^{(k)}(\mathbf{x}_2) - \mathbf{y}_*^{(k)}(\mathbf{x}_1)\| \leq \frac{C_g L_f}{\mu}\|\mathbf{x}_2 - \mathbf{x}_1\|\,.$$

(22)

$\square$

**Lemma 2.** *Given Assumptions 1-4, $\Phi^{(k)}$ is $L_\Phi$-smooth, where $L_\Phi = \frac{2C_g^2 L_f^2}{\mu} + C_f L_g$.*

*Proof.* Since $\Phi^{(k)}(\mathbf{x}) = f^{(k)}(\frac{1}{K}\sum_{k'=1}^{K} g^{(k')}(\mathbf{x}), \mathbf{y}_*(\mathbf{x}))$, we can get

$$\left\| \nabla\Phi^{(k)}(\mathbf{x}_1) - \nabla\Phi^{(k)}(\mathbf{x}_2) \right\|$$

$$= \left\| \Big(\frac{1}{K}\sum_{k'=1}^{K}\nabla g^{(k')}(\mathbf{x}_1)\Big)^T \nabla_g f^{(k)}(\frac{1}{K}\sum_{k'=1}^{K} g^{(k')}(\mathbf{x}_1), \mathbf{y}_*(\mathbf{x}_1)) \right.$$

$$- \Big(\frac{1}{K}\sum_{k'=1}^{K}\nabla g^{(k')}(\mathbf{x}_1)\Big)^T \nabla_g f^{(k)}(\frac{1}{K}\sum_{k'=1}^{K} g^{(k')}(\mathbf{x}_2), \mathbf{y}_*(\mathbf{x}_2))$$

$$+ \Big(\frac{1}{K}\sum_{k'=1}^{K}\nabla g^{(k')}(\mathbf{x}_1)\Big)^T \nabla_g f^{(k)}(\frac{1}{K}\sum_{k'=1}^{K} g^{(k')}(\mathbf{x}_2), \mathbf{y}_*(\mathbf{x}_2))$$

$$\left. - \Big(\frac{1}{K}\sum_{k'=1}^{K}\nabla g^{(k')}(\mathbf{x}_2)\Big)^T \nabla_g f^{(k)}(\frac{1}{K}\sum_{k'=1}^{K} g^{(k')}(\mathbf{x}_2), \mathbf{y}_*(\mathbf{x}_2)) \right\|$$

$$\leq \left\| \Big(\frac{1}{K}\sum_{k'=1}^{K}\nabla g^{(k')}(\mathbf{x}_1)\Big)^T \nabla_g f^{(k)}(\frac{1}{K}\sum_{k'=1}^{K} g^{(k')}(\mathbf{x}_1), \mathbf{y}_*(\mathbf{x}_1)) \right.$$

$$\left. - \Big(\frac{1}{K}\sum_{k'=1}^{K}\nabla g^{(k')}(\mathbf{x}_1)\Big)^T \nabla_g f^{(k)}(\frac{1}{K}\sum_{k'=1}^{K} g^{(k')}(\mathbf{x}_2), \mathbf{y}_*(\mathbf{x}_2)) \right\| \tag{23}$$

$$+ \left\| \Big(\frac{1}{K}\sum_{k'=1}^{K}\nabla g^{(k')}(\mathbf{x}_1)\Big)^T \nabla_g f^{(k)}(\frac{1}{K}\sum_{k'=1}^{K} g^{(k')}(\mathbf{x}_2), \mathbf{y}_*(\mathbf{x}_2)) \right.$$

$$\left. - \Big(\frac{1}{K}\sum_{k'=1}^{K}\nabla g^{(k')}(\mathbf{x}_2)\Big)^T \nabla_g f^{(k)}(\frac{1}{K}\sum_{k'=1}^{K} g^{(k')}(\mathbf{x}_2), \mathbf{y}_*(\mathbf{x}_2)) \right\|$$

$$\leq C_g \left\| \nabla_g f^{(k)}(\frac{1}{K}\sum_{k'=1}^{K} g^{(k')}(\mathbf{x}_1), \mathbf{y}_*(\mathbf{x}_1)) - \nabla_g f^{(k)}(\frac{1}{K}\sum_{k'=1}^{K} g^{(k')}(\mathbf{x}_2), \mathbf{y}_*(\mathbf{x}_2)) \right\|$$

$$+ C_f \left\| \frac{1}{K}\sum_{k'=1}^{K}\nabla g^{(k')}(\mathbf{x}_1) - \frac{1}{K}\sum_{k'=1}^{K}\nabla g^{(k')}(\mathbf{x}_2) \right\|$$

$$\leq C_g L_f \left\| \Big(\frac{1}{K}\sum_{k'=1}^{K} g^{(k')}(\mathbf{x}_1), \mathbf{y}_*(\mathbf{x}_1)\Big) - \Big(\frac{1}{K}\sum_{k'=1}^{K} g^{(k')}(\mathbf{x}_2), \mathbf{y}_*(\mathbf{x}_2)\Big) \right\| + C_f L_g \|\mathbf{x}_1 - \mathbf{x}_2\|$$

$$\leq C_g L_f (C_g \|\mathbf{x}_1 - \mathbf{x}_2\| + \frac{C_g L_f}{\mu}\|\mathbf{x}_1 - \mathbf{x}_2\|) + C_f L_g \|\mathbf{x}_1 - \mathbf{x}_2\|$$

$$\leq \Big(\frac{2C_g^2 L_f^2}{\mu} + C_f L_g\Big) \|\mathbf{x}_1 - \mathbf{x}_2\|,$$

where the second inequality holds due to Assumption 2, the third inequality holds due to Assumption 1, the fourth inequality holds due to Assumption 2 and Lemma 1, the last step holds due to $L_f/\mu > 1$. $\square$

**Lemma 3.** *Given Assumption 1-4 and $\eta \leq \frac{1}{2\gamma_x L_\Phi}$, we can get*

$$\mathbb{E}[\Phi(\bar{\mathbf{x}}_{t+1})] \leq \mathbb{E}[\Phi(\bar{\mathbf{x}}_t)] - \frac{\gamma_x \eta}{2}\mathbb{E}[\|\nabla\Phi(\bar{\mathbf{x}}_t)\|^2] - \frac{\gamma_x \eta}{4}\mathbb{E}[\|\bar{\mathbf{u}}_t\|^2] + 3\gamma_x \eta C_g^2 L_f^2 \mathbb{E}[\|\mathbf{y}_*(\bar{\mathbf{x}}_t) - \bar{\mathbf{y}}_t\|^2]$$

$$+ 6\gamma_x \eta (C_g^4 L_f^2 + C_f^2 L_g^2)\frac{1}{K}\sum_{k=1}^{K}\Big[\Big\|\bar{\mathbf{x}}_t - \mathbf{x}_t^{(k)}\Big\|^2\Big] + 3\gamma_x \eta C_g^2 L_f^2 \frac{1}{K}\sum_{k=1}^{K}\mathbb{E}\Big[\Big\|\bar{\mathbf{y}}_t - \mathbf{y}_t^{(k)}\Big\|^2\Big]$$

$$+ 3\gamma_x \eta \mathbb{E}\Big[\Big\|\frac{1}{K}\sum_{k=1}^{K}\nabla_\mathbf{x} f^{(k)}(g^{(k)}(\mathbf{x}_t^{(k)}), \mathbf{y}_t^{(k)}) - \bar{\mathbf{u}}_t\Big\|^2\Big].$$

$$\tag{24}$$

*Proof.* Because $\Phi(\bar{\mathbf{x}}_t)$ is $L_\phi$-smooth, we can get

$$
\mathbb{E}[\Phi(\bar{\mathbf{x}}_{t+1})] \le \mathbb{E}[\Phi(\bar{\mathbf{x}}_t)] + \mathbb{E}[\langle \nabla\Phi(\bar{\mathbf{x}}_t), \bar{\mathbf{x}}_{t+1} - \bar{\mathbf{x}}_t \rangle] + \frac{L_\Phi}{2}\mathbb{E}[\|\bar{\mathbf{x}}_{t+1} - \bar{\mathbf{x}}_t\|^2]
$$

$$
= \mathbb{E}[\Phi(\bar{\mathbf{x}}_t)] - \gamma_x\eta\,\mathbb{E}[\langle \nabla\Phi(\bar{\mathbf{x}}_t), \bar{\mathbf{u}}_t \rangle] + \frac{\gamma_x^2\eta^2 L_\Phi}{2}\mathbb{E}[\|\bar{\mathbf{u}}_t\|^2]
$$

$$
= \mathbb{E}[\Phi(\bar{\mathbf{x}}_t)] - \frac{\gamma_x\eta}{2}\mathbb{E}[\|\nabla\Phi(\bar{\mathbf{x}}_t)\|^2] + \Big(\frac{\gamma_x^2\eta^2 L_\Phi}{2} - \frac{\gamma_x\eta}{2}\Big)\mathbb{E}[\|\bar{\mathbf{u}}_t\|^2] + \frac{\gamma_x\eta}{2}\mathbb{E}[\|\nabla\Phi(\bar{\mathbf{x}}_t) - \bar{\mathbf{u}}_t\|^2]
$$

$$
\le \mathbb{E}[\Phi(\bar{\mathbf{x}}_t)] - \frac{\gamma_x\eta}{2}\mathbb{E}[\|\nabla\Phi(\bar{\mathbf{x}}_t)\|^2] - \frac{\gamma_x\eta}{4}\mathbb{E}[\|\bar{\mathbf{u}}_t\|^2]
$$

$$
+ 3\gamma_x\eta\,\mathbb{E}\Big[\underbrace{\Big\|\nabla\Phi(\bar{\mathbf{x}}_t) - \frac{1}{K}\sum_{k=1}^{K}\nabla_{\mathbf{x}}f^{(k)}\big(\frac{1}{K}\sum_{k'=1}^{K}g^{(k')}(\bar{\mathbf{x}}_t), \bar{\mathbf{y}}_t\big)\Big\|^2}_{T_1}\Big]
$$

$$
+ 3\gamma_x\eta\,\mathbb{E}\Big[\underbrace{\Big\|\frac{1}{K}\sum_{k=1}^{K}\nabla_{\mathbf{x}}f^{(k)}\big(\frac{1}{K}\sum_{k'=1}^{K}g^{(k')}(\bar{\mathbf{x}}_t), \bar{\mathbf{y}}_t\big) - \frac{1}{K}\sum_{k=1}^{K}\nabla_{\mathbf{x}}f^{(k)}(g^{(k)}(\mathbf{x}_t^{(k)}), \mathbf{y}_t^{(k)})\Big\|^2}_{T_2}\Big]
$$

$$
+ 3\gamma_x\eta\,\mathbb{E}\Big[\Big\|\frac{1}{K}\sum_{k=1}^{K}\nabla_{\mathbf{x}}f^{(k)}(g^{(k)}(\mathbf{x}_t^{(k)}), \mathbf{y}_t^{(k)}) - \bar{\mathbf{u}}_t\Big\|^2\Big]\,,
$$

$$\tag{25}$$

where the last inequality holds due to $\eta \le \frac{1}{2\gamma_x L_\phi}$. As for $T_1$, we can get

$$
T_1 = \mathbb{E}\Big[\Big\|\frac{1}{K}\sum_{k=1}^{K}\nabla\Phi^{(k)}(\bar{\mathbf{x}}_t) - \frac{1}{K}\sum_{k=1}^{K}\nabla_{\mathbf{x}}f^{(k)}\big(\frac{1}{K}\sum_{k'=1}^{K}g^{(k')}(\bar{\mathbf{x}}_t), \bar{\mathbf{y}}_t\big)\Big\|^2\Big]
$$

$$
\le \frac{1}{K}\sum_{k=1}^{K}\mathbb{E}\Big[\Big\|\Big(\frac{1}{K}\sum_{k'=1}^{K}\nabla g^{(k')}(\bar{\mathbf{x}}_t)\Big)^T \nabla_g f^{(k)}\big(\frac{1}{K}\sum_{k'=1}^{K}g^{(k')}(\bar{\mathbf{x}}_t), \mathbf{y}_*(\bar{\mathbf{x}}_t)\big)
$$

$$
- \Big(\frac{1}{K}\sum_{k'=1}^{K}\nabla g^{(k')}(\bar{\mathbf{x}}_t)\Big)^T \nabla_g f^{(k)}\big(\frac{1}{K}\sum_{k'=1}^{K}g^{(k')}(\bar{\mathbf{x}}_t), \bar{\mathbf{y}}_t\big)\Big\|^2\Big]
$$

$$
\le \frac{1}{K}\sum_{k=1}^{K}\mathbb{E}\Big[\Big\|\frac{1}{K}\sum_{k'=1}^{K}\nabla g^{(k')}(\bar{\mathbf{x}}_t)\Big\|^2\Big\|\nabla_g f^{(k)}\big(\frac{1}{K}\sum_{k'=1}^{K}g^{(k')}(\bar{\mathbf{x}}_t), \mathbf{y}_*(\bar{\mathbf{x}}_t)\big) - \nabla_g f^{(k)}\big(\frac{1}{K}\sum_{k'=1}^{K}g^{(k')}(\bar{\mathbf{x}}_t), \bar{\mathbf{y}}_t\big)\Big\|^2\Big]
$$

$$
\le C_g^2 L_f^2\,\mathbb{E}[\|\mathbf{y}_*(\bar{\mathbf{x}}_t) - \bar{\mathbf{y}}_t\|^2]\,,
$$

$$\tag{26}$$

where the last step holds due to Assumptions 1-2. As for $T_2$, we can get

$$
T_2 = \mathbb{E}\Big[\Big\|\frac{1}{K}\sum_{k=1}^{K}\nabla_{\mathbf{x}}f^{(k)}\big(\frac{1}{K}\sum_{k'=1}^{K}g^{(k')}(\bar{\mathbf{x}}_t), \bar{\mathbf{y}}_t\big) - \frac{1}{K}\sum_{k=1}^{K}\nabla_{\mathbf{x}}f^{(k)}(g^{(k)}(\bar{\mathbf{x}}_t), \mathbf{y}_t^{(k)})
$$

$$
+ \frac{1}{K}\sum_{k=1}^{K}\nabla_{\mathbf{x}}f^{(k)}(g^{(k)}(\bar{\mathbf{x}}_t), \mathbf{y}_t^{(k)}) - \frac{1}{K}\sum_{k=1}^{K}\nabla_{\mathbf{x}}f^{(k)}(g^{(k)}(\mathbf{x}_t^{(k)}), \mathbf{y}_t^{(k)})\Big\|^2\Big]
$$

$$
\le 2\,\mathbb{E}\Big[\underbrace{\Big\|\frac{1}{K}\sum_{k=1}^{K}\nabla_{\mathbf{x}}f^{(k)}\big(\frac{1}{K}\sum_{k'=1}^{K}g^{(k')}(\bar{\mathbf{x}}_t), \bar{\mathbf{y}}_t\big) - \frac{1}{K}\sum_{k=1}^{K}\nabla_{\mathbf{x}}f^{(k)}(g^{(k)}(\bar{\mathbf{x}}_t), \mathbf{y}_t^{(k)})\Big\|^2}_{T_3}\Big] \tag{27}
$$

$$
+ 2\,\mathbb{E}\Big[\underbrace{\Big\|\frac{1}{K}\sum_{k=1}^{K}\nabla_{\mathbf{x}}f^{(k)}(g^{(k)}(\bar{\mathbf{x}}_t), \mathbf{y}_t^{(k)}) - \frac{1}{K}\sum_{k=1}^{K}\nabla_{\mathbf{x}}f^{(k)}(g^{(k)}(\mathbf{x}_t^{(k)}), \mathbf{y}_t^{(k)})\Big\|^2}_{T_4}\Big]\,.
$$

Then, as for $T_3$, we can get

$$
\begin{aligned}
T_3 &\leq \frac{1}{K}\sum_{k=1}^{K}\mathbb{E}\Big[\Big\|\nabla_{\mathbf{x}}f^{(k)}\big(\frac{1}{K}\sum_{k'=1}^{K}g^{(k')}(\bar{\mathbf{x}}_t),\bar{\mathbf{y}}_t\big)-\nabla_{\mathbf{x}}f^{(k)}(g^{(k)}(\bar{\mathbf{x}}_t),\mathbf{y}_t^{(k)})\Big\|^2\Big]\\
&=\frac{1}{K}\sum_{k=1}^{K}\mathbb{E}\Big[\Big\|\big(\frac{1}{K}\sum_{k'=1}^{K}\nabla g^{(k')}(\bar{\mathbf{x}}_t)\big)^T\nabla_g f^{(k)}\big(\frac{1}{K}\sum_{k'=1}^{K}g^{(k')}(\bar{\mathbf{x}}_t),\bar{\mathbf{y}}_t\big)-\nabla g^{(k)}(\bar{\mathbf{x}}_t)^T\nabla_g f^{(k)}(g^{(k)}(\bar{\mathbf{x}}_t),\mathbf{y}_t^{(k)})\Big\|^2\Big]\\
&\leq C_g^2\frac{1}{K}\sum_{k=1}^{K}\mathbb{E}\Big[\Big\|\nabla_g f^{(k)}\big(\frac{1}{K}\sum_{k'=1}^{K}g^{(k')}(\bar{\mathbf{x}}_t),\bar{\mathbf{y}}_t\big)-\nabla_g f^{(k)}(g^{(k)}(\bar{\mathbf{x}}_t),\mathbf{y}_t^{(k)})\Big\|^2\Big]\\
&\leq C_g^2 L_f^2\frac{1}{K}\sum_{k=1}^{K}\mathbb{E}\Big[\Big\|\bar{\mathbf{y}}_t-\mathbf{y}_t^{(k)}\Big\|^2\Big],
\end{aligned}
$$
(28)

where the third step holds due to the homogeneous data distribution assumption and Assumption 2, the last step also holds due to the homogeneous data distribution assumption and Assumption 1.

As for $T_4$, we can get

$$
\begin{aligned}
T_4 &\leq \frac{1}{K}\sum_{k=1}^{K}\mathbb{E}\Big[\Big\|\nabla_{\mathbf{x}}f^{(k)}(g^{(k)}(\bar{\mathbf{x}}_t),\mathbf{y}_t^{(k)})-\nabla_{\mathbf{x}}f^{(k)}(g^{(k)}(\mathbf{x}_t^{(k)}),\mathbf{y}_t^{(k)})\Big\|^2\Big]\\
&=\frac{1}{K}\sum_{k=1}^{K}\mathbb{E}\Big[\Big\|\nabla g^{(k)}(\bar{\mathbf{x}}_t)^T\nabla_g f^{(k)}(g^{(k)}(\bar{\mathbf{x}}_t),\mathbf{y}_t^{(k)})-\nabla g^{(k)}(\mathbf{x}_t^{(k)})^T\nabla_g f^{(k)}(g^{(k)}(\mathbf{x}_t^{(k)}),\mathbf{y}_t^{(k)})\Big\|^2\Big]\\
&\leq 2\frac{1}{K}\sum_{k=1}^{K}\mathbb{E}\Big[\Big\|\nabla g^{(k)}(\bar{\mathbf{x}}_t)^T\nabla_g f^{(k)}(g^{(k)}(\bar{\mathbf{x}}_t),\mathbf{y}_t^{(k)})-\nabla g^{(k)}(\bar{\mathbf{x}}_t)^T\nabla_g f^{(k)}(g^{(k)}(\mathbf{x}_t^{(k)}),\mathbf{y}_t^{(k)})\Big\|^2\Big]\\
&\quad +2\frac{1}{K}\sum_{k=1}^{K}\mathbb{E}\Big[\Big\|\nabla g^{(k)}(\bar{\mathbf{x}}_t)^T\nabla_g f^{(k)}(g^{(k)}(\mathbf{x}_t^{(k)}),\mathbf{y}_t^{(k)})-\nabla g^{(k)}(\mathbf{x}_t^{(k)})^T\nabla_g f^{(k)}(g^{(k)}(\mathbf{x}_t^{(k)}),\mathbf{y}_t^{(k)})\Big\|^2\Big]\\
&\leq 2C_g^2 L_f^2\frac{1}{K}\sum_{k=1}^{K}\mathbb{E}\Big[\Big\|g^{(k)}(\bar{\mathbf{x}}_t)-g^{(k)}(\mathbf{x}_t^{(k)})\Big\|^2\Big]+2C_f^2\frac{1}{K}\sum_{k=1}^{K}\mathbb{E}\Big[\Big\|\nabla g^{(k)}(\bar{\mathbf{x}}_t)-\nabla g^{(k)}(\mathbf{x}_t^{(k)})\Big\|^2\Big]\\
&\leq 2C_g^4 L_f^2\frac{1}{K}\sum_{k=1}^{K}\mathbb{E}\Big[\Big\|\bar{\mathbf{x}}_t-\mathbf{x}_t^{(k)}\Big\|^2\Big]+2C_f^2 L_g^2\frac{1}{K}\sum_{k=1}^{K}\Big[\Big\|\bar{\mathbf{x}}_t-\mathbf{x}_t^{(k)}\Big\|^2\Big]\\
&=2(C_g^4 L_f^2+C_f^2 L_g^2)\frac{1}{K}\sum_{k=1}^{K}\Big[\Big\|\bar{\mathbf{x}}_t-\mathbf{x}_t^{(k)}\Big\|^2\Big],
\end{aligned}
$$
(29)

where fourth and fifth steps hold due to Assumptions 1 and 2. By combining $T_3$ and $T_4$, we can get

$$
T_2 \leq 2C_g^2 L_f^2\frac{1}{K}\sum_{k=1}^{K}\mathbb{E}\Big[\Big\|\bar{\mathbf{y}}_t-\mathbf{y}_t^{(k)}\Big\|^2\Big]+4(C_g^4 L_f^2+C_f^2 L_g^2)\frac{1}{K}\sum_{k=1}^{K}\Big[\Big\|\bar{\mathbf{x}}_t-\mathbf{x}_t^{(k)}\Big\|^2\Big]. \tag{30}
$$

By combining $T_1$ and $T_2$, we can get

$$
\begin{aligned}
\mathbb{E}[\Phi(\bar{\mathbf{x}}_{t+1})]&\leq\mathbb{E}[\Phi(\bar{\mathbf{x}}_t)]-\frac{\gamma_x\eta}{2}\mathbb{E}[\|\nabla\Phi(\bar{\mathbf{x}}_t)\|^2]-\frac{\gamma_x\eta}{4}\mathbb{E}[\|\bar{\mathbf{u}}_t\|^2]+3\gamma_x\eta C_g^2 L_f^2\mathbb{E}[\|\mathbf{y}_*(\bar{\mathbf{x}}_t)-\bar{\mathbf{y}}_t\|^2]\\
&\quad +12\gamma_x\eta(C_g^4 L_f^2+C_f^2 L_g^2)\frac{1}{K}\sum_{k=1}^{K}\Big[\Big\|\bar{\mathbf{x}}_t-\mathbf{x}_t^{(k)}\Big\|^2\Big]+6\gamma_x\eta C_g^2 L_f^2\frac{1}{K}\sum_{k=1}^{K}\mathbb{E}\Big[\Big\|\bar{\mathbf{y}}_t-\mathbf{y}_t^{(k)}\Big\|^2\Big]\\
&\quad +3\gamma_x\eta\mathbb{E}\Big[\Big\|\frac{1}{K}\sum_{k=1}^{K}\nabla_{\mathbf{x}}f^{(k)}(g^{(k)}(\mathbf{x}_t^{(k)}),\mathbf{y}_t^{(k)})-\bar{\mathbf{u}}_t\Big\|^2\Big],
\end{aligned}
$$
(31)

which completes the proof. $\qquad\square$

**Lemma 4.** *Given Assumption 1-4 and $\beta_x\eta \in (0,1)$, we have*

$$\frac{1}{K}\sum_{k=1}^{K}\mathbb{E}[\|\mathbf{u}_{t+1}^{(k)} - \bar{\mathbf{u}}_{t+1}\|^2] \le 6p^2\beta_x^2\eta^2 C_g^2 C_f^2 \ . \tag{32}$$

*Proof.*

$$\frac{1}{K}\sum_{k=1}^{K}\mathbb{E}[\|\mathbf{u}_{t+1}^{(k)} - \bar{\mathbf{u}}_{t+1}\|^2]$$

$$= \frac{1}{K}\sum_{k=1}^{K}\mathbb{E}\Big[\Big\|(1-\beta_x\eta)\mathbf{u}_t^{(k)} + \beta_x\eta\nabla g^{(k)}(\mathbf{x}_{t+1}^{(k)};\xi_{t+1}^{(k)})^T\nabla_g f^{(k)}(\mathbf{h}_{t+1}^{(k)},\mathbf{y}_{t+1}^{(k)};\zeta_{t+1}^{(k)})$$

$$- (1-\beta_x\eta)\bar{\mathbf{u}}_t - \beta_x\eta\frac{1}{K}\sum_{k'=1}^{K}\nabla g^{(k')}(\mathbf{x}_{t+1}^{(k')};\xi_{t+1}^{(k')})^T\nabla_g f^{(k')}(\mathbf{h}_{t+1}^{(k')},\mathbf{y}_{t+1}^{(k')};\zeta_{t+1}^{(k')})\Big\|^2\Big]$$

$$\le (1-\beta_x\eta)^2(1+\frac{1}{p})\frac{1}{K}\sum_{k=1}^{K}\mathbb{E}[\|\mathbf{u}_t^{(k)} - \bar{\mathbf{u}}_t\|^2]$$

$$+ (1+p)\beta_x^2\eta^2\frac{1}{K}\sum_{k=1}^{K}\mathbb{E}\Big[\Big\|\nabla g^{(k)}(\mathbf{x}_{t+1}^{(k)};\xi_{t+1}^{(k)})^T\nabla_g f^{(k)}(\mathbf{h}_{t+1}^{(k)},\mathbf{y}_{t+1}^{(k)};\zeta_{t+1}^{(k)})$$

$$- \frac{1}{K}\sum_{k'=1}^{K}\nabla g^{(k')}(\mathbf{x}_{t+1}^{(k')};\xi_{t+1}^{(k')})^T\nabla_g f^{(k')}(\mathbf{h}_{t+1}^{(k')},\mathbf{y}_{t+1}^{(k')};\zeta_{t+1}^{(k')})\Big\|^2\Big]$$

$$\le (1+\frac{1}{p})\frac{1}{K}\sum_{k=1}^{K}\mathbb{E}[\|\mathbf{u}_t^{(k)} - \bar{\mathbf{u}}_t\|^2] + 2p\beta_x^2\eta^2\frac{1}{K}\sum_{k=1}^{K}\mathbb{E}\Big[\Big\|\nabla g^{(k)}(\mathbf{x}_{t+1}^{(k)};\xi_{t+1}^{(k)})^T\nabla_g f^{(k)}(\mathbf{h}_{t+1}^{(k)},\mathbf{y}_{t+1}^{(k)};\zeta_{t+1}^{(k)})$$

$$- \frac{1}{K}\sum_{k'=1}^{K}\nabla g^{(k')}(\mathbf{x}_{t+1}^{(k')};\xi_{t+1}^{(k')})^T\nabla_g f^{(k')}(\mathbf{h}_{t+1}^{(k')},\mathbf{y}_{t+1}^{(k')};\zeta_{t+1}^{(k')})\Big\|^2\Big]$$

$$\le (1+\frac{1}{p})\frac{1}{K}\sum_{k=1}^{K}\mathbb{E}[\|\mathbf{u}_t^{(k)} - \bar{\mathbf{u}}_t\|^2] + 2p\beta_x^2\eta^2\frac{1}{K}\sum_{k=1}^{K}\mathbb{E}\Big[\Big\|\nabla g^{(k)}(\mathbf{x}_{t+1}^{(k)};\xi_{t+1}^{(k)})^T\nabla_g f^{(k)}(\mathbf{h}_{t+1}^{(k)},\mathbf{y}_{t+1}^{(k)};\zeta_{t+1}^{(k)})\Big\|^2\Big]$$

$$\le (1+\frac{1}{p})\frac{1}{K}\sum_{k=1}^{K}\mathbb{E}[\|\mathbf{u}_t^{(k)} - \bar{\mathbf{u}}_t\|^2] + 2p\beta_x^2\eta^2 C_g^2 C_f^2$$

$$\le 2p\beta_x^2\eta^2 C_g^2 C_f^2 \sum_{t'=s_t p}^{t}(1+\frac{1}{p})^{t-t'}$$

$$\le 6p^2\beta_x^2\eta^2 C_g^2 C_f^2 \ ,$$

$$\tag{33}$$

where $s_t = \lfloor(t+1)/p\rfloor$, the third step holds due to $\beta_x\eta \in (0,1)$ and $1+p < 2p$, the fifth step holds due to Assumption 2, the last step holds due to $(1+\frac{1}{p})^p < 3$. $\qquad\square$

**Lemma 5.** *Given Assumption 1-4, $\beta_y\eta \in (0,1)$, and $\eta \le \frac{1}{10p^2\gamma_y L_f}$ , we have*

$$\frac{1}{T}\sum_{t=0}^{T-1}\frac{1}{K}\sum_{k=1}^{K}\mathbb{E}[\|\mathbf{v}_t^{(k)} - \bar{\mathbf{v}}_t\|^2]$$

$$\le 3456\beta_y^2\alpha^2 p^4\eta^4 C_g^2 L_f^2\sigma_g^2 + 41472\beta_y^2\alpha^2\gamma_x^2\beta_x^2 p^8\eta^8 C_g^6 C_f^2 L_f^2 + 96\beta_y^2 p^2\eta^2\sigma_f^2 \ . \tag{34}$$

*Proof.*

$$\frac{1}{K}\sum_{k=1}^{K}\mathbb{E}[\|\mathbf{v}_{t+1}^{(k)}-\bar{\mathbf{v}}_{t+1}\|^2]$$

$$=\frac{1}{K}\sum_{k=1}^{K}\mathbb{E}\Big[\Big\|(1-\beta_y\eta)\mathbf{v}_t^{(k)}+\beta_y\eta\nabla_y f^{(k)}(\mathbf{h}_{t+1}^{(k)},\mathbf{y}_{t+1}^{(k)};\zeta_{t+1}^{(k)})$$

$$-(1-\beta_y\eta)\bar{\mathbf{v}}_t-\beta_y\eta\frac{1}{K}\sum_{k'=1}^{K}\nabla_y f^{(k')}(\mathbf{h}_{t+1}^{(k')},\mathbf{y}_{t+1}^{(k')};\zeta_{t+1}^{(k')})\Big\|^2\Big]$$

$$\leq(1-\beta_y\eta)^2(1+\frac{1}{p})\frac{1}{K}\sum_{k=1}^{K}\mathbb{E}[\|\mathbf{v}_t^{(k)}-\bar{\mathbf{v}}_t\|^2]+(1+p)\beta_y^2\eta^2\frac{1}{K}\sum_{k=1}^{K}\mathbb{E}\Big[\Big\|\nabla_y f^{(k)}(\mathbf{h}_{t+1}^{(k)},\mathbf{y}_{t+1}^{(k)};\zeta_{t+1}^{(k)})$$

$$-\frac{1}{K}\sum_{k'=1}^{K}\nabla_y f^{(k')}(\mathbf{h}_{t+1}^{(k')},\mathbf{y}_{t+1}^{(k')};\zeta_{t+1}^{(k')})\Big\|^2\Big]$$

$$\leq(1+\frac{1}{p})\frac{1}{K}\sum_{k=1}^{K}\mathbb{E}[\|\mathbf{v}_t^{(k)}-\bar{\mathbf{v}}_t\|^2]+2p\beta_y^2\eta^2\frac{1}{K}\sum_{k=1}^{K}\mathbb{E}\Big[\Big\|\nabla_y f^{(k)}(\mathbf{h}_{t+1}^{(k)},\mathbf{y}_{t+1}^{(k)};\zeta_{t+1}^{(k)})$$

$$-\frac{1}{K}\sum_{k'=1}^{K}\nabla_y f^{(k')}(\mathbf{h}_{t+1}^{(k')},\mathbf{y}_{t+1}^{(k')};\zeta_{t+1}^{(k')})\Big\|^2\Big]\,,$$

(35)

where the third step holds due to $\beta_y\eta\in(0,1)$ and $1+p<2p$.

The last term can be bounded as follows:

$$\frac{1}{K}\sum_{k=1}^{K}\mathbb{E}\Big[\Big\|\nabla_y f^{(k)}(\mathbf{h}_{t+1}^{(k)},\mathbf{y}_{t+1}^{(k)};\zeta_{t+1}^{(k)})-\frac{1}{K}\sum_{k'=1}^{K}\nabla_y f^{(k')}(\mathbf{h}_{t+1}^{(k')},\mathbf{y}_{t+1}^{(k')};\zeta_{t+1}^{(k')})\Big\|^2\Big]$$

$$\leq 4\frac{1}{K}\sum_{k=1}^{K}\mathbb{E}\Big[\Big\|\nabla_y f^{(k)}(\mathbf{h}_{t+1}^{(k)},\mathbf{y}_{t+1}^{(k)};\zeta_{t+1}^{(k)})-\nabla_y f^{(k)}(\mathbf{h}_{t+1}^{(k)},\mathbf{y}_{t+1}^{(k)})\Big\|^2\Big]$$

$$+4\frac{1}{K}\sum_{k=1}^{K}\mathbb{E}\Big[\Big\|\nabla_y f^{(k)}(\mathbf{h}_{t+1}^{(k)},\mathbf{y}_{t+1}^{(k)})-\nabla_y f^{(k)}(\bar{\mathbf{h}}_{t+1},\bar{\mathbf{y}}_{t+1})\Big\|^2\Big]$$

$$+4\frac{1}{K}\sum_{k=1}^{K}\mathbb{E}\Big[\Big\|\nabla_y f^{(k)}(\bar{\mathbf{h}}_{t+1},\bar{\mathbf{y}}_{t+1})-\frac{1}{K}\sum_{k'=1}^{K}\nabla_y f^{(k')}(\mathbf{h}_{t+1}^{(k')},\mathbf{y}_{t+1}^{(k')})\Big\|^2\Big]$$

$$+4\frac{1}{K}\sum_{k=1}^{K}\mathbb{E}\Big[\Big\|\frac{1}{K}\sum_{k'=1}^{K}\nabla_y f^{(k')}(\mathbf{h}_{t+1}^{(k')},\mathbf{y}_{t+1}^{(k')})-\frac{1}{K}\sum_{k'=1}^{K}\nabla_y f^{(k')}(\mathbf{h}_{t+1}^{(k')},\mathbf{y}_{t+1}^{(k')};\zeta_{t+1}^{(k')})\Big\|^2\Big]$$

$$\leq 8\sigma_f^2+8L_f^2\frac{1}{K}\sum_{k=1}^{K}\mathbb{E}[\|\mathbf{h}_{t+1}^{(k)}-\bar{\mathbf{h}}_{t+1}\|^2]+8L_f^2\frac{1}{K}\sum_{k=1}^{K}\mathbb{E}[\|\mathbf{y}_{t+1}^{(k)}-\bar{\mathbf{y}}_{t+1}\|^2]\,.$$

(36)

Then, by combining these two inequalities, we can get

$$\frac{1}{K}\sum_{k=1}^{K}\mathbb{E}[\|\mathbf{v}_{t+1}^{(k)}-\bar{\mathbf{v}}_{t+1}\|^2]\leq(1+\frac{1}{p})\frac{1}{K}\sum_{k=1}^{K}\mathbb{E}[\|\mathbf{v}_t^{(k)}-\bar{\mathbf{v}}_t\|^2]+16p\beta_y^2\eta^2\sigma_f^2$$

$$+16p\beta_y^2\eta^2 L_f^2\frac{1}{K}\sum_{k=1}^{K}\mathbb{E}[\|\mathbf{h}_{t+1}^{(k)}-\bar{\mathbf{h}}_{t+1}\|^2]+16p\beta_y^2\eta^2 L_f^2\frac{1}{K}\sum_{k=1}^{K}\mathbb{E}[\|\mathbf{y}_{t+1}^{(k)}-\bar{\mathbf{y}}_{t+1}\|^2]\,.$$

(37)

In addition, we have

$$\frac{1}{K}\sum_{k=1}^{K}\mathbb{E}[\|\mathbf{y}_{t+1}^{(k)} - \bar{\mathbf{y}}_{t+1}\|^2] = \frac{1}{K}\sum_{k=1}^{K}\mathbb{E}[\|\mathbf{y}_{s_t p}^{(k)} + \gamma_y\eta\sum_{t'=s_t p}^{t}\mathbf{v}_{t'}^{(k)} - \bar{\mathbf{y}}_{s_t p} - \gamma_y\eta\sum_{t'=s_t p}^{t}\bar{\mathbf{v}}_{t'}\|^2]$$

$$\leq p\gamma_y^2\eta^2\frac{1}{K}\sum_{k=1}^{K}\sum_{t'=s_t p}^{t}\mathbb{E}[\|\mathbf{v}_{t'}^{(k)} - \bar{\mathbf{v}}_{t'}\|^2],$$

(38)

where $s_t = \lfloor(t+1)/p\rfloor$. Thus, we can get

$$\frac{1}{K}\sum_{k=1}^{K}\mathbb{E}[\|\mathbf{v}_{t+1}^{(k)} - \bar{\mathbf{v}}_{t+1}\|^2] \leq (1+\frac{1}{p})\frac{1}{K}\sum_{k=1}^{K}\mathbb{E}[\|\mathbf{v}_t^{(k)} - \bar{\mathbf{v}}_t\|^2] + 16p\beta_y^2\eta^2\sigma_f^2$$

$$+ 16p\beta_y^2\eta^2 L_f^2\frac{1}{K}\sum_{k=1}^{K}\mathbb{E}[\|\mathbf{h}_{t+1}^{(k)} - \bar{\mathbf{h}}_{t+1}\|^2] + 16p^2\gamma_y^2\beta_y^2\eta^4 L_f^2\frac{1}{K}\sum_{k=1}^{K}\sum_{t'=s_t p}^{t}\mathbb{E}[\|\mathbf{v}_{t'}^{(k)} - \bar{\mathbf{v}}_{t'}\|^2]$$

$$\leq 16p^2\gamma_y^2\beta_y^2\eta^4 L_f^2\sum_{t'=s_t p}^{t}(1+\frac{1}{p})^{t-t'}\frac{1}{K}\sum_{k=1}^{K}\sum_{t''=s_t p}^{t'}\mathbb{E}[\|\mathbf{v}_{t''}^{(k)} - \bar{\mathbf{v}}_{t''}\|^2]$$

$$+ 16p\beta_y^2\eta^2 L_f^2\sum_{t'=s_t p}^{t}(1+\frac{1}{p})^{t-t'}\frac{1}{K}\sum_{k=1}^{K}\mathbb{E}[\|\mathbf{h}_{t'+1}^{(k)} - \bar{\mathbf{h}}_{t'+1}\|^2] + 16p\beta_y^2\eta^2\sigma_f^2\sum_{t'=s_t p}^{t}(1+\frac{1}{p})^{t-t'}$$

$$\leq 48p^3\gamma_y^2\beta_y^2\eta^4 L_f^2\frac{1}{K}\sum_{k=1}^{K}\sum_{t'=s_t p}^{t}\mathbb{E}[\|\mathbf{v}_{t'}^{(k)} - \bar{\mathbf{v}}_{t'}\|^2]$$

$$+ 48p\beta_y^2\eta^2 L_f^2\sum_{t'=s_t p}^{t}\frac{1}{K}\sum_{k=1}^{K}\mathbb{E}[\|\mathbf{h}_{t'+1}^{(k)} - \bar{\mathbf{h}}_{t'+1}\|^2] + 48p^2\beta_y^2\eta^2\sigma_f^2,$$

(39)

where the last step holds due to $(1+\frac{1}{p})^p < 3$. Then, we can get

$$\frac{1}{T}\sum_{t=0}^{T-1}\frac{1}{K}\sum_{k=1}^{K}\mathbb{E}[\|\mathbf{v}_t^{(k)} - \bar{\mathbf{v}}_t\|^2] \leq 48p^4\gamma_y^2\beta_y^2\eta^4 L_f^2\frac{1}{T}\sum_{t=0}^{T-1}\frac{1}{K}\sum_{k=1}^{K}\mathbb{E}[\|\mathbf{v}_t^{(k)} - \bar{\mathbf{v}}_t\|^2]$$

$$+ 48p^2\beta_y^2\eta^2 L_f^2\frac{1}{T}\sum_{t=0}^{T-1}\frac{1}{K}\sum_{k=1}^{K}\mathbb{E}[\|\mathbf{h}_t^{(k)} - \bar{\mathbf{h}}_t\|^2] + 48p^2\beta_y^2\eta^2\sigma_f^2$$

$$\leq 48p^4\gamma_y^2\eta^2 L_f^2\frac{1}{T}\sum_{t=0}^{T-1}\frac{1}{K}\sum_{k=1}^{K}\mathbb{E}[\|\mathbf{v}_t^{(k)} - \bar{\mathbf{v}}_t\|^2] + 48p^2\beta_y^2\eta^2 L_f^2\frac{1}{T}\sum_{t=0}^{T-1}\frac{1}{K}\sum_{k=1}^{K}\mathbb{E}[\|\mathbf{h}_t^{(k)} - \bar{\mathbf{h}}_t\|^2]$$

$$+ 48p^2\beta_y^2\eta^2\sigma_f^2,$$

(40)

where the last step holds due to $\beta_y\eta \leq 1$. By setting $\eta \leq \frac{1}{10p^2\gamma_y L_f}$ such that $1 - 48p^4\gamma_y^2\eta^2 L_f^2 \geq \frac{1}{2}$, we can get

$$\frac{1}{T}\sum_{t=0}^{T-1}\frac{1}{K}\sum_{k=1}^{K}\mathbb{E}[\|\mathbf{v}_t^{(k)} - \bar{\mathbf{v}}_t\|^2] \leq 96p^2\beta_y^2\eta^2 L_f^2\frac{1}{T}\sum_{t=0}^{T-1}\frac{1}{K}\sum_{k=1}^{K}\mathbb{E}[\|\mathbf{h}_t^{(k)} - \bar{\mathbf{h}}_t\|^2] + 96p^2\beta_y^2\eta^2\sigma_f^2$$

$$\leq 3456\beta_y^2\alpha^2 p^4\eta^4 C_g^2 L_f^2\sigma_g^2 + 41472\beta_y^2\alpha^2\gamma_x^2\beta_x^2 p^8\eta^8 C_g^6 C_f^2 L_f^2 + 96\beta_y^2 p^2\eta^2\sigma_f^2.$$

(41)

where the last step holds due to Lemma 9.

□

**Lemma 6.** *Given Assumption 1-4, we have*

$$\frac{1}{K}\sum_{k=1}^{K}\mathbb{E}[\|\mathbf{x}_{t+1}^{(k)} - \bar{\mathbf{x}}_{t+1}\|^2] \leq 6\gamma_x^2\beta_x^2 p^4\eta^4 C_g^2 C_f^2.$$

(42)

*Proof.*

$$\frac{1}{K}\sum_{k=1}^{K}\mathbb{E}[\|\mathbf{x}_{t+1}^{(k)} - \bar{\mathbf{x}}_{t+1}\|^2] = \frac{1}{K}\sum_{k=1}^{K}\mathbb{E}[\|\mathbf{x}_{s_t p}^{(k)} - \gamma_x \eta \sum_{t'=s_t p}^{t}\mathbf{u}_{t'}^{(k)} - \bar{\mathbf{x}}_{s_t p} + \gamma_x \eta \sum_{t'=s_t p}^{t}\bar{\mathbf{u}}_{t'}\|^2]$$

$$\leq p\gamma_x^2\eta^2\frac{1}{K}\sum_{k=1}^{K}\sum_{t'=s_t p}^{t}\mathbb{E}[\|\mathbf{u}_{t'}^{(k)} - \bar{\mathbf{u}}_{t'}\|^2] \leq 6\gamma_x^2\beta_x^2 p^4\eta^4 C_g^2 C_f^2 \,,$$

$$(43)$$

where $s_t = \lfloor(t+1)/p\rfloor$, the last step holds due to Lemma 4. $\qquad\square$

**Lemma 7.** *Given Assumption 1-4, we have*

$$\frac{1}{T}\sum_{t=0}^{T-1}\frac{1}{K}\sum_{k=1}^{K}\mathbb{E}[\|\mathbf{y}_t^{(k)} - \bar{\mathbf{y}}_t\|^2]$$

$$\leq 3456\gamma_y^2\beta_y^2\alpha^2 p^6\eta^6 C_g^2 L_f^2\sigma_g^2 + 41472\gamma_y^2\beta_y^2\alpha^2\gamma_x^2\beta_x^2 p^{10}\eta^{10} C_g^6 C_f^2 L_f^2 + 96\gamma_y^2\beta_y^2 p^4\eta^4\sigma_f^2 \,.$$

$$(44)$$

*Proof.*

$$\frac{1}{K}\sum_{k=1}^{K}\mathbb{E}[\|\mathbf{y}_{t+1}^{(k)} - \bar{\mathbf{y}}_{t+1}\|^2] = \frac{1}{K}\sum_{k=1}^{K}\mathbb{E}[\|\mathbf{y}_{s_t p}^{(k)} + \gamma_y \eta \sum_{t'=s_t p}^{t}\mathbf{v}_{t'}^{(k)} - \bar{\mathbf{y}}_{s_t p} - \gamma_y \eta \sum_{t'=s_t p}^{t}\bar{\mathbf{v}}_{t'}\|^2]$$

$$\leq p\gamma_y^2\eta^2\frac{1}{K}\sum_{k=1}^{K}\sum_{t'=s_t p}^{t}\mathbb{E}[\|\mathbf{v}_{t'}^{(k)} - \bar{\mathbf{v}}_{t'}\|^2] \,,$$

$$(45)$$

where $s_t = \lfloor(t+1)/p\rfloor$. Then, it is easy to know

$$\frac{1}{T}\sum_{t=0}^{T-1}\frac{1}{K}\sum_{k=1}^{K}\mathbb{E}[\|\mathbf{y}_t^{(k)} - \bar{\mathbf{y}}_t\|^2] \leq p^2\gamma_y^2\eta^2\frac{1}{T}\sum_{t=0}^{T-1}\frac{1}{K}\sum_{k=1}^{K}\mathbb{E}[\|\mathbf{v}_t^{(k)} - \bar{\mathbf{v}}_t\|^2]$$

$$\leq 3456\gamma_y^2\beta_y^2\alpha^2 p^6\eta^6 C_g^2 L_f^2\sigma_g^2 + 41472\gamma_y^2\beta_y^2\alpha^2\gamma_x^2\beta_x^2 p^{10}\eta^{10} C_g^6 C_f^2 L_f^2 + 96\gamma_y^2\beta_y^2 p^4\eta^4\sigma_f^2 \,.$$

$$(46)$$

where the last step holds due to Lemma 5.

$\qquad\square$

**Lemma 8.** *Given Assumptions 1-4, we can get*

$$\frac{1}{K}\sum_{k=1}^{K}\mathbb{E}\Big[\Big\|g^{(k)}(\mathbf{x}_t^{(k)}) - \frac{1}{K}\sum_{k'=1}^{K}g^{(k')}(\mathbf{x}_t^{(k')})\Big\|^2\Big] \leq 24\gamma_x^2\beta_x^2 p^4\eta^4 C_g^4 C_f^2 \,. \qquad (47)$$

*Proof.*

$$\frac{1}{K}\sum_{k=1}^{K}\mathbb{E}\Big[\Big\|g^{(k)}(\mathbf{x}_t^{(k)}) - \frac{1}{K}\sum_{k'=1}^{K}g^{(k')}(\mathbf{x}_t^{(k')})\Big\|^2\Big]$$

$$\leq \frac{1}{K}\sum_{k=1}^{K}\mathbb{E}\Big[\Big\|g^{(k)}(\mathbf{x}_t^{(k)}) - g^{(k)}(\bar{\mathbf{x}}_t) + g(\bar{\mathbf{x}}_t) - \frac{1}{K}\sum_{k'=1}^{K}g^{(k')}(\mathbf{x}_t^{(k')})\Big\|^2\Big]$$

$$\leq 2\frac{1}{K}\sum_{k=1}^{K}\mathbb{E}\Big[\Big\|g^{(k)}(\mathbf{x}_t^{(k)}) - g^{(k)}(\bar{\mathbf{x}}_t)\Big\|^2\Big] + 2\frac{1}{K}\sum_{k=1}^{K}\mathbb{E}\Big[\Big\|g(\bar{\mathbf{x}}_t) - \frac{1}{K}\sum_{k'=1}^{K}g^{(k')}(\mathbf{x}_t^{(k')})\Big\|^2\Big]$$

$$\leq 4C_g^2\frac{1}{K}\sum_{k=1}^{K}\mathbb{E}[\|\mathbf{x}_t^{(k)} - \bar{\mathbf{x}}_t\|^2]$$

$$\leq 24\gamma_x^2\beta_x^2 p^4\eta^4 C_g^4 C_f^2 \,,$$

$$(48)$$

where the second step holds due to $g^{(k)}(\bar{\mathbf{x}}_t) = g(\bar{\mathbf{x}}_t)$ for the homogeneous data distribution, the third step holds due to Assumption 2, the last step holds due to Lemma 6. $\qquad\square$

**Lemma 9.** *Given Assumptions 1-4, we can get*

$$\frac{1}{K}\sum_{k=1}^{K}\mathbb{E}\Big[\Big\|\mathbf{h}_{t+1}^{(k)} - \frac{1}{K}\sum_{k'=1}^{K}\mathbf{h}_{t+1}^{(k')}\Big\|^2\Big] \le 36\alpha^2 p^2\eta^2 C_g^2\sigma_g^2 + 432\alpha^2\gamma_x^2\beta_x^2 p^6\eta^6 C_g^6 C_f^2 . \qquad (49)$$

*Proof.*

$$\frac{1}{K}\sum_{k=1}^{K}\mathbb{E}\Big[\Big\|\mathbf{h}_{t+1}^{(k)} - \frac{1}{K}\sum_{k'=1}^{K}\mathbf{h}_{t+1}^{(k')}\Big\|^2\Big]$$

$$= \frac{1}{K}\sum_{k=1}^{K}\mathbb{E}\Big[\Big\|(1-\alpha\eta)\mathbf{h}_t^{(k)} + \alpha\eta g^{(k)}(\mathbf{x}_{t+1}^{(k)};\xi_{t+1}^{(k)}) - (1-\alpha\eta)\frac{1}{K}\sum_{k'=1}^{K}\mathbf{h}_t^{(k')} - \alpha\eta\frac{1}{K}\sum_{k'=1}^{K}g^{(k')}(\mathbf{x}_{t+1}^{(k')};\xi_{t+1}^{(k')})\Big\|^2\Big]$$

$$\le (1-\alpha\eta)^2(1+\frac{1}{p})\frac{1}{K}\sum_{k=1}^{K}\mathbb{E}\Big[\Big\|\mathbf{h}_t^{(k)} - \frac{1}{K}\sum_{k'=1}^{K}\mathbf{h}_t^{(k')}\Big\|^2\Big]$$

$$+ \alpha^2\eta^2(1+p)\frac{1}{K}\sum_{k=1}^{K}\mathbb{E}\Big[\Big\|g^{(k)}(\mathbf{x}_{t+1}^{(k)};\xi_{t+1}^{(k)}) - \frac{1}{K}\sum_{k'=1}^{K}g^{(k')}(\mathbf{x}_{t+1}^{(k')};\xi_{t+1}^{(k')})\Big\|^2\Big]$$

$$\le (1+\frac{1}{p})\frac{1}{K}\sum_{k=1}^{K}\mathbb{E}\Big[\Big\|\mathbf{h}_t^{(k)} - \frac{1}{K}\sum_{k'=1}^{K}\mathbf{h}_t^{(k')}\Big\|^2\Big] + 2p\alpha^2\eta^2\frac{1}{K}\sum_{k=1}^{K}\mathbb{E}\Big[\Big\|g^{(k)}(\mathbf{x}_{t+1}^{(k)};\xi_{t+1}^{(k)}) - \frac{1}{K}\sum_{k'=1}^{K}g^{(k')}(\mathbf{x}_{t+1}^{(k')};\xi_{t+1}^{(k')})\Big\|^2\Big]$$

$$\le 2p\alpha^2\eta^2 C_g^2\sum_{t'=s_t p}^{t}(1+\frac{1}{p})^{t-t'}\frac{1}{K}\sum_{k=1}^{K}\mathbb{E}\Big[\Big\|g^{(k)}(\mathbf{x}_{t'}^{(k)};\xi_{t'}^{(k)}) - \frac{1}{K}\sum_{k'=1}^{K}g^{(k')}(\mathbf{x}_{t'}^{(k')};\xi_{t'}^{(k')})\Big\|^2\Big] ,$$

$$(50)$$

where $s_t = \lfloor (t+1)/p \rfloor$, the third step holds due to $\alpha\eta \in (0,1)$ and $1+p < 2p$. In addition, we can get

$$\frac{1}{K}\sum_{k=1}^{K}\mathbb{E}\Big[\Big\|g^{(k)}(\mathbf{x}_t^{(k)};\xi_t^{(k)}) - \frac{1}{K}\sum_{k'=1}^{K}g^{(k')}(\mathbf{x}_t^{(k')};\xi_t^{(k')})\Big\|^2\Big]$$

$$= \frac{1}{K}\sum_{k=1}^{K}\mathbb{E}\Big[\Big\|g^{(k)}(\mathbf{x}_t^{(k)};\xi_t^{(k)}) - g^{(k)}(\mathbf{x}_t^{(k)}) + g^{(k)}(\mathbf{x}_t^{(k)})$$

$$- \frac{1}{K}\sum_{k'=1}^{K}g^{(k')}(\mathbf{x}_t^{(k')}) + \frac{1}{K}\sum_{k'=1}^{K}g^{(k')}(\mathbf{x}_t^{(k')}) - \frac{1}{K}\sum_{k'=1}^{K}g^{(k')}(\mathbf{x}_t^{(k')};\xi_t^{(k')})\Big\|^2\Big] \qquad (51)$$

$$\le 6\sigma_g^2 + 3\frac{1}{K}\sum_{k=1}^{K}\mathbb{E}\Big[\Big\|g^{(k)}(\mathbf{x}_t^{(k)}) - \frac{1}{K}\sum_{k'=1}^{K}g^{(k')}(\mathbf{x}_t^{(k')})\Big\|^2\Big]$$

$$\le 6\sigma_g^2 + 72\gamma_x^2\beta_x^2 p^4\eta^4 C_g^4 C_f^2 ,$$

where the last step holds due to Lemma 8. Therefore, we can get

$$\frac{1}{K}\sum_{k=1}^{K}\mathbb{E}\Big[\Big\|\mathbf{h}_{t+1}^{(k)} - \frac{1}{K}\sum_{k'=1}^{K}\mathbf{h}_{t+1}^{(k')}\Big\|^2\Big]$$

$$\le 2p\alpha^2\eta^2 C_g^2\sum_{t'=s_t p}^{t}(1+\frac{1}{p})^{t-t'}\frac{1}{K}\sum_{k=1}^{K}\mathbb{E}\Big[\Big\|g^{(k)}(\mathbf{x}_{t'}^{(k)};\xi_{t'}^{(k)}) - \frac{1}{K}\sum_{k'=1}^{K}g^{(k')}(\mathbf{x}_{t'}^{(k')};\xi_{t'}^{(k')})\Big\|^2\Big] \qquad (52)$$

$$\le 2p\alpha^2\eta^2 C_g^2(6\sigma_g^2 + 72\gamma_x^2\beta_x^2 p^4\eta^4 C_g^4 C_f^2)\sum_{t'=s_t p}^{t}(1+\frac{1}{p})^{t-t'}$$

$$\le 36\alpha^2 p^2\eta^2 C_g^2\sigma_g^2 + 432\alpha^2\gamma_x^2\beta_x^2 p^6\eta^6 C_g^6 C_f^2 .$$

where the last step holds due to $(1+\frac{1}{p})^p < 3$. $\qquad\qquad\square$

**Lemma 10.** *Given Assumptions 1-4, we can get*

$$\frac{1}{K}\sum_{k=1}^{K}\mathbb{E}\Big[\Big\|\nabla_x f^{(k)}(g^{(k)}(\mathbf{x}_t^{(k)}),\mathbf{y}_t^{(k)}) - \nabla_x f^{(k)}(g^{(k)}(\mathbf{x}_{t-1}^{(k)}),\mathbf{y}_{t-1}^{(k)})\Big\|^2\Big]$$

$$\leq 4\gamma_x^2\eta^2(C_g^4 L_f^2 + C_f^2 L_g^2)\mathbb{E}\Big[\big\|\bar{\mathbf{u}}_{t-1}\big\|^2\Big] + 4\gamma_y^2\eta^2 C_g^2 L_f^2\mathbb{E}\Big[\big\|\bar{\mathbf{v}}_{t-1}\big\|^2\Big] + 24\beta_x^2\gamma_x^2 p^2\eta^4(C_g^4 L_f^2 + C_f^2 L_g^2)C_g^2 C_f^2$$

$$+ 13824\beta_y^2\gamma_y^2\alpha^2 p^4\eta^6 C_g^4 L_f^4\sigma_g^2 + 165888\beta_y^2\gamma_y^2\alpha^2\gamma_x^2\beta_x^2 p^8\eta^{10} C_g^8 C_f^2 L_f^4 + 384\beta_y^2\gamma_y^2 p^2\eta^4 C_g^2 L_f^2\sigma_f^2 \,.$$

$$(53)$$

*Proof.*

$$\frac{1}{K}\sum_{k=1}^{K}\mathbb{E}\Big[\Big\|\nabla_x f^{(k)}(g^{(k)}(\mathbf{x}_t^{(k)}),\mathbf{y}_t^{(k)}) - \nabla_x f^{(k)}(g^{(k)}(\mathbf{x}_{t-1}^{(k)}),\mathbf{y}_{t-1}^{(k)})\Big\|^2\Big]$$

$$= \frac{1}{K}\sum_{k=1}^{K}\mathbb{E}\Big[\Big\|\nabla g^{(k)}(\mathbf{x}_t^{(k)})^T\nabla_g f^{(k)}(g^{(k)}(\mathbf{x}_t^{(k)}),\mathbf{y}_t^{(k)}) - \nabla g^{(k)}(\mathbf{x}_t^{(k)})^T\nabla_g f^{(k)}(g^{(k)}(\mathbf{x}_{t-1}^{(k)}),\mathbf{y}_{t-1}^{(k)})$$

$$+ \nabla g^{(k)}(\mathbf{x}_t^{(k)})^T\nabla_g f^{(k)}(g^{(k)}(\mathbf{x}_{t-1}^{(k)}),\mathbf{y}_{t-1}^{(k)}) - \nabla g^{(k)}(\mathbf{x}_{t-1}^{(k)})^T\nabla_g f^{(k)}(g^{(k)}(\mathbf{x}_{t-1}^{(k)}),\mathbf{y}_{t-1}^{(k)})\Big\|^2\Big]$$

$$\leq 2\frac{1}{K}\sum_{k=1}^{K}\mathbb{E}\Big[\Big\|\nabla g^{(k)}(\mathbf{x}_t^{(k)})^T\nabla_g f^{(k)}(g^{(k)}(\mathbf{x}_t^{(k)}),\mathbf{y}_t^{(k)}) - \nabla g^{(k)}(\mathbf{x}_t^{(k)})^T\nabla_g f^{(k)}(g^{(k)}(\mathbf{x}_{t-1}^{(k)}),\mathbf{y}_{t-1}^{(k)})\Big\|^2\Big]$$

$$+ 2\frac{1}{K}\sum_{k=1}^{K}\mathbb{E}\Big[\Big\|\nabla g^{(k)}(\mathbf{x}_t^{(k)})^T\nabla_g f^{(k)}(g^{(k)}(\mathbf{x}_{t-1}^{(k)}),\mathbf{y}_{t-1}^{(k)}) - \nabla g^{(k)}(\mathbf{x}_{t-1}^{(k)})^T\nabla_g f^{(k)}(g^{(k)}(\mathbf{x}_{t-1}^{(k)}),\mathbf{y}_{t-1}^{(k)})\Big\|^2\Big]$$

$$\leq 2C_g^2 L_f^2\Big(C_g^2\frac{1}{K}\sum_{k=1}^{K}\mathbb{E}\Big[\big\|\mathbf{x}_t^{(k)} - \mathbf{x}_{t-1}^{(k)}\big\|^2\Big] + \frac{1}{K}\sum_{k=1}^{K}\mathbb{E}\Big[\big\|\mathbf{y}_t^{(k)} - \mathbf{y}_{t-1}^{(k)}\big\|^2\Big]\Big) + 2C_f^2 L_g^2\frac{1}{K}\sum_{k=1}^{K}\mathbb{E}\Big[\big\|\mathbf{x}_t^{(k)} - \mathbf{x}_{t-1}^{(k)}\big\|^2\Big]$$

$$= 2(C_g^4 L_f^2 + C_f^2 L_g^2)\frac{1}{K}\sum_{k=1}^{K}\mathbb{E}\Big[\big\|\mathbf{x}_t^{(k)} - \mathbf{x}_{t-1}^{(k)}\big\|^2\Big] + 2C_g^2 L_f^2\frac{1}{K}\sum_{k=1}^{K}\mathbb{E}\Big[\big\|\mathbf{y}_t^{(k)} - \mathbf{y}_{t-1}^{(k)}\big\|^2\Big]$$

$$\leq 4\gamma_x^2\eta^2(C_g^4 L_f^2 + C_f^2 L_g^2)\frac{1}{K}\sum_{k=1}^{K}\mathbb{E}\Big[\big\|\mathbf{u}_{t-1}^{(k)} - \bar{\mathbf{u}}_{t-1}\big\|^2\Big] + 4\gamma_x^2\eta^2(C_g^4 L_f^2 + C_f^2 L_g^2)\mathbb{E}\Big[\big\|\bar{\mathbf{u}}_{t-1}\big\|^2\Big]$$

$$+ 4\gamma_y^2\eta^2 C_g^2 L_f^2\frac{1}{K}\sum_{k=1}^{K}\mathbb{E}\Big[\big\|\mathbf{v}_{t-1}^{(k)} - \bar{\mathbf{v}}_{t-1}\big\|^2\Big] + 4\gamma_y^2\eta^2 C_g^2 L_f^2\mathbb{E}\Big[\big\|\bar{\mathbf{v}}_{t-1}\big\|^2\Big]$$

$$\leq 4\gamma_x^2\eta^2(C_g^4 L_f^2 + C_f^2 L_g^2)\mathbb{E}\Big[\big\|\bar{\mathbf{u}}_{t-1}\big\|^2\Big] + 4\gamma_y^2\eta^2 C_g^2 L_f^2\mathbb{E}\Big[\big\|\bar{\mathbf{v}}_{t-1}\big\|^2\Big] + 24\beta_x^2\gamma_x^2 p^2\eta^4(C_g^4 L_f^2 + C_f^2 L_g^2)C_g^2 C_f^2$$

$$+ 13824\beta_y^2\gamma_y^2\alpha^2 p^4\eta^6 C_g^4 L_f^4\sigma_g^2 + 165888\beta_y^2\gamma_y^2\alpha^2\gamma_x^2\beta_x^2 p^8\eta^{10} C_g^8 C_f^2 L_f^4 + 384\beta_y^2\gamma_y^2 p^2\eta^4 C_g^2 L_f^2\sigma_f^2 \,,$$

$$(54)$$

where the third step holds due to Assumptions 1, 2, the last step holds due to Lemma 4 and Lemma 5. □

**Lemma 11.** *Given Assumptions 1-4 and $\alpha\eta\in(0,1)$, we can get*

$$\frac{1}{T}\sum_{t=0}^{T-1}\mathbb{E}\Big[\Big\|\bar{\mathbf{h}}_t - \frac{1}{K}\sum_{k=1}^{K}g^{(k)}(\mathbf{x}_t^{(k)})\Big\|^2\Big] \leq \frac{2\gamma_x^2 C_g^2}{\alpha^2}\frac{1}{T}\sum_{t=0}^{T-1}\mathbb{E}[\|\bar{\mathbf{u}}_t\|^2] + \frac{\sigma_g^2}{\alpha\eta T K} + \frac{12\gamma_x^2\beta_x^2 p^2\eta^2 C_g^4 C_f^2}{\alpha^2} + \frac{\alpha\eta\sigma_g^2}{K} \,.$$

$$(55)$$

*Proof.*

$$\mathbb{E}\Big[\Big\|\bar{\mathbf{h}}_{t+1} - \frac{1}{K}\sum_{k=1}^{K}g^{(k)}(\mathbf{x}_{t+1}^{(k)})\Big\|^2\Big] = \mathbb{E}\Big[\Big\|(1-\alpha\eta)\frac{1}{K}\sum_{k=1}^{K}\mathbf{h}_t^{(k)} + \alpha\eta\frac{1}{K}\sum_{k=1}^{K}g^{(k)}(\mathbf{x}_{t+1}^{(k)};\xi_{t+1}^{(k)}) - \frac{1}{K}\sum_{k=1}^{K}g^{(k)}(\mathbf{x}_{t+1}^{(k)})\Big\|^2\Big]$$

$$= \mathbb{E}\Big[\Big\|(1-\alpha\eta)\frac{1}{K}\sum_{k=1}^{K}\Big(\mathbf{h}_t^{(k)} - g^{(k)}(\mathbf{x}_t^{(k)})\Big) + (1-\alpha\eta)\frac{1}{K}\sum_{k=1}^{K}\Big(g^{(k)}(\mathbf{x}_{t+1}^{(k)}) - g^{(k)}(\mathbf{x}_t^{(k)})\Big)$$

$$+ \alpha\eta\frac{1}{K}\sum_{k=1}^{K}\Big(g^{(k)}(\mathbf{x}_{t+1}^{(k)};\xi_{t+1}^{(k)}) - g^{(k)}(\mathbf{x}_{t+1}^{(k)})\Big)\Big\|^2\Big]$$

$$= \mathbb{E}\Big[\Big\|(1-\alpha\eta)\frac{1}{K}\sum_{k=1}^{K}\Big(\mathbf{h}_t^{(k)} - g^{(k)}(\mathbf{x}_t^{(k)})\Big) + (1-\alpha\eta)\frac{1}{K}\sum_{k=1}^{K}\Big(g^{(k)}(\mathbf{x}_{t+1}^{(k)}) - g^{(k)}(\mathbf{x}_t^{(k)})\Big)\Big\|^2\Big]$$

$$+ \mathbb{E}\Big[\Big\|\alpha\eta\frac{1}{K}\sum_{k=1}^{K}\Big(g^{(k)}(\mathbf{x}_{t+1}^{(k)};\xi_{t+1}^{(k)}) - g^{(k)}(\mathbf{x}_{t+1}^{(k)})\Big)\Big\|^2\Big]$$

$$\le (1-\alpha\eta)^2(1+\frac{1}{a})\mathbb{E}\Big[\Big\|\frac{1}{K}\sum_{k=1}^{K}\Big(\mathbf{h}_t^{(k)} - g^{(k)}(\mathbf{x}_t^{(k)})\Big)\Big\|^2\Big]$$

$$+ (1-\alpha\eta)^2(1+a)\mathbb{E}\Big[\Big\|\frac{1}{K}\sum_{k=1}^{K}\Big(g^{(k)}(\mathbf{x}_{t+1}^{(k)}) - g^{(k)}(\mathbf{x}_t^{(k)})\Big)\Big\|^2\Big] + \frac{\alpha^2\eta^2\sigma_g^2}{K}$$

$$\le (1-\alpha\eta)\mathbb{E}\Big[\Big\|\frac{1}{K}\sum_{k=1}^{K}\Big(\mathbf{h}_t^{(k)} - g^{(k)}(\mathbf{x}_t^{(k)})\Big)\Big\|^2\Big] + \frac{C_g^2}{\alpha\eta}\frac{1}{K}\sum_{k=1}^{K}\mathbb{E}\Big[\Big\|\mathbf{x}_{t+1}^{(k)} - \mathbf{x}_t^{(k)}\Big\|^2\Big] + \frac{\alpha^2\eta^2\sigma_g^2}{K}$$

$$\le (1-\alpha\eta)\mathbb{E}\Big[\Big\|\bar{\mathbf{h}}_t - \frac{1}{K}\sum_{k=1}^{K}g^{(k)}(\mathbf{x}_t^{(k)})\Big\|^2\Big] + \frac{2\eta\gamma_x^2 C_g^2}{\alpha}\frac{1}{K}\sum_{k=1}^{K}\mathbb{E}\Big[\Big\|\mathbf{u}_t^{(k)} - \bar{\mathbf{u}}_t\Big\|^2\Big] + \frac{2\eta\gamma_x^2 C_g^2}{\alpha}\mathbb{E}[\|\bar{\mathbf{u}}_t\|^2] + \frac{\alpha^2\eta^2\sigma_g^2}{K}$$

$$\le (1-\alpha\eta)\mathbb{E}\Big[\Big\|\bar{\mathbf{h}}_t - \frac{1}{K}\sum_{k=1}^{K}g^{(k)}(\mathbf{x}_t^{(k)})\Big\|^2\Big] + \frac{2\eta\gamma_x^2 C_g^2}{\alpha}\mathbb{E}[\|\bar{\mathbf{u}}_t\|^2] + \frac{12\gamma_x^2\beta_x^2 p^2\eta^3 C_g^4 C_f^2}{\alpha} + \frac{\alpha^2\eta^2\sigma_g^2}{K},$$

$$(56)$$

where the fifth step holds due to $a = \frac{1-\alpha\eta}{\alpha\eta}$ and $\alpha\eta < 1$, the second to last step holds due to Assumption 3, the second to last step holds due to Lemma 4.

It can be reformulated as follows:

$$\alpha\eta\mathbb{E}\Big[\Big\|\bar{\mathbf{h}}_t - \frac{1}{K}\sum_{k=1}^{K}g^{(k)}(\mathbf{x}_t^{(k)})\Big\|^2\Big] \le \mathbb{E}\Big[\Big\|\bar{\mathbf{h}}_t - \frac{1}{K}\sum_{k=1}^{K}g^{(k)}(\mathbf{x}_t^{(k)})\Big\|^2\Big]$$

$$- \mathbb{E}\Big[\Big\|\bar{\mathbf{h}}_{t+1} - \frac{1}{K}\sum_{k=1}^{K}g^{(k)}(\mathbf{x}_{t+1}^{(k)})\Big\|^2\Big] + \frac{2\eta\gamma_x^2 C_g^2}{\alpha}\mathbb{E}[\|\bar{\mathbf{u}}_t\|^2] + \frac{12\gamma_x^2\beta_x^2 p^2\eta^3 C_g^4 C_f^2}{\alpha} + \frac{\alpha^2\eta^2\sigma_g^2}{K}.$$

$$(57)$$

By summing over $t$ from 0 to $T-1$, we can get

$$\frac{1}{T}\sum_{t=0}^{T-1}\mathbb{E}\Big[\Big\|\bar{\mathbf{h}}_t - \frac{1}{K}\sum_{k=1}^{K}g^{(k)}(\mathbf{x}_t^{(k)})\Big\|^2\Big]$$

$$\le \frac{1}{\alpha\eta T}\mathbb{E}\Big[\Big\|\bar{\mathbf{h}}_0 - \frac{1}{K}\sum_{k=1}^{K}g^{(k)}(\mathbf{x}_0^{(k)})\Big\|^2\Big] + \frac{2\gamma_x^2 C_g^2}{\alpha^2}\frac{1}{T}\sum_{t=0}^{T-1}\mathbb{E}[\|\bar{\mathbf{u}}_t\|^2] + \frac{12\gamma_x^2\beta_x^2 p^2\eta^2 C_g^4 C_f^2}{\alpha^2} + \frac{\alpha\eta\sigma_g^2}{K} \quad (58)$$

$$\le \frac{2\gamma_x^2 C_g^2}{\alpha^2}\frac{1}{T}\sum_{t=0}^{T-1}\mathbb{E}[\|\bar{\mathbf{u}}_t\|^2] + \frac{\sigma_g^2}{\alpha\eta T K} + \frac{12\gamma_x^2\beta_x^2 p^2\eta^2 C_g^4 C_f^2}{\alpha^2} + \frac{\alpha\eta\sigma_g^2}{K},$$

where the last step holds due to the following inequality:

$$\mathbb{E}\Big[\Big\|\bar{\mathbf{h}}_0 - \frac{1}{K}\sum_{k=1}^{K}g^{(k)}(\mathbf{x}_0^{(k)})\Big\|^2\Big] = \mathbb{E}\Big[\Big\|\frac{1}{K}\sum_{k=1}^{K}g^{(k)}(\mathbf{x}_0^{(k)};\xi_0^{(k)}) - \frac{1}{K}\sum_{k=1}^{K}g^{(k)}(\mathbf{x}_0^{(k)})\Big\|^2\Big] \le \frac{\sigma_g^2}{K}. \quad (59)$$

$\square$

**Lemma 12.** *Given Assumptions 1-4, we can get*

$$\mathbb{E}\Big[\Big\|\frac{1}{K}\sum_{k=1}^{K}\Big(\nabla g^{(k)}(\mathbf{x}_t^{(k)})^T\nabla_g f^{(k)}(\mathbf{h}_t^{(k)},\mathbf{y}_t^{(k)})-\nabla g^{(k)}(\mathbf{x}_t^{(k)};\xi_t^{(k)})^T\nabla_g f^{(k)}(\mathbf{h}_t^{(k)},\mathbf{y}_t^{(k)})\Big)\Big\|^2\Big]\leq\frac{C_f^2\sigma_{g'}^2}{K},$$

$$\mathbb{E}\Big[\Big\|\frac{1}{K}\sum_{k=1}^{K}\Big(\nabla g^{(k)}(\mathbf{x}_t^{(k)};\xi_t^{(k)})^T\nabla_g f^{(k)}(\mathbf{h}_t^{(k)},\mathbf{y}_t^{(k)})-\nabla g^{(k)}(\mathbf{x}_t^{(k)};\xi_t^{(k)})^T\nabla_g f^{(k)}(\mathbf{h}_t^{(k)},\mathbf{y}_t^{(k)};\zeta_t^{(k)})\Big)\Big\|^2\Big]\leq\frac{C_g^2\sigma_f^2}{K}.$$

(60)

*Proof.*

$$\mathbb{E}\Big[\Big\|\frac{1}{K}\sum_{k=1}^{K}\Big(\nabla g^{(k)}(\mathbf{x}_t^{(k)})^T\nabla_g f^{(k)}(\mathbf{h}_t^{(k)},\mathbf{y}_t^{(k)})-\nabla g^{(k)}(\mathbf{x}_t^{(k)};\xi_t^{(k)})^T\nabla_g f^{(k)}(\mathbf{h}_t^{(k)},\mathbf{y}_t^{(k)})\Big)\Big\|^2\Big]$$

$$=\mathbb{E}_{\mathcal{F}}\mathbb{E}_{\xi|\mathcal{F}}\mathbb{E}_{\zeta|\mathcal{F}}\Big[\Big\|\frac{1}{K}\sum_{k=1}^{K}\Big(\nabla g^{(k)}(\mathbf{x}_t^{(k)})^T\nabla_g f^{(k)}(\mathbf{h}_t^{(k)},\mathbf{y}_t^{(k)})-\nabla g^{(k)}(\mathbf{x}_t^{(k)};\xi_t^{(k)})^T\nabla_g f^{(k)}(\mathbf{h}_t^{(k)},\mathbf{y}_t^{(k)})\Big)\Big\|^2\Big]$$

$$=\mathbb{E}_{\mathcal{F}}\mathbb{E}_{\xi|\mathcal{F}}\Big[\Big\|\frac{1}{K}\sum_{k=1}^{K}\Big(\nabla g^{(k)}(\mathbf{x}_t^{(k)})-\nabla g^{(k)}(\mathbf{x}_t^{(k)};\xi_t^{(k)})\Big)^T\nabla_g f^{(k)}(\mathbf{h}_t^{(k)},\mathbf{y}_t^{(k)})\Big\|^2\Big]$$

$$=\frac{1}{K^2}\sum_{k=1}^{K}\mathbb{E}_{\mathcal{F}}\mathbb{E}_{\xi|\mathcal{F}}\Big[\Big\|\Big(\nabla g^{(k)}(\mathbf{x}_t^{(k)})-\nabla g^{(k)}(\mathbf{x}_t^{(k)};\xi_t^{(k)})\Big)^T\nabla_g f^{(k)}(\mathbf{h}_t^{(k)},\mathbf{y}_t^{(k)})\Big\|^2\Big]$$

$$\leq\frac{\sigma_{g'}^2 C_f^2}{K},$$

(61)

where $\mathcal{F}$ denotes all random factors except the sampling operation in the $t$-th iteration, $\xi$ and $\zeta$ are independent given $\mathcal{F}$, the third step holds since $(\nabla g^{(k)}(\mathbf{x}_t^{(k)})-\nabla g^{(k)}(\mathbf{x}_t^{(k)};\xi_t^{(k)}))^T\nabla_g f^{(k)}(\mathbf{h}_t^{(k)},\mathbf{y}_t^{(k)})$ are independent random vectors regarding $\xi$ across workers and the mean is zero, the last step holds due to Assumptions 1, 2.

Similarly, we can get

$$\mathbb{E}\Big[\Big\|\frac{1}{K}\sum_{k=1}^{K}\Big(\nabla g^{(k)}(\mathbf{x}_t^{(k)};\xi_t^{(k)})^T\nabla_g f^{(k)}(\mathbf{h}_t^{(k)},\mathbf{y}_t^{(k)})-\nabla g^{(k)}(\mathbf{x}_t^{(k)};\xi_t^{(k)})^T\nabla_g f^{(k)}(\mathbf{h}_t^{(k)},\mathbf{y}_t^{(k)};\zeta_t^{(k)})\Big)\Big\|^2\Big]$$

$$=\mathbb{E}_{\mathcal{F}}\mathbb{E}_{\xi|\mathcal{F}}\mathbb{E}_{\zeta|\mathcal{F}}\Big[\Big\|\frac{1}{K}\sum_{k=1}^{K}\Big(\nabla g^{(k)}(\mathbf{x}_t^{(k)};\xi_t^{(k)})^T\nabla_g f^{(k)}(\mathbf{h}_t^{(k)},\mathbf{y}_t^{(k)})-\nabla g^{(k)}(\mathbf{x}_t^{(k)};\xi_t^{(k)})^T\nabla_g f^{(k)}(\mathbf{h}_t^{(k)},\mathbf{y}_t^{(k)};\zeta_t^{(k)})\Big)\Big\|^2\Big]$$

$$=\mathbb{E}_{\mathcal{F}}\mathbb{E}_{\xi|\mathcal{F}}\mathbb{E}_{\zeta|\mathcal{F}}\Big[\Big\|\frac{1}{K}\sum_{k=1}^{K}\nabla g^{(k)}(\mathbf{x}_t^{(k)};\xi_t^{(k)})^T\Big(\nabla_g f^{(k)}(\mathbf{h}_t^{(k)},\mathbf{y}_t^{(k)})-\nabla_g f^{(k)}(\mathbf{h}_t^{(k)},\mathbf{y}_t^{(k)};\zeta_t^{(k)})\Big)\Big\|^2\Big]$$

$$=\mathbb{E}_{\mathcal{F}}\mathbb{E}_{\xi|\mathcal{F}}\frac{1}{K^2}\sum_{k=1}^{K}\mathbb{E}_{\zeta|\mathcal{F}}\Big[\Big\|\nabla g^{(k)}(\mathbf{x}_t^{(k)};\xi_t^{(k)})^T\Big(\nabla_g f^{(k)}(\mathbf{h}_t^{(k)},\mathbf{y}_t^{(k)})-\nabla_g f^{(k)}(\mathbf{h}_t^{(k)},\mathbf{y}_t^{(k)};\zeta_t^{(k)})\Big)\Big\|^2\Big]$$

$$=\frac{1}{K^2}\sum_{k=1}^{K}\mathbb{E}_{\mathcal{F}}\mathbb{E}_{\xi|\mathcal{F}}\mathbb{E}_{\zeta|\mathcal{F}}\Big[\Big\|\nabla g^{(k)}(\mathbf{x}_t^{(k)};\xi_t^{(k)})^T\Big(\nabla_g f^{(k)}(\mathbf{h}_t^{(k)},\mathbf{y}_t^{(k)})-\nabla_g f^{(k)}(\mathbf{h}_t^{(k)},\mathbf{y}_t^{(k)};\zeta_t^{(k)})\Big)\Big\|^2\Big]$$

$$\leq\frac{C_g^2\sigma_f^2}{K},$$

(62)

where $\mathcal{F}$ denotes all random factors except the sampling operation in the $t$-th iteration, $\xi$ and $\zeta$ are independent given $\mathcal{F}$, the third step holds since $\nabla g^{(k)}(\mathbf{x}_t^{(k)};\xi_t^{(k)})^T\Big(\nabla_g f^{(k)}(\mathbf{h}_t^{(k)},\mathbf{y}_t^{(k)})-\nabla_g f^{(k)}(\mathbf{h}_t^{(k)},\mathbf{y}_t^{(k)};\zeta_t^{(k)})\Big)$ are independent random vectors regarding $\zeta$ across workers and the mean is zero, the last step holds due to Assumptions 1, 2. $\qquad\square$

**Lemma 13.** *Given Assumption 1-4, $\beta_x\eta \in (0,1)$, and $\eta < 1$, we can get*

$$\frac{1}{T}\sum_{t=0}^{T-1}\mathbb{E}\Big[\Big\|\frac{1}{K}\sum_{k=1}^{K}\nabla_x f^{(k)}(g^{(k)}(\mathbf{x}_t^{(k)}),\mathbf{y}_t^{(k)}) - \frac{1}{K}\sum_{k=1}^{K}\mathbf{u}_t^{(k)}\Big\|^2\Big]$$

$$\leq \frac{3C_g^2 L_f^2\sigma_g^2 + 3C_f^2\sigma_{g'}^2 + 3C_g^2\sigma_f^2}{\beta_x\eta T} + 6C_g^2\frac{1}{T}\sum_{t=0}^{T-1}\mathbb{E}\Big[\Big\|\bar{\mathbf{h}}_t - \frac{1}{K}\sum_{k=1}^{K}g^{(k)}(\mathbf{x}_t^{(k)})\Big\|^2\Big]$$

$$+ \frac{8\gamma_x^2(C_g^4 L_f^2 + C_f^2 L_g^2)}{\beta_x^2}\frac{1}{T}\sum_{t=0}^{T-1}\mathbb{E}[\|\bar{\mathbf{u}}_t\|^2] + \frac{8\gamma_y^2 C_g^2 L_f^2}{\beta_x^2}\frac{1}{T}\sum_{t=0}^{T-1}\mathbb{E}[\|\bar{\mathbf{v}}_t\|^2] + \frac{12\gamma_x^2 C_g^4}{\alpha}\frac{1}{T}\sum_{t=0}^{T-1}\mathbb{E}[\|\bar{\mathbf{u}}_t\|^2]$$

$$+ 48\gamma_x^2 p^2\eta^2(C_g^4 L_f^2 + C_f^2 L_g^2)C_g^2 C_f^2 + 144\gamma_x^2\beta_{xf}^2 p^4\eta^4 C_g^6 C_f^4 + 216\alpha^2 p^2\eta^2 C_g^4\sigma_g^2 + 2592\alpha^2\gamma_x^2\beta_x^2 p^6\eta^6 C_g^8 C_f^2$$

$$+ \frac{27648\beta_y^2\gamma_y^2\alpha^2 p^4\eta^4 C_g^4 L_f^4\sigma_g^2}{\beta_x^2} + \frac{331776\beta_y^2\gamma_y^2\alpha^2\gamma_x^2\beta_x^2 p^8\eta^8 C_g^8 C_f^2 L_f^4}{\beta_x^2} + \frac{786\beta_y^2\gamma_y^2 p^2\eta^2 C_g^2 L_f^2\sigma_f^2}{\beta_x^2}$$

$$+ \frac{2\beta_x\eta C_f^2\sigma_{g'}^2}{K} + \frac{2\beta_x\eta C_g^2\sigma_f^2}{K} + \frac{72\gamma_x^2\beta_x^2 p^2\eta^2 C_g^6 C_f^2}{\alpha} + \frac{6\alpha^2\eta C_g^2\sigma_g^2}{K} .$$

$$(63)$$

*Proof.*

$$\mathbb{E}\Big[\Big\|\frac{1}{K}\sum_{k=1}^{K}\nabla_x f^{(k)}(g^{(k)}(\mathbf{x}_t^{(k)}),\mathbf{y}_t^{(k)}) - \frac{1}{K}\sum_{k=1}^{K}\mathbf{u}_t^{(k)}\Big\|^2\Big]$$

$$= \mathbb{E}\Big[\Big\|(1-\beta_x\eta)\frac{1}{K}\sum_{k=1}^{K}(\nabla_x f^{(k)}(g^{(k)}(\mathbf{x}_{t-1}^{(k)}),\mathbf{y}_{t-1}^{(k)}) - \mathbf{u}_{t-1}^{(k)})$$

$$+ (1-\beta_x\eta)\frac{1}{K}\sum_{k=1}^{K}(\nabla_x f^{(k)}(g^{(k)}(\mathbf{x}_t^{(k)}),\mathbf{y}_t^{(k)}) - \nabla_x f^{(k)}(g^{(k)}(\mathbf{x}_{t-1}^{(k)}),\mathbf{y}_{t-1}^{(k)}))$$

$$+ \beta_x\eta\frac{1}{K}\sum_{k=1}^{K}\Big(\nabla g^{(k)}(\mathbf{x}_t^{(k)})^T\nabla_g f^{(k)}(g^{(k)}(\mathbf{x}_t^{(k)}),\mathbf{y}_t^{(k)}) - \nabla g^{(k)}(\mathbf{x}_t^{(k)})^T\nabla_g f^{(k)}(\mathbf{h}_t^{(k)},\mathbf{y}_t^{(k)})$$

$$+ \nabla g^{(k)}(\mathbf{x}_t^{(k)})^T\nabla_g f^{(k)}(\mathbf{h}_t^{(k)},\mathbf{y}_t^{(k)}) - \nabla g^{(k)}(\mathbf{x}_t^{(k)};\xi_t^{(k)})^T\nabla_g f^{(k)}(\mathbf{h}_t^{(k)},\mathbf{y}_t^{(k)})$$

$$+ \nabla g^{(k)}(\mathbf{x}_t^{(k)};\xi_t^{(k)})^T\nabla_g f^{(k)}(\mathbf{h}_t^{(k)},\mathbf{y}_t^{(k)}) - \nabla g^{(k)}(\mathbf{x}_t^{(k)};\xi_t^{(k)})^T\nabla_g f^{(k)}(\mathbf{h}_t^{(k)},\mathbf{y}_t^{(k)};\zeta_t^{(k)})\Big)\Big\|^2\Big]$$

$$= \mathbb{E}\Big[\Big\|(1-\beta_x\eta)\frac{1}{K}\sum_{k=1}^{K}(\nabla_x f^{(k)}(g^{(k)}(\mathbf{x}_{t-1}^{(k)}),\mathbf{y}_{t-1}^{(k)}) - \mathbf{u}_{t-1}^{(k)})$$

$$+ (1-\beta_x\eta)\frac{1}{K}\sum_{k=1}^{K}(\nabla_x f^{(k)}(g^{(k)}(\mathbf{x}_t^{(k)}),\mathbf{y}_t^{(k)}) - \nabla_x f^{(k)}(g^{(k)}(\mathbf{x}_{t-1}^{(k)}),\mathbf{y}_{t-1}^{(k)}))$$

$$+ \beta_x\eta\frac{1}{K}\sum_{k=1}^{K}(\nabla g^{(k)}(\mathbf{x}_t^{(k)})^T\nabla_g f^{(k)}(g^{(k)}(\mathbf{x}_t^{(k)}),\mathbf{y}_t^{(k)}) - \nabla g^{(k)}(\mathbf{x}_t^{(k)})^T\nabla_g f^{(k)}(\mathbf{h}_t^{(k)},\mathbf{y}_t^{(k)}))\Big\|^2\Big]$$

$$+ \beta_x\eta^2\mathbb{E}\Big[\Big\|\frac{1}{K}\sum_{k=1}^{K}\Big(\nabla g^{(k)}(\mathbf{x}_t^{(k)})^T\nabla_g f^{(k)}(\mathbf{h}_t^{(k)},\mathbf{y}_t^{(k)}) - \nabla g^{(k)}(\mathbf{x}_t^{(k)};\xi_t^{(k)})^T\nabla_g f^{(k)}(\mathbf{h}_t^{(k)},\mathbf{y}_t^{(k)})$$

$$+ \nabla g^{(k)}(\mathbf{x}_t^{(k)};\xi_t^{(k)})^T\nabla_g f^{(k)}(\mathbf{h}_t^{(k)},\mathbf{y}_t^{(k)}) - \nabla g^{(k)}(\mathbf{x}_t^{(k)};\xi_t^{(k)})^T\nabla_g f^{(k)}(\mathbf{h}_t^{(k)},\mathbf{y}_t^{(k)};\zeta_t^{(k)})\Big)\Big\|^2\Big]$$

$$\triangleq T_1 + \beta_x^2\eta^2 T_2$$

$$(64)$$

where the second step holds due to $\mathbb{E}[\nabla g^{(k)}(\mathbf{x}_t^{(k)};\xi_t^{(k)})] = \nabla g^{(k)}(\mathbf{x}_t^{(k)})$ and $\mathbb{E}[f^{(k)}(\mathbf{h}_t^{(k)},\mathbf{y}_t^{(k)};\zeta_t^{(k)})] = f^{(k)}(\mathbf{h}_t^{(k)},\mathbf{y}_t^{(k)})$, $T_1$ denotes the first expectation and $T_2$ denotes

the second expectation. Then, $T_1$ can be bounded as follows:

$$T_1 \leq (1 - \beta_x \eta)^2 (1 + a) \mathbb{E}\left[\left\| \frac{1}{K} \sum_{k=1}^{K} \nabla_x f^{(k)}(g^{(k)}(\mathbf{x}_{t-1}^{(k)}), \mathbf{y}_{t-1}^{(k)}) - \frac{1}{K} \sum_{k=1}^{K} \mathbf{u}_{t-1}^{(k)} \right\|^2\right]$$

$$+ 2(1 - \beta_x \eta)^2 (1 + \frac{1}{a}) \mathbb{E}\left[\left\| \frac{1}{K} \sum_{k=1}^{K} \nabla_x f^{(k)}(g^{(k)}(\mathbf{x}_t^{(k)}), \mathbf{y}_t^{(k)}) - \frac{1}{K} \sum_{k=1}^{K} \nabla_x f^{(k)}(g^{(k)}(\mathbf{x}_{t-1}^{(k)}), \mathbf{y}_{t-1}^{(k)}) \right\|^2\right]$$

$$+ 2\beta_x^2 \eta^2 (1 + \frac{1}{a}) \mathbb{E}\left[\left\| \frac{1}{K} \sum_{k=1}^{K} \nabla g^{(k)}(\mathbf{x}_t^{(k)})^T \nabla_g f^{(k)}(g^{(k)}(\mathbf{x}_t^{(k)}), \mathbf{y}_t^{(k)}) \right.\right.$$

$$\left.\left. - \frac{1}{K} \sum_{k=1}^{K} \nabla g^{(k)}(\mathbf{x}_t^{(k)})^T \nabla_g f^{(k)}(\mathbf{h}_t^{(k)}, \mathbf{y}_t^{(k)}) \right\|^2\right]$$

$$\leq (1 - \beta_x \eta) \mathbb{E}\left[\left\| \frac{1}{K} \sum_{k=1}^{K} \nabla_x f^{(k)}(g^{(k)}(\mathbf{x}_{t-1}^{(k)}), \mathbf{y}_{t-1}^{(k)}) - \frac{1}{K} \sum_{k=1}^{K} \mathbf{u}_{t-1}^{(k)} \right\|^2\right]$$

$$+ \frac{2}{\beta_x \eta} \mathbb{E}\left[\left\| \frac{1}{K} \sum_{k=1}^{K} \nabla_x f^{(k)}(g^{(k)}(\mathbf{x}_t^{(k)}), \mathbf{y}_t^{(k)}) - \frac{1}{K} \sum_{k=1}^{K} \nabla_x f^{(k)}(g^{(k)}(\mathbf{x}_{t-1}^{(k)}), \mathbf{y}_{t-1}^{(k)}) \right\|^2\right]$$

$$+ 2\beta_x \eta \mathbb{E}\left[\left\| \frac{1}{K} \sum_{k=1}^{K} \nabla g^{(k)}(\mathbf{x}_t^{(k)})^T \nabla_g f^{(k)}(g^{(k)}(\mathbf{x}_t^{(k)}), \mathbf{y}_t^{(k)}) \right.\right.$$

$$\left.\left. - \frac{1}{K} \sum_{k=1}^{K} \nabla g^{(k)}(\mathbf{x}_t^{(k)})^T \nabla_g f^{(k)}(\mathbf{h}_t^{(k)}, \mathbf{y}_t^{(k)}) \right\|^2\right]$$

$$\leq (1 - \beta_x \eta) \mathbb{E}\left[\left\| \frac{1}{K} \sum_{k=1}^{K} \nabla_x f^{(k)}(g^{(k)}(\mathbf{x}_{t-1}^{(k)}), \mathbf{y}_{t-1}^{(k)}) - \frac{1}{K} \sum_{k=1}^{K} \mathbf{u}_{t-1}^{(k)} \right\|^2\right]$$

$$+ \frac{2}{\beta_x \eta} \frac{1}{K} \sum_{k=1}^{K} \mathbb{E}\left[\left\| \nabla_x f^{(k)}(g^{(k)}(\mathbf{x}_t^{(k)}), \mathbf{y}_t^{(k)}) - \nabla_x f^{(k)}(g^{(k)}(\mathbf{x}_{t-1}^{(k)}), \mathbf{y}_{t-1}^{(k)}) \right\|^2\right]$$

$$+ \underbrace{2\beta_x \eta \frac{1}{K} \sum_{k=1}^{K} \mathbb{E}\left[\left\| \nabla g^{(k)}(\mathbf{x}_t^{(k)})^T \nabla_g f^{(k)}(g^{(k)}(\mathbf{x}_t^{(k)}), \mathbf{y}_t^{(k)}) - \nabla g^{(k)}(\mathbf{x}_t^{(k)})^T \nabla_g f^{(k)}(\mathbf{h}_t^{(k)}, \mathbf{y}_t^{(k)}) \right\|^2\right]}_{T_3}$$

$$\leq (1 - \beta_x \eta) \mathbb{E}\left[\left\| \frac{1}{K} \sum_{k=1}^{K} \nabla_x f^{(k)}(g^{(k)}(\mathbf{x}_{t-1}^{(k)}), \mathbf{y}_{t-1}^{(k)}) - \frac{1}{K} \sum_{k=1}^{K} \mathbf{u}_{t-1}^{(k)} \right\|^2\right] + 48\beta_x \gamma_x^2 p^2 \eta^3 (C_g^4 L_f^2 + C_f^2 L_g^2) C_g^2 C_f^2$$

$$+ \frac{27648 \beta_y^2 \gamma_y^2 \alpha^2 p^4 \eta^5 C_g^4 L_f^4 \sigma_g^2}{\beta_x} + \frac{331776 \beta_y^2 \gamma_y^2 \alpha^2 \gamma_x^2 \beta_x^2 p^8 \eta^9 C_g^8 C_f^2 L_f^4}{\beta_x} + \frac{786 \beta_y^2 \gamma_y^2 p^2 \eta^3 C_g^2 L_f^2 \sigma_f^2}{\beta_x}$$

$$+ 144 \gamma_x^2 \beta_x^3 p^4 \eta^5 C_g^6 C_f^4 + 216 \beta_x \alpha^2 p^2 \eta^3 C_g^4 \sigma_g^2 + 2592 \alpha^2 \gamma_x^2 \beta_x^3 p^6 \eta^7 C_g^8 C_f^2$$

$$+ \frac{8 \gamma_x^2 \eta (C_g^4 L_f^2 + C_f^2 L_g^2)}{\beta_x} \mathbb{E}[\|\bar{\mathbf{u}}_{t-1}\|^2] + \frac{8 \gamma_y^2 \eta C_g^2 L_f^2}{\beta_x} \mathbb{E}[\|\bar{\mathbf{v}}_{t-1}\|^2] + \frac{12 \beta_x \eta \gamma_x^2 C_g^4}{\alpha} \mathbb{E}[\|\bar{\mathbf{u}}_{t-1}\|^2]$$

$$+ 6\beta_x \eta C_g^2 \mathbb{E}\left[\left\| \bar{\mathbf{h}}_{t-1} - \frac{1}{K} \sum_{k=1}^{K} g^{(k)}(\mathbf{x}_{t-1}^{(k)}) \right\|^2\right] + \frac{72 \gamma_x^2 \beta_x^3 p^2 \eta^3 C_g^6 C_f^2}{\alpha} + \frac{6 \alpha^2 \eta^2 \beta_x C_g^2 \sigma_g^2}{K},$$

$$(65)$$

where the second step holds due to $a = \frac{\beta_x \eta}{1 - \beta_x \eta}$ and $0 < \beta_x \eta < 1$, the last step holds due to Lemma 10 and the following inequality.

$$T_3 = \frac{1}{K} \sum_{k=1}^{K} \mathbb{E}\Big[\Big\|\nabla g^{(k)}(\mathbf{x}_t^{(k)})^T \nabla_g f^{(k)}(g^{(k)}(\mathbf{x}_t^{(k)}), \mathbf{y}_t^{(k)})$$

$$- \nabla g^{(k)}(\mathbf{x}_t^{(k)})^T \nabla_g f^{(k)}(\frac{1}{K} \sum_{k'=1}^{K} g^{(k')}(\mathbf{x}_t^{(k')}), \mathbf{y}_t^{(k)})$$

$$+ \nabla g^{(k)}(\mathbf{x}_t^{(k)})^T \nabla_g f^{(k)}(\frac{1}{K} \sum_{k'=1}^{K} g^{(k')}(\mathbf{x}_t^{(k')}), \mathbf{y}_t^{(k)})$$

$$- \nabla g^{(k)}(\mathbf{x}_t^{(k)})^T \nabla_g f^{(k)}(\frac{1}{K} \sum_{k'=1}^{K} \mathbf{h}_t^{(k')}, \mathbf{y}_t^{(k)})$$

$$+ \nabla g^{(k)}(\mathbf{x}_t^{(k)})^T \nabla_g f^{(k)}(\frac{1}{K} \sum_{k'=1}^{K} \mathbf{h}_t^{(k')}, \mathbf{y}_t^{(k)}) - \nabla g^{(k)}(\mathbf{x}_t^{(k)})^T \nabla_g f^{(k)}(\mathbf{h}_t^{(k)}, \mathbf{y}_t^{(k)})\Big\|^2\Big]$$

$$\leq 3\frac{1}{K} \sum_{k=1}^{K} \mathbb{E}\Big[\Big\|\nabla g^{(k)}(\mathbf{x}_t^{(k)})^T \nabla_g f^{(k)}(g^{(k)}(\mathbf{x}_t^{(k)}), \mathbf{y}_t^{(k)})$$

$$- \nabla g^{(k)}(\mathbf{x}_t^{(k)})^T \nabla_g f^{(k)}(\frac{1}{K} \sum_{k'=1}^{K} g^{(k')}(\mathbf{x}_t^{(k')}), \mathbf{y}_t^{(k)})\Big\|^2\Big]$$

$$+ 3\frac{1}{K} \sum_{k=1}^{K} \mathbb{E}\Big[\Big\|\nabla g^{(k)}(\mathbf{x}_t^{(k)})^T \nabla_g f^{(k)}(\frac{1}{K} \sum_{k'=1}^{K} g^{(k')}(\mathbf{x}_t^{(k')}), \mathbf{y}_t^{(k)})$$

$$- \nabla g^{(k)}(\mathbf{x}_t^{(k)})^T \nabla_g f^{(k)}(\frac{1}{K} \sum_{k'=1}^{K} \mathbf{h}_t^{(k')}, \mathbf{y}_t^{(k)})\Big\|^2\Big]$$

$$+ 3\frac{1}{K} \sum_{k=1}^{K} \mathbb{E}\Big[\Big\|\nabla g^{(k)}(\mathbf{x}_t^{(k)})^T \nabla_g f^{(k)}(\frac{1}{K} \sum_{k'=1}^{K} \mathbf{h}_t^{(k')}, \mathbf{y}_t^{(k)}) - \nabla g^{(k)}(\mathbf{x}_t^{(k)})^T \nabla_g f^{(k)}(\mathbf{h}_t^{(k)}, \mathbf{y}_t^{(k)})\Big\|^2\Big]$$

$$\leq 3C_g^2 \frac{1}{K} \sum_{k=1}^{K} \mathbb{E}\Big[\Big\|\nabla_g f^{(k)}(g^{(k)}(\mathbf{x}_t^{(k)}), \mathbf{y}_t^{(k)}) - \nabla_g f^{(k)}(\frac{1}{K} \sum_{k'=1}^{K} g^{(k')}(\mathbf{x}_t^{(k')}), \mathbf{y}_t^{(k)})\Big\|^2\Big]$$

$$+ 3C_g^2 \frac{1}{K} \sum_{k=1}^{K} \mathbb{E}\Big[\Big\|\nabla_g f^{(k)}(\frac{1}{K} \sum_{k'=1}^{K} g^{(k')}(\mathbf{x}_t^{(k')}), \mathbf{y}_t^{(k)}) - \nabla_g f^{(k)}(\frac{1}{K} \sum_{k'=1}^{K} \mathbf{h}_t^{(k')}, \mathbf{y}_t^{(k)})\Big\|^2\Big]$$

$$+ 3C_g^2 \frac{1}{K} \sum_{k=1}^{K} \mathbb{E}\Big[\Big\|\nabla_g f^{(k)}(\frac{1}{K} \sum_{k'=1}^{K} \mathbf{h}_t^{(k')}, \mathbf{y}_t^{(k)}) - \nabla_g f^{(k)}(\mathbf{h}_t^{(k)}, \mathbf{y}_t^{(k)})\Big\|^2\Big]$$

$$\leq 3C_g^2 L_f^2 \frac{1}{K} \sum_{k=1}^{K} \mathbb{E}\Big[\Big\|g^{(k)}(\mathbf{x}_t^{(k)}) - \frac{1}{K} \sum_{k'=1}^{K} g^{(k')}(\mathbf{x}_t^{(k')})\Big\|^2\Big] + 3C_g^2 \mathbb{E}\Big[\Big\|\frac{1}{K} \sum_{k=1}^{K} g^{(k)}(\mathbf{x}_t^{(k)}) - \frac{1}{K} \sum_{k=1}^{K} \mathbf{h}_t^{(k)}\Big\|^2\Big]$$

$$+ 3C_g^2 \frac{1}{K} \sum_{k=1}^{K} \mathbb{E}\Big[\Big\|\frac{1}{K} \sum_{k'=1}^{K} \mathbf{h}_t^{(k')} - \mathbf{h}_t^{(k)}\Big\|^2\Big]$$

$$\leq 72\gamma_x^2 \beta_x^2 p^4 \eta^4 C_g^6 C_f^4 + 108\alpha^2 p^2 \eta^2 C_g^4 \sigma_g^2 + 1296\alpha^2 \gamma_x^2 \beta_x^2 p^6 \eta^6 C_g^8 C_f^2$$

$$+ 3C_g^2 \mathbb{E}\Big[\Big\|\bar{\mathbf{h}}_{t-1} - \frac{1}{K} \sum_{k=1}^{K} g^{(k)}(\mathbf{x}_{t-1}^{(k)})\Big\|^2\Big] + \frac{6\eta\gamma_x^2 C_g^4}{\alpha} \mathbb{E}[\|\bar{\mathbf{u}}_{t-1}\|^2] + \frac{36\gamma_x^2 \beta_x^2 p^2 \eta^3 C_g^6 C_f^2}{\alpha} + \frac{3\alpha^2 \eta^2 C_g^2 \sigma_g^2}{K},$$

$$\leq 72\gamma_x^2 \beta_x^2 p^4 \eta^4 C_g^6 C_f^4 + 108\alpha^2 p^2 \eta^2 C_g^4 \sigma_g^2 + 1296\alpha^2 \gamma_x^2 \beta_x^2 p^6 \eta^6 C_g^8 C_f^2$$

$$+ 3C_g^2 \mathbb{E}\Big[\Big\|\bar{\mathbf{h}}_{t-1} - \frac{1}{K} \sum_{k=1}^{K} g^{(k)}(\mathbf{x}_{t-1}^{(k)})\Big\|^2\Big] + \frac{6\gamma_x^2 C_g^4}{\alpha} \mathbb{E}[\|\bar{\mathbf{u}}_{t-1}\|^2] + \frac{36\gamma_x^2 \beta_x^2 p^2 \eta^2 C_g^6 C_f^2}{\alpha} + \frac{3\alpha^2 \eta C_g^2 \sigma_g^2}{K},$$

(66)

where the second to last step holds due to Lemma 8, Lemma 9, and Eq. (56), the last step holds due to $\eta < 1$. As for $T_2$, we can get

$$
\begin{aligned}
T_2 &\leq 2\mathbb{E}\Big[\Big\|\frac{1}{K}\sum_{k=1}^{K}\Big(\nabla g^{(k)}(\mathbf{x}_t^{(k)})^T\nabla_g f^{(k)}(\mathbf{h}_t^{(k)},\mathbf{y}_t^{(k)}) - \nabla g^{(k)}(\mathbf{x}_t^{(k)};\xi_t^{(k)})^T\nabla_g f^{(k)}(\mathbf{h}_t^{(k)},\mathbf{y}_t^{(k)})\Big)\Big\|^2\Big] \\
&\quad + 2\mathbb{E}\Big[\Big\|\frac{1}{K}\sum_{k=1}^{K}\Big(\nabla g^{(k)}(\mathbf{x}_t^{(k)};\xi_t^{(k)})^T\nabla_g f^{(k)}(\mathbf{h}_t^{(k)},\mathbf{y}_t^{(k)}) - \nabla g^{(k)}(\mathbf{x}_t^{(k)};\xi_t^{(k)})^T\nabla_g f^{(k)}(\mathbf{h}_t^{(k)},\mathbf{y}_t^{(k)};\zeta_t^{(k)})\Big)\Big\|^2\Big] \\
&\leq \frac{2C_f^2\sigma_{g'}^2}{K} + \frac{2C_g^2\sigma_f^2}{K}\,,
\end{aligned}
\tag{67}
$$

where the last step holds due to Lemma 12. By combining $T_1$ and $T_2$, we can get

$$
\begin{aligned}
&\mathbb{E}\Big[\Big\|\frac{1}{K}\sum_{k=1}^{K}\nabla_x f^{(k)}(g^{(k)}(\mathbf{x}_t^{(k)}),\mathbf{y}_t^{(k)}) - \frac{1}{K}\sum_{k=1}^{K}\mathbf{u}_t^{(k)}\Big\|^2\Big] \\
&\leq (1-\beta_x\eta)\mathbb{E}\Big[\Big\|\frac{1}{K}\sum_{k=1}^{K}\nabla_x f^{(k)}(g^{(k)}(\mathbf{x}_{t-1}^{(k)}),\mathbf{y}_{t-1}^{(k)}) - \frac{1}{K}\sum_{k=1}^{K}\mathbf{u}_{t-1}^{(k)}\Big\|^2\Big] + 48\beta_x\gamma_x^2 p^2\eta^3(C_g^4 L_f^2 + C_f^2 L_g^2)C_g^2 C_f^2 \\
&\quad + \frac{27648\beta_y^2\gamma_y^2\alpha^2 p^4\eta^5 C_g^4 L_f^4\sigma_g^2}{\beta_x} + \frac{331776\beta_y^2\gamma_y^2\alpha^2\gamma_x^2\beta_x^2 p^8\eta^9 C_g^8 C_f^2 L_f^4}{\beta_x} + \frac{786\beta_y^2\gamma_y^2 p^2\eta^3 C_g^2 L_f^2\sigma_f^2}{\beta_x} \\
&\quad + 144\gamma_x^2\beta_x^3 p^4\eta^5 C_g^6 C_f^4 + 216\beta_x\alpha^2 p^2\eta^3 C_g^4\sigma_g^2 + 2592\alpha^2\gamma_x^2\beta_x^3 p^6\eta^7 C_g^8 C_f^2 \\
&\quad + \frac{8\gamma_x^2\eta(C_g^4 L_f^2 + C_f^2 L_g^2)}{\beta_x}\mathbb{E}[\|\bar{\mathbf{u}}_{t-1}\|^2] + \frac{8\gamma_y^2\eta C_g^2 L_f^2}{\beta_x}\mathbb{E}[\|\bar{\mathbf{v}}_{t-1}\|^2] + \frac{12\beta_x\eta\gamma_x^2 C_g^4}{\alpha}\mathbb{E}[\|\bar{\mathbf{u}}_{t-1}\|^2] \\
&\quad + 6\beta_x\eta C_g^2\mathbb{E}\Big[\Big\|\bar{\mathbf{h}}_{t-1} - \frac{1}{K}\sum_{k=1}^{K}g^{(k)}(\mathbf{x}_{t-1}^{(k)})\Big\|^2\Big] + \frac{72\gamma_x^2\beta_x^3 p^2\eta^3 C_g^6 C_f^2}{\alpha} + \frac{6\alpha^2\eta^2\beta_x C_g^2\sigma_g^2}{K} \\
&\quad + \frac{2\beta_x^2\eta^2 C_f^2\sigma_{g'}^2}{K} + \frac{2\beta_x^2\eta^2 C_g^2\sigma_f^2}{K}\,,
\end{aligned}
\tag{68}
$$

It can be reformulated as follows:

$$
\begin{aligned}
&\beta_x\eta\mathbb{E}\Big[\Big\|\frac{1}{K}\sum_{k=1}^{K}\nabla_x f^{(k)}(g^{(k)}(\mathbf{x}_t^{(k)}),\mathbf{y}_t^{(k)}) - \frac{1}{K}\sum_{k=1}^{K}\mathbf{u}_t^{(k)}\Big\|^2\Big] \\
&\leq \mathbb{E}\Big[\Big\|\frac{1}{K}\sum_{k=1}^{K}\nabla_x f^{(k)}(g^{(k)}(\mathbf{x}_t^{(k)}),\mathbf{y}_t^{(k)}) - \frac{1}{K}\sum_{k=1}^{K}\mathbf{u}_t^{(k)}\Big\|^2\Big] \\
&\quad - \mathbb{E}\Big[\Big\|\frac{1}{K}\sum_{k=1}^{K}\nabla_x f^{(k)}(g^{(k)}(\mathbf{x}_{t+1}^{(k)}),\mathbf{y}_{t+1}^{(k)}) - \frac{1}{K}\sum_{k=1}^{K}\mathbf{u}_{t+1}^{(k)}\Big\|^2\Big] + \frac{8\gamma_x^2\eta(C_g^4 L_f^2 + C_f^2 L_g^2)}{\beta_x}\mathbb{E}[\|\bar{\mathbf{u}}_t\|^2] \\
&\quad + \frac{8\gamma_y^2\eta C_g^2 L_f^2}{\beta_x}\mathbb{E}[\|\bar{\mathbf{v}}_t\|^2] + \frac{12\beta_x\eta\gamma_x^2 C_g^4}{\alpha}\mathbb{E}[\|\bar{\mathbf{u}}_t\|^2] + 6\beta_x\eta C_g^2\mathbb{E}\Big[\Big\|\bar{\mathbf{h}}_t - \frac{1}{K}\sum_{k=1}^{K}g^{(k)}(\mathbf{x}_t^{(k)})\Big\|^2\Big] \\
&\quad + 48\beta_x\gamma_x^2 p^2\eta^3(C_g^4 L_f^2 + C_f^2 L_g^2)C_g^2 C_f^2 + 144\gamma_x^2\beta_x^3 p^4\eta^5 C_g^6 C_f^4 + 216\beta_x\alpha^2 p^2\eta^3 C_g^4\sigma_g^2 + 2592\alpha^2\gamma_x^2\beta_x^3 p^6\eta^7 C_g^8 C_f^2 \\
&\quad + \frac{27648\beta_y^2\gamma_y^2\alpha^2 p^4\eta^5 C_g^4 L_f^4\sigma_g^2}{\beta_x} + \frac{331776\beta_y^2\gamma_y^2\alpha^2\gamma_x^2\beta_x^2 p^8\eta^9 C_g^8 C_f^2 L_f^4}{\beta_x} + \frac{786\beta_y^2\gamma_y^2 p^2\eta^3 C_g^2 L_f^2\sigma_f^2}{\beta_x} \\
&\quad + \frac{2\beta_x^2\eta^2 C_f^2\sigma_{g'}^2}{K} + \frac{2\beta_x^2\eta^2 C_g^2\sigma_f^2}{K} + \frac{72\gamma_x^2\beta_x^3 p^2\eta^3 C_g^6 C_f^2}{\alpha} + \frac{6\alpha^2\eta^2\beta_x C_g^2\sigma_g^2}{K}\,.
\end{aligned}
\tag{69}
$$

By summing over t from 0 to T - 1, we can get

$$
\frac{1}{T}\sum_{t=0}^{T-1}\mathbb{E}\Big[\Big\|\frac{1}{K}\sum_{k=1}^{K}\nabla_x f^{(k)}(g^{(k)}(\mathbf{x}_t^{(k)}),\mathbf{y}_t^{(k)}) - \frac{1}{K}\sum_{k=1}^{K}\mathbf{u}_t^{(k)}\Big\|^2\Big]
$$

$$
\leq \frac{1}{\beta_x\eta T}\mathbb{E}\Big[\Big\|\frac{1}{K}\sum_{k=1}^{K}\nabla_x f^{(k)}(g^{(k)}(\mathbf{x}_0^{(k)}),\mathbf{y}_0^{(k)}) - \frac{1}{K}\sum_{k=1}^{K}\mathbf{u}_0^{(k)}\Big\|^2\Big] + 6C_g^2\frac{1}{T}\sum_{t=0}^{T-1}\mathbb{E}\Big[\Big\|\bar{\mathbf{h}}_t - \frac{1}{K}\sum_{k=1}^{K}g^{(k)}(\mathbf{x}_t^{(k)})\Big\|^2\Big]
$$

$$
+ \frac{8\gamma_x^2(C_g^4 L_f^2 + C_f^2 L_g^2)}{\beta_x^2}\frac{1}{T}\sum_{t=0}^{T-1}\mathbb{E}[\|\bar{\mathbf{u}}_t\|^2] + \frac{8\gamma_y^2 C_g^2 L_f^2}{\beta_x^2}\frac{1}{T}\sum_{t=0}^{T-1}\mathbb{E}[\|\bar{\mathbf{v}}_t\|^2] + \frac{12\gamma_x^2 C_g^4}{\alpha}\frac{1}{T}\sum_{t=0}^{T-1}\mathbb{E}[\|\bar{\mathbf{u}}_t\|^2]
$$

$$
+ 48\gamma_x^2 p^2\eta^2(C_g^4 L_f^2 + C_f^2 L_g^2)C_g^2 C_f^2 + 144\gamma_x^2\beta_x^2 p^4\eta^4 C_g^6 C_f^4 + 216\alpha^2 p^2\eta^2 C_g^4\sigma_g^2 + 2592\alpha^2\gamma_x^2\beta_x^2 p^6\eta^6 C_g^8 C_f^2
$$

$$
+ \frac{27648\beta_y^2\gamma_y^2\alpha^2 p^4\eta^4 C_g^4 L_f^4\sigma_g^2}{\beta_x^2} + \frac{331776\beta_y^2\gamma_y^2\alpha^2\gamma_x^2\beta_x^2 p^8\eta^8 C_g^8 C_f^2 L_f^4}{\beta_x^2} + \frac{786\beta_y^2\gamma_y^2 p^2\eta^2 C_g^2 L_f^2\sigma_f^2}{\beta_x^2}
$$

$$
+ \frac{2\beta_x\eta C_f^2\sigma_{g'}^2}{K} + \frac{2\beta_x\eta C_g^2\sigma_f^2}{K} + \frac{72\gamma_x^2\beta_x^2 p^2\eta^2 C_g^6 C_f^2}{\alpha} + \frac{6\alpha^2\eta C_g^2\sigma_g^2}{K}
$$

$$
\leq \frac{3C_g^2 L_f^2\sigma_g^2 + 3C_f^2\sigma_{g'}^2 + 3C_g^2\sigma_f^2}{\beta_x\eta T} + 6C_g^2\frac{1}{T}\sum_{t=0}^{T-1}\mathbb{E}\Big[\Big\|\bar{\mathbf{h}}_t - \frac{1}{K}\sum_{k=1}^{K}g^{(k)}(\mathbf{x}_t^{(k)})\Big\|^2\Big]
$$

$$
+ \frac{8\gamma_x^2(C_g^4 L_f^2 + C_f^2 L_g^2)}{\beta_x^2}\frac{1}{T}\sum_{t=0}^{T-1}\mathbb{E}[\|\bar{\mathbf{u}}_t\|^2] + \frac{8\gamma_y^2 C_g^2 L_f^2}{\beta_x^2}\frac{1}{T}\sum_{t=0}^{T-1}\mathbb{E}[\|\bar{\mathbf{v}}_t\|^2] + \frac{12\gamma_x^2 C_g^4}{\alpha}\frac{1}{T}\sum_{t=0}^{T-1}\mathbb{E}[\|\bar{\mathbf{u}}_t\|^2]
$$

$$
+ 48\gamma_x^2 p^2\eta^2(C_g^4 L_f^2 + C_f^2 L_g^2)C_g^2 C_f^2 + 144\gamma_x^2\beta_x^2 p^4\eta^4 C_g^6 C_f^4 + 216\alpha^2 p^2\eta^2 C_g^4\sigma_g^2 + 2592\alpha^2\gamma_x^2\beta_x^2 p^6\eta^6 C_g^8 C_f^2
$$

$$
+ \frac{27648\beta_y^2\gamma_y^2\alpha^2 p^4\eta^4 C_g^4 L_f^4\sigma_g^2}{\beta_x^2} + \frac{331776\beta_y^2\gamma_y^2\alpha^2\gamma_x^2\beta_x^2 p^8\eta^8 C_g^8 C_f^2 L_f^4}{\beta_x^2} + \frac{786\beta_y^2\gamma_y^2 p^2\eta^2 C_g^2 L_f^2\sigma_f^2}{\beta_x^2}
$$

$$
+ \frac{2\beta_x\eta C_f^2\sigma_{g'}^2}{K} + \frac{2\beta_x\eta C_g^2\sigma_f^2}{K} + \frac{72\gamma_x^2\beta_x^2 p^2\eta^2 C_g^6 C_f^2}{\alpha} + \frac{6\alpha^2\eta C_g^2\sigma_g^2}{K},
$$

(70)

where the second step holds due to the following inequality:

$$
\mathbb{E}\Big[\Big\|\frac{1}{K}\sum_{k=1}^{K}\nabla_x f^{(k)}(g^{(k)}(\mathbf{x}_0^{(k)}),\mathbf{y}_0^{(k)}) - \frac{1}{K}\sum_{k=1}^{K}\mathbf{u}_0^{(k)}\Big\|^2\Big]
$$

$$
= \mathbb{E}\Big[\Big\|\frac{1}{K}\sum_{k=1}^{K}\nabla_x g^{(k)}(x^{(k)})^T\nabla_g f^{(k)}(g^{(k)}(\mathbf{x}_0^{(k)}),\mathbf{y}_0^{(k)}) - \frac{1}{K}\sum_{k=1}^{K}\nabla g^{(k)}(x_0^{(k)};\xi_0^{(k)})^T\nabla_g f^{(k)}(h_0^{(k)},y_0^{(k)};\zeta_0^{(k)})\Big\|^2\Big]
$$

$$
= \mathbb{E}\Big[\Big\|\frac{1}{K}\sum_{k=1}^{K}\nabla_x g^{(k)}(x_0^{(k)})^T\nabla_g f^{(k)}(g^{(k)}(\mathbf{x}_0^{(k)}),\mathbf{y}_0^{(k)}) - \frac{1}{K}\sum_{k=1}^{K}\nabla g^{(k)}(x_0^{(k)})^T\nabla_g f^{(k)}(h_0^{(k)}),y_0^{(k)})
$$

$$
+ \frac{1}{K}\sum_{k=1}^{K}\nabla g^{(k)}(x_0^{(k)})^T\nabla_g f^{(k)}(h_0^{(k)}),y_0^{(k)}) - \frac{1}{K}\sum_{k=1}^{K}\nabla g^{(k)}(x_0^{(k)};\xi_0^{(k)})^T\nabla_g f^{(k)}(h_0^{(k)}),y_0^{(k)})
$$

$$
+ \frac{1}{K}\sum_{k=1}^{K}\nabla g^{(k)}(x_0^{(k)};\xi_0^{(k)})^T\nabla_g f^{(k)}(h_0^{(k)}),y_0^{(k)}) - \frac{1}{K}\sum_{k=1}^{K}\nabla g^{(k)}(x_0^{(k)};\xi_0^{(k)})^T\nabla_g f^{(k)}(h_0^{(k)},y_0^{(k)};\zeta_0^{(k)})\Big\|^2\Big]
$$

$$
\leq 3C_g^2 L_f^2\frac{1}{K}\sum_{k=1}^{K}\mathbb{E}\Big[\Big\|g^{(k)}(x_0^{(k)}) - h_0^{(k)}\Big\|^2\Big] + 3C_f^2\sigma_{g'}^2 + 3C_g^2\sigma_f^2
$$

$$
\leq 3C_g^2 L_f^2\sigma_g^2 + 3C_f^2\sigma_{g'}^2 + 3C_g^2\sigma_f^2 .
$$

(71)

□

**Lemma 14.** *Given Assumption 1-4, $\beta_y \eta \in (0,1)$, and $\eta < 1$, we can get*

$$\frac{1}{T}\sum_{t=0}^{T-1}\mathbb{E}\Big[\Big\|\frac{1}{K}\sum_{k=1}^{K}\nabla_y f^{(k)}(g^{(k)}(\mathbf{x}_t^{(k)}),\mathbf{y}_t^{(k)}) - \frac{1}{K}\sum_{k=1}^{K}\mathbf{v}_t^{(k)}\Big\|^2\Big]$$

$$\leq \frac{2L_f^2\sigma_g^2 + 2\sigma_f^2}{\beta_y\eta T} + 6L_f^2\frac{1}{T}\sum_{t=0}^{T-1}\mathbb{E}\Big[\Big\|\bar{\mathbf{h}}_t - \frac{1}{K}\sum_{k=1}^{K}g^{(k)}(\mathbf{x}_t^{(k)})\Big\|^2\Big]$$

$$+ \frac{12L_f^2\gamma_x^2 C_g^2}{\alpha}\frac{1}{T}\sum_{t=0}^{T-1}\mathbb{E}[\|\bar{\mathbf{u}}_t\|^2] + \frac{4\gamma_x^2 L_f^2 C_g^2}{\beta_y^2}\frac{1}{T}\sum_{t=0}^{T-1}\mathbb{E}[\|\bar{\mathbf{u}}_t\|^2] + \frac{4\gamma_y^2 L_f^2}{\beta_y^2}\frac{1}{T}\sum_{t=0}^{T-1}\mathbb{E}[\|\bar{\mathbf{v}}_t\|^2] \qquad (72)$$

$$+ 144\gamma_x^2\beta_x^2 p^4\eta^4 C_g^4 C_f^2 L_f^2 + 216\alpha^2 p^2\eta^2 C_g^2 L_f^2\sigma_g^2 + 2592\alpha^2\gamma_x^2\beta_x^2 p^6\eta^6 C_g^6 C_f^2 L_f^2$$

$$+ 13824\gamma_y^2\alpha^2 p^4\eta^4 C_g^2 L_f^4\sigma_g^2 + 165888\gamma_y^2\alpha^2\gamma_x^2\beta_x^2 p^8\eta^8 C_g^6 C_f^2 L_f^4 + 384\gamma_y^2 p^2\eta^2 L_f^2\sigma_f^2$$

$$+ \frac{72\gamma_x^2\beta_x^2 p^2\eta^2 C_g^4 C_f^2 L_f^2}{\alpha} + \frac{6\alpha^2\eta\sigma_g^2 L_f^2}{K} + \frac{\beta_y\eta\sigma_f^2}{K} + \frac{24\gamma_x^2 p^2\beta_x^2\eta^2 L_f^2 C_g^4 C_f^2}{\beta_y^2}.$$

*Proof.*

$$\mathbb{E}\Big[\Big\|\frac{1}{K}\sum_{k=1}^{K}\nabla_y f^{(k)}(g^{(k)}(\mathbf{x}_t^{(k)}),\mathbf{y}_t^{(k)}) - \frac{1}{K}\sum_{k=1}^{K}\mathbf{v}_t^{(k)}\Big\|^2\Big]$$

$$= \mathbb{E}\Big[\Big\|(1-\beta_y\eta)\Big(\frac{1}{K}\sum_{k=1}^{K}\nabla_y f^{(k)}(g^{(k)}(\mathbf{x}_{t-1}^{(k)}),\mathbf{y}_{t-1}^{(k)}) - \frac{1}{K}\sum_{k=1}^{K}\mathbf{v}_{t-1}^{(k)}\Big)$$

$$+ (1-\beta_y\eta)\Big(\frac{1}{K}\sum_{k=1}^{K}\nabla_y f^{(k)}(g^{(k)}(\mathbf{x}_t^{(k)}),\mathbf{y}_t^{(k)}) - \frac{1}{K}\sum_{k=1}^{K}\nabla_y f^{(k)}(g^{(k)}(\mathbf{x}_{t-1}^{(k)}),\mathbf{y}_{t-1}^{(k)})\Big)$$

$$+ \beta_y\eta\Big(\frac{1}{K}\sum_{k=1}^{K}\nabla_y f^{(k)}(g^{(k)}(\mathbf{x}_t^{(k)}),\mathbf{y}_t^{(k)}) - \frac{1}{K}\sum_{k=1}^{K}\nabla_y f^{(k)}(\mathbf{h}_t^{(k)},\mathbf{y}_t^{(k)})$$

$$+ \frac{1}{K}\sum_{k=1}^{K}\nabla_y f^{(k)}(\mathbf{h}_t^{(k)},\mathbf{y}_t^{(k)}) - \frac{1}{K}\sum_{k=1}^{K}\nabla_y f^{(k)}(\mathbf{h}_t^{(k)},\mathbf{y}_t^{(k)};\zeta_t^{(k)})\Big)\Big\|^2\Big]$$

$$\leq (1-\beta_y\eta)^2(1+a)\mathbb{E}\Big[\Big\|\frac{1}{K}\sum_{k=1}^{K}\nabla_y f^{(k)}(g^{(k)}(\mathbf{x}_{t-1}^{(k)}),\mathbf{y}_{t-1}^{(k)}) - \frac{1}{K}\sum_{k=1}^{K}\mathbf{v}_{t-1}^{(k)}\Big\|^2\Big]$$

$$+ 2(1-\beta_y\eta)^2(1+\frac{1}{a})\mathbb{E}\Big[\Big\|\frac{1}{K}\sum_{k=1}^{K}\nabla_y f^{(k)}(g^{(k)}(\mathbf{x}_t^{(k)}),\mathbf{y}_t^{(k)}) - \frac{1}{K}\sum_{k=1}^{K}\nabla_y f^{(k)}(g^{(k)}(\mathbf{x}_{t-1}^{(k)}),\mathbf{y}_{t-1}^{(k)})\Big\|^2\Big]$$

$$+ 2\beta_y^2\eta^2(1+\frac{1}{a})\mathbb{E}\Big[\Big\|\frac{1}{K}\sum_{k=1}^{K}\nabla_y f^{(k)}(g^{(k)}(\mathbf{x}_t^{(k)}),\mathbf{y}_t^{(k)}) - \frac{1}{K}\sum_{k=1}^{K}\nabla_y f^{(k)}(\mathbf{h}_t^{(k)},\mathbf{y}_t^{(k)})\Big\|^2\Big]$$

$$+ \beta_y^2\eta^2\mathbb{E}\Big[\Big\|\frac{1}{K}\sum_{k=1}^{K}\nabla_y f^{(k)}(\mathbf{h}_t^{(k)},\mathbf{y}_t^{(k)}) - \frac{1}{K}\sum_{k=1}^{K}\nabla_y f^{(k)}(\mathbf{h}_t^{(k)},\mathbf{y}_t^{(k)};\zeta_t^{(k)})\Big\|^2\Big]$$

$$\leq (1-\beta_y\eta)\mathbb{E}\Big[\Big\|\frac{1}{K}\sum_{k=1}^{K}\nabla_y f^{(k)}(g^{(k)}(\mathbf{x}_{t-1}^{(k)}),\mathbf{y}_{t-1}^{(k)}) - \frac{1}{K}\sum_{k=1}^{K}\mathbf{v}_{t-1}^{(k)}\Big\|^2\Big]$$

$$+ \underbrace{\frac{2}{\beta_y\eta}\mathbb{E}\Big[\Big\|\frac{1}{K}\sum_{k=1}^{K}\nabla_y f^{(k)}(g^{(k)}(\mathbf{x}_t^{(k)}),\mathbf{y}_t^{(k)}) - \frac{1}{K}\sum_{k=1}^{K}\nabla_y f^{(k)}(g^{(k)}(\mathbf{x}_{t-1}^{(k)}),\mathbf{y}_{t-1}^{(k)})\Big\|^2\Big]}_{T_1}$$

$$+ \underbrace{2\beta_y\eta\,\mathbb{E}\Big[\Big\|\frac{1}{K}\sum_{k=1}^{K}\nabla_y f^{(k)}(g^{(k)}(\mathbf{x}_t^{(k)}),\mathbf{y}_t^{(k)}) - \frac{1}{K}\sum_{k=1}^{K}\nabla_y f^{(k)}(\mathbf{h}_t^{(k)},\mathbf{y}_t^{(k)})\Big\|^2\Big]}_{T_2} + \frac{\beta_y^2\eta^2\sigma_f^2}{K}.$$

$$(73)$$

Then, for $T_1$, we can get

$$
\begin{aligned}
T_1 &\leq L_f^2 \frac{1}{K} \sum_{k=1}^K \mathbb{E}\Big[\big\|g^{(k)}(\mathbf{x}_t^{(k)}) - g^{(k)}(\mathbf{x}_{t-1}^{(k)})\big\|^2\Big] + L_f^2 \frac{1}{K} \sum_{k=1}^K \mathbb{E}\Big[\big\|\mathbf{y}_t^{(k)} - \mathbf{y}_{t-1}^{(k)}\big\|^2\Big] \\
&\leq L_f^2 C_g^2 \frac{1}{K} \sum_{k=1}^K \mathbb{E}\Big[\big\|\mathbf{x}_t^{(k)} - \mathbf{x}_{t-1}^{(k)}\big\|^2\Big] + L_f^2 \frac{1}{K} \sum_{k=1}^K \mathbb{E}\Big[\big\|\mathbf{y}_t^{(k)} - \mathbf{y}_{t-1}^{(k)}\big\|^2\Big] \\
&\leq 2\gamma_x^2 \eta^2 L_f^2 C_g^2 \frac{1}{K} \sum_{k=1}^K \mathbb{E}\Big[\big\|\mathbf{u}_{t-1}^{(k)} - \bar{\mathbf{u}}_{t-1}\big\|^2\Big] + 2\gamma_x^2 \eta^2 L_f^2 C_g^2 \mathbb{E}[\|\bar{\mathbf{u}}_{t-1}\|^2] \\
&\quad + 2\gamma_y^2 \eta^2 L_f^2 \frac{1}{K} \sum_{k=1}^K \mathbb{E}\Big[\big\|\mathbf{v}_{t-1}^{(k)} - \bar{\mathbf{v}}_{t-1}\big\|^2\Big] + 2\gamma_y^2 \eta^2 L_f^2 \mathbb{E}[\|\bar{\mathbf{v}}_{t-1}\|^2] \, , \\
&\leq 12\gamma_x^2 p^2 \beta_x^2 \eta^4 L_f^2 C_g^4 C_f^2 + 2\gamma_x^2 \eta^2 L_f^2 C_g^2 \mathbb{E}[\|\bar{\mathbf{u}}_{t-1}\|^2] \\
&\quad + 6912 \beta_y^2 \gamma_y^2 \alpha^2 p^4 \eta^6 C_g^2 L_f^4 \sigma_g^2 + 82944 \beta_y^2 \gamma_y^2 \alpha^2 \gamma_x^2 \beta_x^2 p^8 \eta^{10} C_g^6 C_f^2 L_f^4 + 192 \beta_y^2 \gamma_y^2 p^2 \eta^4 L_f^2 \sigma_f^2 + 2\gamma_y^2 \eta^4 L_f^2 \mathbb{E}[\|\bar{\mathbf{v}}_{t-1}\|^2] \, ,
\end{aligned}
\tag{74}
$$

where these inequalities hold due to Assumptions 1-2.

As for $T_2$, we can get

$$
\begin{aligned}
T_2 &= \mathbb{E}\Big[\Big\|\frac{1}{K} \sum_{k=1}^K \nabla_y f^{(k)}(g^{(k)}(\mathbf{x}_t^{(k)}), \mathbf{y}_t^{(k)}) - \frac{1}{K} \sum_{k=1}^K \nabla_y f^{(k)}(\frac{1}{K} \sum_{k'=1}^K g^{(k')}(\mathbf{x}_t^{(k')}), \mathbf{y}_t^{(k)}) \\
&\quad + \frac{1}{K} \sum_{k=1}^K \nabla_y f^{(k)}(\frac{1}{K} \sum_{k'=1}^K g^{(k')}(\mathbf{x}_t^{(k')}), \mathbf{y}_t^{(k)}) - \frac{1}{K} \sum_{k=1}^K \nabla_y f^{(k)}(\frac{1}{K} \sum_{k'=1}^K \mathbf{h}_t^{(k')}, \mathbf{y}_t^{(k)}) \\
&\quad + \frac{1}{K} \sum_{k=1}^K \nabla_y f^{(k)}(\frac{1}{K} \sum_{k'=1}^K \mathbf{h}_t^{(k')}, \mathbf{y}_t^{(k)}) - \frac{1}{K} \sum_{k=1}^K \nabla_y f^{(k)}(\mathbf{h}_t^{(k)}, \mathbf{y}_t^{(k)})\Big\|^2\Big] \\
&\leq 3\mathbb{E}\Big[\Big\|\frac{1}{K} \sum_{k=1}^K \nabla_y f^{(k)}(g^{(k)}(\mathbf{x}_t^{(k)}), \mathbf{y}_t^{(k)}) - \frac{1}{K} \sum_{k=1}^K \nabla_y f^{(k)}(\frac{1}{K} \sum_{k'=1}^K g^{(k')}(\mathbf{x}_t^{(k')}), \mathbf{y}_t^{(k)})\Big\|^2\Big] \\
&\quad + 3\mathbb{E}\Big[\Big\|\frac{1}{K} \sum_{k=1}^K \nabla_y f^{(k)}(\frac{1}{K} \sum_{k'=1}^K g^{(k')}(\mathbf{x}_t^{(k')}), \mathbf{y}_t^{(k)}) - \frac{1}{K} \sum_{k=1}^K \nabla_y f^{(k)}(\frac{1}{K} \sum_{k'=1}^K \mathbf{h}_t^{(k')}, \mathbf{y}_t^{(k)})\Big\|^2\Big] \\
&\quad + 3\mathbb{E}\Big[\Big\|\frac{1}{K} \sum_{k=1}^K \nabla_y f^{(k)}(\frac{1}{K} \sum_{k'=1}^K \mathbf{h}_t^{(k')}, \mathbf{y}_t^{(k)}) - \frac{1}{K} \sum_{k=1}^K \nabla_y f^{(k)}(\mathbf{h}_t^{(k)}, \mathbf{y}_t^{(k)})\Big\|^2\Big] \\
&\leq 3L_f^2 \frac{1}{K} \sum_{k=1}^K \mathbb{E}\Big[\Big\|g^{(k)}(\mathbf{x}_t^{(k)}) - \frac{1}{K} \sum_{k'=1}^K g^{(k')}(\mathbf{x}_t^{(k')})\Big\|^2\Big] + 3L_f^2 \mathbb{E}\Big[\Big\|\frac{1}{K} \sum_{k=1}^K g^{(k)}(\mathbf{x}_t^{(k)}) - \frac{1}{K} \sum_{k=1}^K \mathbf{h}_t^{(k)}\Big\|^2\Big] \\
&\quad + 3L_f^2 \frac{1}{K} \sum_{k=1}^K \mathbb{E}\Big[\Big\|\frac{1}{K} \sum_{k'=1}^K \mathbf{h}_t^{(k')} - \mathbf{h}_t^{(k)}\Big\|^2\Big] \\
&\leq 72\gamma_x^2 \beta_x^2 p^4 \eta^4 C_g^4 C_f^2 L_f^2 + 108\alpha^2 p^2 \eta^2 C_g^2 L_f^2 \sigma_g^2 + 1296\alpha^2 \gamma_x^2 \beta_x^2 p^6 \eta^6 C_g^6 C_f^2 L_f^2 \\
&\quad + 3L_f^2 \mathbb{E}\Big[\Big\|\bar{\mathbf{h}}_{t-1} - \frac{1}{K} \sum_{k=1}^K g^{(k)}(\mathbf{x}_{t-1}^{(k)})\Big\|^2\Big] + \frac{6\eta L_f^2 \gamma_x^2 C_g^2}{\alpha} \mathbb{E}[\|\bar{\mathbf{u}}_{t-1}\|^2] + \frac{36\gamma_x^2 \beta_x^2 p^2 \eta^3 C_g^4 C_f^2 L_f^2}{\alpha} + \frac{3\alpha^2 \eta^2 \sigma_g^2 L_f^2}{K} \\
&\leq 72\gamma_x^2 \beta_x^2 p^4 \eta^4 C_g^4 C_f^2 L_f^2 + 108\alpha^2 p^2 \eta^2 C_g^2 L_f^2 \sigma_g^2 + 1296\alpha^2 \gamma_x^2 \beta_x^2 p^6 \eta^6 C_g^6 C_f^2 L_f^2 \\
&\quad + 3L_f^2 \mathbb{E}\Big[\Big\|\bar{\mathbf{h}}_{t-1} - \frac{1}{K} \sum_{k=1}^K g^{(k)}(\mathbf{x}_{t-1}^{(k)})\Big\|^2\Big] + \frac{6L_f^2 \gamma_x^2 C_g^2}{\alpha} \mathbb{E}[\|\bar{\mathbf{u}}_{t-1}\|^2] + \frac{36\gamma_x^2 \beta_x^2 p^2 \eta^2 C_g^4 C_f^2 L_f^2}{\alpha} + \frac{3\alpha^2 \eta \sigma_g^2 L_f^2}{K} \, ,
\end{aligned}
\tag{75}
$$

where the last step holds due to Lemmas 8, 9, and Eq. (56), the last step holds due to $\eta < 1$.

Then, combining $T_1$, $T_2$, Lemma 4 and Lemma 5, we can get

$$\mathbb{E}\Big[\Big\|\frac{1}{K}\sum_{k=1}^{K}\nabla_y f^{(k)}(g^{(k)}(\mathbf{x}_t^{(k)}),\mathbf{y}_t^{(k)})-\frac{1}{K}\sum_{k=1}^{K}\mathbf{v}_t^{(k)}\Big\|^2\Big]$$

$$\leq (1-\beta_y\eta)\mathbb{E}\Big[\Big\|\frac{1}{K}\sum_{k=1}^{K}\nabla_y f^{(k)}(g^{(k)}(\mathbf{x}_{t-1}^{(k)}),\mathbf{y}_{t-1}^{(k)})-\frac{1}{K}\sum_{k=1}^{K}\mathbf{v}_{t-1}^{(k)}\Big\|^2\Big]+6\beta_y\eta L_f^2\mathbb{E}\Big[\Big\|\bar{\mathbf{h}}_{t-1}-\frac{1}{K}\sum_{k=1}^{K}g^{(k)}(\mathbf{x}_{t-1}^{(k)})\Big\|^2\Big]$$

$$+\frac{12\beta_y\eta L_f^2\gamma_x^2 C_g^2}{\alpha}\mathbb{E}[\|\bar{\mathbf{u}}_{t-1}\|^2]+\frac{4\gamma_x^2\eta L_f^2 C_g^2}{\beta_y}\mathbb{E}[\|\bar{\mathbf{u}}_{t-1}\|^2]+\frac{4\gamma_y^2\eta L_f^2}{\beta_y}\mathbb{E}[\|\bar{\mathbf{v}}_{t-1}\|^2]$$

$$+144\beta_y\gamma_x^2\beta_x^2 p^4\eta^5 C_g^4 C_f^2 L_f^2+216\beta_y\alpha^2 p^2\eta^3 C_g^2 L_f^2\sigma_g^2+2592\beta_y\alpha^2\gamma_x^2\beta_x^2 p^6\eta^7 C_g^6 C_f^2 L_f^2$$

$$+13824\beta_y\gamma_y^2\alpha^2 p^4\eta^5 C_g^2 L_f^4\sigma_g^2+165888\beta_y\gamma_y^2\alpha^2\gamma_x^2\beta_x^2 p^8\eta^9 C_g^6 C_f^2 L_f^4+384\beta_y\gamma_y^2 p^2\eta^3 L_f^2\sigma_f^2$$

$$+\frac{72\beta_y\gamma_x^2\beta_x^2 p^2\eta^3 C_g^4 C_f^2 L_f^2}{\alpha}+\frac{6\beta_y\alpha^2\eta^2\sigma_g^2 L_f^2}{K}+\frac{\beta_y^2\eta^2\sigma_f^2}{K}+\frac{24\gamma_x^2 p^2\beta_x^2\eta^3 L_f^2 C_g^4 C_f^2}{\beta_y}. \tag{76}$$

It can be reformulated as

$$\beta_y\eta\mathbb{E}\Big[\Big\|\frac{1}{K}\sum_{k=1}^{K}\nabla_y f^{(k)}(g^{(k)}(\mathbf{x}_t^{(k)}),\mathbf{y}_t^{(k)})-\frac{1}{K}\sum_{k=1}^{K}\mathbf{v}_t^{(k)}\Big\|^2\Big]$$

$$\leq\mathbb{E}\Big[\Big\|\frac{1}{K}\sum_{k=1}^{K}\nabla_y f^{(k)}(g^{(k)}(\mathbf{x}_t^{(k)}),\mathbf{y}_t^{(k)})-\frac{1}{K}\sum_{k=1}^{K}\mathbf{v}_t^{(k)}\Big\|^2\Big]-\mathbb{E}\Big[\Big\|\frac{1}{K}\sum_{k=1}^{K}\nabla_y f^{(k)}(g^{(k)}(\mathbf{x}_{t+1}^{(k)}),\mathbf{y}_{t+1}^{(k)})-\frac{1}{K}\sum_{k=1}^{K}\mathbf{v}_{t+1}^{(k)}\Big\|^2\Big]$$

$$+6\beta_y\eta L_f^2\mathbb{E}\Big[\Big\|\bar{\mathbf{h}}_t-\frac{1}{K}\sum_{k=1}^{K}g^{(k)}(\mathbf{x}_t^{(k)})\Big\|^2\Big]+\frac{12\beta_y\eta L_f^2\gamma_x^2 C_g^2}{\alpha}\mathbb{E}[\|\bar{\mathbf{u}}_t\|^2]+\frac{4\gamma_x^2\eta L_f^2 C_g^2}{\beta_y}\mathbb{E}[\|\bar{\mathbf{u}}_t\|^2]+\frac{4\gamma_y^2\eta L_f^2}{\beta_y}\mathbb{E}[\|\bar{\mathbf{v}}_t\|^2]$$

$$+144\beta_y\gamma_x^2\beta_x^2 p^4\eta^5 C_g^4 C_f^2 L_f^2+216\beta_y\alpha^2 p^2\eta^3 C_g^2 L_f^2\sigma_g^2+2592\beta_y\alpha^2\gamma_x^2\beta_x^2 p^6\eta^7 C_g^6 C_f^2 L_f^2$$

$$+13824\beta_y\gamma_y^2\alpha^2 p^4\eta^5 C_g^2 L_f^4\sigma_g^2+165888\beta_y\gamma_y^2\alpha^2\gamma_x^2\beta_x^2 p^8\eta^9 C_g^6 C_f^2 L_f^4+384\beta_y\gamma_y^2 p^2\eta^3 L_f^2\sigma_f^2$$

$$+\frac{72\beta_y\gamma_x^2\beta_x^2 p^2\eta^3 C_g^4 C_f^2 L_f^2}{\alpha}+\frac{6\beta_y\alpha^2\eta^2\sigma_g^2 L_f^2}{K}+\frac{\beta_y^2\eta^2\sigma_f^2}{K}+\frac{24\gamma_x^2 p^2\beta_x^2\eta^3 L_f^2 C_g^4 C_f^2}{\beta_y}. \tag{77}$$

By summing over $t$ from 0 to $T-1$, we can get

$$\frac{1}{T}\sum_{t=0}^{T-1}\mathbb{E}\Big[\Big\|\frac{1}{K}\sum_{k=1}^{K}\nabla_y f^{(k)}(g^{(k)}(\mathbf{x}_t^{(k)}),\mathbf{y}_t^{(k)})-\frac{1}{K}\sum_{k=1}^{K}\mathbf{v}_t^{(k)}\Big\|^2\Big]$$

$$\leq\frac{1}{\beta_y\eta T}\mathbb{E}\Big[\Big\|\frac{1}{K}\sum_{k=1}^{K}\nabla_y f^{(k)}(g^{(k)}(\mathbf{x}_0^{(k)}),\mathbf{y}_0^{(k)})-\frac{1}{K}\sum_{k=1}^{K}\mathbf{v}_0^{(k)}\Big\|^2\Big]+6L_f^2\frac{1}{T}\sum_{t=0}^{T-1}\mathbb{E}\Big[\Big\|\bar{\mathbf{h}}_t-\frac{1}{K}\sum_{k=1}^{K}g^{(k)}(\mathbf{x}_t^{(k)})\Big\|^2\Big]$$

$$+\frac{12L_f^2\gamma_x^2 C_g^2}{\alpha}\frac{1}{T}\sum_{t=0}^{T-1}\mathbb{E}[\|\bar{\mathbf{u}}_t\|^2]+\frac{4\gamma_x^2 L_f^2 C_g^2}{\beta_y^2}\frac{1}{T}\sum_{t=0}^{T-1}\mathbb{E}[\|\bar{\mathbf{u}}_t\|^2]+\frac{4\gamma_y^2 L_f^2}{\beta_y^2}\frac{1}{T}\sum_{t=0}^{T-1}\mathbb{E}[\|\bar{\mathbf{v}}_t\|^2]$$

$$+144\gamma_x^2\beta_x^2 p^4\eta^4 C_g^4 C_f^2 L_f^2+216\alpha^2 p^2\eta^2 C_g^2 L_f^2\sigma_g^2+2592\alpha^2\gamma_x^2\beta_x^2 p^6\eta^6 C_g^6 C_f^2 L_f^2$$

$$+13824\gamma_y^2\alpha^2 p^4\eta^4 C_g^2 L_f^4\sigma_g^2+165888\gamma_y^2\alpha^2\gamma_x^2\beta_x^2 p^8\eta^8 C_g^6 C_f^2 L_f^4+384\gamma_y^2 p^2\eta^2 L_f^2\sigma_f^2$$

$$+\frac{72\gamma_x^2\beta_x^2 p^2\eta^2 C_g^4 C_f^2 L_f^2}{\alpha}+\frac{6\alpha^2\eta\sigma_g^2 L_f^2}{K}+\frac{\beta_y\eta\sigma_f^2}{K}+\frac{24\gamma_x^2 p^2\beta_x^2\eta^2 L_f^2 C_g^4 C_f^2}{\beta_y^2}$$

$$\leq\frac{2L_f^2\sigma_g^2+2\sigma_f^2}{\beta_y\eta T}+6L_f^2\frac{1}{T}\sum_{t=0}^{T-1}\mathbb{E}\Big[\Big\|\bar{\mathbf{h}}_t-\frac{1}{K}\sum_{k=1}^{K}g^{(k)}(\mathbf{x}_t^{(k)})\Big\|^2\Big]$$

$$+\frac{12L_f^2\gamma_x^2 C_g^2}{\alpha}\frac{1}{T}\sum_{t=0}^{T-1}\mathbb{E}[\|\bar{\mathbf{u}}_t\|^2]+\frac{4\gamma_x^2 L_f^2 C_g^2}{\beta_y^2}\frac{1}{T}\sum_{t=0}^{T-1}\mathbb{E}[\|\bar{\mathbf{u}}_t\|^2]+\frac{4\gamma_y^2 L_f^2}{\beta_y^2}\frac{1}{T}\sum_{t=0}^{T-1}\mathbb{E}[\|\bar{\mathbf{v}}_t\|^2]$$

$$+144\gamma_x^2\beta_x^2 p^4\eta^4 C_g^4 C_f^2 L_f^2+216\alpha^2 p^2\eta^2 C_g^2 L_f^2\sigma_g^2+2592\alpha^2\gamma_x^2\beta_x^2 p^6\eta^6 C_g^6 C_f^2 L_f^2$$

$$+13824\gamma_y^2\alpha^2 p^4\eta^4 C_g^2 L_f^4\sigma_g^2+165888\gamma_y^2\alpha^2\gamma_x^2\beta_x^2 p^8\eta^8 C_g^6 C_f^2 L_f^4+384\gamma_y^2 p^2\eta^2 L_f^2\sigma_f^2$$

$$+\frac{72\gamma_x^2\beta_x^2 p^2\eta^2 C_g^4 C_f^2 L_f^2}{\alpha}+\frac{6\alpha^2\eta\sigma_g^2 L_f^2}{K}+\frac{\beta_y\eta\sigma_f^2}{K}+\frac{24\gamma_x^2 p^2\beta_x^2\eta^2 L_f^2 C_g^4 C_f^2}{\beta_y^2}. \tag{78}$$

In addition, we have

$$
\mathbb{E}\Big[\Big\|\frac{1}{K}\sum_{k=1}^{K}\nabla_y f^{(k)}(g^{(k)}(\mathbf{x}_0^{(k)}), \mathbf{y}_0^{(k)}) - \frac{1}{K}\sum_{k=1}^{K}\mathbf{v}_0^{(k)}\Big\|^2\Big]
$$

$$
= \mathbb{E}\Big[\Big\|\frac{1}{K}\sum_{k=1}^{K}\nabla_y f^{(k)}(g^{(k)}(\mathbf{x}_0^{(k)}), \mathbf{y}_0^{(k)}) - \frac{1}{K}\sum_{k=1}^{K}\nabla_y f^{(k)}(g^{(k)}(\mathbf{x}_0^{(k)};\xi_0^{(k)}), \mathbf{y}_0^{(k)};\zeta_0^{(k)})\Big\|^2\Big]
$$

$$
= \mathbb{E}\Big[\Big\|\frac{1}{K}\sum_{k=1}^{K}\nabla_y f^{(k)}(g^{(k)}(\mathbf{x}_0^{(k)}), \mathbf{y}_0^{(k)}) - \frac{1}{K}\sum_{k=1}^{K}\nabla_y f^{(k)}(g^{(k)}(\mathbf{x}_0^{(k)};\xi_0^{(k)}), \mathbf{y}_0^{(k)})
$$

$$
\qquad + \frac{1}{K}\sum_{k=1}^{K}\nabla_y f^{(k)}(g^{(k)}(\mathbf{x}_0^{(k)};\xi_0^{(k)}), \mathbf{y}_0^{(k)}) - \frac{1}{K}\sum_{k=1}^{K}\nabla_y f^{(k)}(g^{(k)}(\mathbf{x}_0^{(k)};\xi_0^{(k)}), \mathbf{y}_0^{(k)};\zeta_0^{(k)})\Big\|^2\Big] \tag{79}
$$

$$
\leq 2L_f^2 \frac{1}{K}\sum_{k=1}^{K}\mathbb{E}\Big[\Big\|g^{(k)}(\mathbf{x}_0^{(k)}) - g^{(k)}(\mathbf{x}_0^{(k)};\xi_0^{(k)})\Big\|^2\Big] + 2\sigma_f^2
$$

$$
\leq 2L_f^2\sigma_g^2 + 2\sigma_f^2 ,
$$

which completes the proof. $\qquad\qquad\square$

**Lemma 15.** *Given Assumption 1-4 and if $\gamma_y \leq \frac{1}{6L_f}$ and $\eta \leq 1$, we have*

$$
\frac{1}{T}\sum_{t=0}^{T-1}\|\bar{\mathbf{y}}_t - \mathbf{y}^*(\bar{\mathbf{x}}_t)\|^2 \leq \frac{4}{\eta\gamma_y\mu T}\|\bar{\mathbf{y}}_0 - \mathbf{y}^*(\bar{\mathbf{x}}_0)\|^2 - \frac{3\gamma_y}{\mu}\frac{1}{T}\sum_{t=0}^{T-1}[\|\bar{\mathbf{v}}_t\|^2] + \frac{50\gamma_x^2 C_g^2 L_f^2}{3\gamma_y^2\mu^4}\frac{1}{T}\sum_{t=0}^{T-1}[\|\bar{\mathbf{u}}_t\|^2]
$$

$$
+ \frac{50}{\mu^2}\frac{1}{T}\sum_{t=0}^{T-1}\Big[\Big\|\frac{1}{K}\sum_{k=1}^{K}\nabla_y f^{(k)}(g^{(k)}(\mathbf{x}_t^{(k)}), \mathbf{y}_t^{(k)}) - \frac{1}{K}\sum_{k=1}^{K}\mathbf{v}_t^{(k)}\Big\|^2\Big] + \frac{300\gamma_x^2\beta_x^2 p^4\eta^4 C_g^4 C_f^2 L_f^2}{\mu^2}
$$

$$
+ \frac{100}{\mu^2}(1728\gamma_y^2\beta_y^2\alpha^2 p^6\eta^6 C_g^2 L_f^4\sigma_g^2 + 20736\gamma_y^2\beta_y^2\alpha^2\gamma_x^2\beta_x^2 p^{10}\eta^{10}C_g^6 C_f^2 L_f^4 + 48\gamma_y^2\beta_y^2 p^4\eta^4 L_f^2\sigma_f^2) . \tag{80}
$$

*Proof.*

$$\|\bar{\mathbf{y}}_{t+1} - \mathbf{y}^*(\bar{\mathbf{x}}_{t+1})\|^2$$

$$\leq (1 - \frac{\eta\gamma_y\mu}{4})\|\bar{\mathbf{y}}_t - \mathbf{y}^*(\bar{\mathbf{x}}_t)\|^2 - \frac{3\eta\gamma_y^2}{4}\|\bar{\mathbf{v}}_t\|^2 + \frac{25\eta\gamma_x^2 C_g^2 L_f^2}{6\gamma_y\mu^3}\|\bar{\mathbf{u}}_t\|^2 + \frac{25\eta\gamma_y}{6\mu}\|\nabla_y f(g(\bar{\mathbf{x}}_t), \bar{\mathbf{y}}_t) - \bar{\mathbf{v}}_t\|^2$$

$$\leq (1 - \frac{\eta\gamma_y\mu}{4})\|\bar{\mathbf{y}}_t - \mathbf{y}^*(\bar{\mathbf{x}}_t)\|^2 - \frac{3\eta\gamma_y^2}{4}\|\bar{\mathbf{v}}_t\|^2 + \frac{25\eta\gamma_x^2 C_g^2 L_f^2}{6\gamma_y\mu^3}\|\bar{\mathbf{u}}_t\|^2$$

$$+ \frac{25\eta\gamma_y}{6\mu}\Big[\Big\|\frac{1}{K}\sum_{k=1}^{K}\nabla_y f^{(k)}(\frac{1}{K}\sum_{k'=1}^{K}g^{(k')}(\bar{\mathbf{x}}_t), \bar{\mathbf{y}}_t) - \frac{1}{K}\sum_{k=1}^{K}\nabla_y f^{(k)}(g^{(k)}(\bar{\mathbf{x}}_t), \mathbf{y}_t^{(k)})$$

$$+ \frac{1}{K}\sum_{k=1}^{K}\nabla_y f^{(k)}(g^{(k)}(\bar{\mathbf{x}}_t), \mathbf{y}_t^{(k)}) - \frac{1}{K}\sum_{k=1}^{K}\nabla_y f^{(k)}(g^{(k)}(\mathbf{x}_t^{(k)}), \mathbf{y}_t^{(k)})$$

$$+ \frac{1}{K}\sum_{k=1}^{K}\nabla_y f^{(k)}(g^{(k)}(\mathbf{x}_t^{(k)}), \mathbf{y}_t^{(k)}) - \frac{1}{K}\sum_{k=1}^{K}\mathbf{v}_t^{(k)}\Big\|^2\Big]$$

$$\leq (1 - \frac{\eta\gamma_y\mu}{4})\|\bar{\mathbf{y}}_t - \mathbf{y}^*(\bar{\mathbf{x}}_t)\|^2 - \frac{3\eta\gamma_y^2}{4}\|\bar{\mathbf{v}}_t\|^2 + \frac{25\eta\gamma_x^2 C_g^2 L_f^2}{6\gamma_y\mu^3}\|\bar{\mathbf{u}}_t\|^2$$

$$+ \frac{25\eta\gamma_y}{2\mu}\Big[\Big\|\frac{1}{K}\sum_{k=1}^{K}\nabla_y f^{(k)}(\frac{1}{K}\sum_{k'=1}^{K}g^{(k')}(\bar{\mathbf{x}}_t), \bar{\mathbf{y}}_t) - \frac{1}{K}\sum_{k=1}^{K}\nabla_y f^{(k)}(g^{(k)}(\bar{\mathbf{x}}_t), \mathbf{y}_t^{(k)})\Big\|^2\Big]$$

$$+ \frac{25\eta\gamma_y}{2\mu}\Big[\Big\|\frac{1}{K}\sum_{k=1}^{K}\nabla_y f^{(k)}(g^{(k)}(\bar{\mathbf{x}}_t), \mathbf{y}_t^{(k)}) - \frac{1}{K}\sum_{k=1}^{K}\nabla_y f^{(k)}(g^{(k)}(\mathbf{x}_t^{(k)}), \mathbf{y}_t^{(k)})\Big\|^2\Big]$$

$$+ \frac{25\eta\gamma_y}{2\mu}\Big[\Big\|\frac{1}{K}\sum_{k=1}^{K}\nabla_y f^{(k)}(g^{(k)}(\mathbf{x}_t^{(k)}), \mathbf{y}_t^{(k)}) - \frac{1}{K}\sum_{k=1}^{K}\mathbf{v}_t^{(k)}\Big\|^2\Big]$$

$$\leq (1 - \frac{\eta\gamma_y\mu}{4})\|\bar{\mathbf{y}}_t - \mathbf{y}^*(\bar{\mathbf{x}}_t)\|^2 - \frac{3\eta\gamma_y^2}{4}\|\bar{\mathbf{v}}_t\|^2 + \frac{25\eta\gamma_x^2 C_g^2 L_f^2}{6\gamma_y\mu^3}\|\bar{\mathbf{u}}_t\|^2 + \frac{25\eta\gamma_y L_f^2}{2\mu}\frac{1}{K}\sum_{k=1}^{K}\Big[\Big\|\bar{\mathbf{y}}_t - \mathbf{y}_t^{(k)}\Big\|^2\Big]$$

$$+ \frac{25\eta\gamma_y L_f^2 C_g^2}{2\mu}\frac{1}{K}\sum_{k=1}^{K}\Big[\Big\|\bar{\mathbf{x}}_t - \mathbf{x}_t^{(k)}\Big\|^2\Big] + \frac{25\eta\gamma_y}{2\mu}\Big[\Big\|\frac{1}{K}\sum_{k=1}^{K}\nabla_y f^{(k)}(g^{(k)}(\mathbf{x}_t^{(k)}), \mathbf{y}_t^{(k)}) - \frac{1}{K}\sum_{k=1}^{K}\mathbf{v}_t^{(k)}\Big\|^2\Big]$$

$$\leq (1 - \frac{\eta\gamma_y\mu}{4})[\|\bar{\mathbf{y}}_t - \mathbf{y}^*(\bar{\mathbf{x}}_t)\|^2] - \frac{3\eta\gamma_y^2}{4}[\|\bar{\mathbf{v}}_t\|^2] + \frac{25\eta\gamma_x^2 C_g^2 L_f^2}{6\gamma_y\mu^3}[\|\bar{\mathbf{u}}_t\|^2]$$

$$+ \frac{25}{\mu}(1728\gamma_y^3\beta_y^2\alpha^2 p^6\eta^7 C_g^2 L_f^4\sigma_g^2 + 20736\gamma_y^3\beta_y^2\alpha^2\gamma_x^2\beta_x^2 p^{10}\eta^{11} C_g^6 C_f^2 L_f^4 + 48\gamma_y^3\beta_y^2 p^4\eta^5 L_f^2\sigma_f^2)$$

$$+ \frac{75\gamma_y\gamma_x^2\beta_x^2 p^4\eta^5 C_g^4 C_f^2 L_f^2}{\mu} + \frac{25\eta\gamma_y}{2\mu}\Big[\Big\|\frac{1}{K}\sum_{k=1}^{K}\nabla_y f^{(k)}(g^{(k)}(\mathbf{x}_t^{(k)}), \mathbf{y}_t^{(k)}) - \frac{1}{K}\sum_{k=1}^{K}\mathbf{v}_t^{(k)}\Big\|^2\Big],$$

$$(81)$$

where the first step holds due to Lemma 5 in [5], the second step holds due to the homogeneous data distribution assumption, the second to last step holds due to Assumption 1 and Assumption 2, the last step holds due to Lemmas 6, 7. We further reformulate it as follows:

$$\frac{\eta\gamma_y\mu}{4}\|\bar{\mathbf{y}}_t - \mathbf{y}^*(\bar{\mathbf{x}}_t)\|^2 \leq \|\bar{\mathbf{y}}_t - \mathbf{y}^*(\bar{\mathbf{x}}_t)\|^2 - \|\bar{\mathbf{y}}_{t+1} - \mathbf{y}^*(\bar{\mathbf{x}}_{t+1})\|^2 - \frac{3\eta\gamma_y^2}{4}[\|\bar{\mathbf{v}}_t\|^2] + \frac{25\eta\gamma_x^2 C_g^2 L_f^2}{6\gamma_y\mu^3}[\|\bar{\mathbf{u}}_t\|^2]$$

$$+ \frac{25\eta\gamma_y}{2\mu}\Big[\Big\|\frac{1}{K}\sum_{k=1}^{K}\nabla_y f^{(k)}(g^{(k)}(\mathbf{x}_t^{(k)}), \mathbf{y}_t^{(k)}) - \frac{1}{K}\sum_{k=1}^{K}\mathbf{v}_t^{(k)}\Big\|^2\Big] + \frac{75\gamma_y\gamma_x^2\beta_x^2 p^4\eta^5 C_g^4 C_f^2 L_f^2}{\mu}$$

$$+ \frac{25}{\mu}(1728\gamma_y^3\beta_y^2\alpha^2 p^6\eta^7 C_g^2 L_f^4\sigma_g^2 + 20736\gamma_y^3\beta_y^2\alpha^2\gamma_x^2\beta_x^2 p^{10}\eta^{11} C_g^6 C_f^2 L_f^4 + 48\gamma_y^3\beta_y^2 p^4\eta^5 L_f^2\sigma_f^2).$$

$$(82)$$

By summing over $t$ from 0 to $T-1$, we can complete the proof. $\qquad\square$

Based on the aforementioned lemmas, we are ready to prove Theorem 1.

*Proof.* At first, from Lemmas 3, we can get

$$\frac{\gamma_x\eta}{2}\mathbb{E}[\|\nabla\Phi(\bar{\mathbf{x}}_t)\|^2] \leq \mathbb{E}[\Phi(\bar{\mathbf{x}}_t)] - \mathbb{E}[\Phi(\bar{\mathbf{x}}_{t+1})] - \frac{\gamma_x\eta}{4}\mathbb{E}[\|\bar{\mathbf{u}}_t\|^2] + 3\gamma_x\eta C_g^2 L_f^2\mathbb{E}[\|\mathbf{y}_*(\bar{\mathbf{x}}_t) - \bar{\mathbf{y}}_t\|^2]$$

$$+ 12\gamma_x\eta(C_g^4 L_f^2 + C_f^2 L_g^2)\frac{1}{K}\sum_{k=1}^{K}\left[\left\|\bar{\mathbf{x}}_t - \mathbf{x}_t^{(k)}\right\|^2\right] + 6\gamma_x\eta C_g^2 L_f^2 \frac{1}{K}\sum_{k=1}^{K}\mathbb{E}\left[\left\|\bar{\mathbf{y}}_t - \mathbf{y}_t^{(k)}\right\|^2\right]$$

$$+ 3\gamma_x\eta\mathbb{E}\left[\left\|\frac{1}{K}\sum_{k=1}^{K}\nabla_{\mathbf{x}}f^{(k)}(g^{(k)}(\mathbf{x}_t^{(k)}), \mathbf{y}_t^{(k)}) - \bar{\mathbf{u}}_t\right\|^2\right].$$

(83)

By summing it over $t$ from 0 to $T-1$, we can get

$$\frac{1}{T}\sum_{t=0}^{T-1}\mathbb{E}[\|\nabla\Phi(\bar{\mathbf{x}}_t)\|^2] \leq \frac{2(\Phi(\bar{\mathbf{x}}_0) - \Phi(\bar{\mathbf{x}}_T))}{\gamma_x\eta T} - \frac{1}{2}\frac{1}{T}\sum_{t=0}^{T-1}\mathbb{E}[\|\bar{\mathbf{u}}_t\|^2]$$

$$+ 24(C_g^4 L_f^2 + C_f^2 L_g^2)\frac{1}{T}\sum_{t=0}^{T-1}\frac{1}{K}\sum_{k=1}^{K}\left[\left\|\bar{\mathbf{x}}_t - \mathbf{x}_t^{(k)}\right\|^2\right] + 12C_g^2 L_f^2 \frac{1}{T}\sum_{t=0}^{T-1}\frac{1}{K}\sum_{k=1}^{K}\mathbb{E}\left[\left\|\bar{\mathbf{y}}_t - \mathbf{y}_t^{(k)}\right\|^2\right]$$

$$+ 6\frac{1}{T}\sum_{t=0}^{T-1}\mathbb{E}\left[\left\|\frac{1}{K}\sum_{k=1}^{K}\nabla_{\mathbf{x}}f^{(k)}(g^{(k)}(\mathbf{x}_t^{(k)}), \mathbf{y}_t^{(k)}) - \bar{\mathbf{u}}_t\right\|^2\right] + 6C_g^2 L_f^2 \frac{1}{T}\sum_{t=0}^{T-1}\mathbb{E}[\|\mathbf{y}_*(\bar{\mathbf{x}}_t) - \bar{\mathbf{y}}_t\|^2]$$

$$\leq \frac{2(\Phi(\bar{\mathbf{x}}_0) - \Phi(\bar{\mathbf{x}}_T))}{\gamma_x\eta T} + \frac{24C_g^2 L_f^2}{\gamma_y\eta\mu T}\|\bar{\mathbf{y}}_0 - \mathbf{y}^*(\bar{\mathbf{x}}_0)\|^2$$

$$+ \left(\frac{300C_g^2 L_f^2(1 + 6L_f^2)}{\mu^2}\frac{2\gamma_x^2 C_g^2}{\alpha^2} + \frac{300C_g^2 L_f^2}{\mu^2}\frac{4\gamma_x^2 L_f^2 C_g^2}{\beta_y^2} + \frac{300C_g^2 L_f^2}{\mu^2}\frac{12L_f^2\gamma_x^2 C_g^2}{\alpha} + \frac{100\gamma_x^2 C_g^4 L_f^4}{\gamma_y\mu^4}\right.$$

$$\left. + \frac{72\gamma_x^2 C_g^4}{\alpha} + \frac{48\gamma_x^2(C_g^4 L_f^2 + C_f^2 L_g^2)}{\beta_x^2} - \frac{1}{2}\right)\frac{1}{T}\sum_{t=0}^{T-1}\mathbb{E}[\|\bar{\mathbf{u}}_t\|^2]$$

$$+ \left(\frac{300C_g^2 L_f^2}{\mu^2}\frac{4\gamma_y^2 L_f^2}{\beta_y^2} + \frac{48\gamma_y^2 C_g^2 L_f^2}{\beta_x^2} - \frac{18\gamma_y C_g^2 L_f^2}{\mu}\right)\frac{1}{T}\sum_{t=0}^{T-1}\mathbb{E}[\|\bar{\mathbf{v}}_t\|^2]$$

$$+ \frac{300C_g^2 L_f^2(1 + 6L_f^2)}{\mu^2}\frac{\sigma_g^2}{\alpha\eta TK} + \frac{300C_g^2 L_f^2(1 + 6L_f^2)}{\mu^2}\frac{12\gamma_x^2\beta_x^2 p^2\eta^2 C_g^4 C_f^2}{\alpha^2} + \frac{300C_g^2 L_f^2(1 + 6L_f^2)}{\mu^2}\frac{\alpha\eta\sigma_g^2}{K}$$

$$+ \frac{600(L_f^2\sigma_g^2 + \sigma_f^2)C_g^2 L_f^2}{\beta_y\eta T\mu^2} + \frac{43200\gamma_x^2\beta_x^2 p^4\eta^4 C_g^6 C_f^2 L_f^4}{\mu^2} + \frac{64800\alpha^2 p^2\eta^2 C_g^4 L_f^6\sigma_g^2}{\mu^2} + \frac{7776000\alpha^2\gamma_x^2\beta_x^2 p^6\eta^6 C_g^8 C_f^2 L_f^4}{\mu^2}$$

$$+ \frac{4147200\gamma_y^2\alpha^2 p^4\eta^4 C_g^4 L_f^6\sigma_g^2}{\mu^2} + \frac{49766400\gamma_y^2\alpha^2\gamma_x^2\beta_x^2 p^8\eta^8 C_g^8 C_f^2 L_f^6}{\mu^2} + \frac{115200\gamma_y^2 p^2\eta^2 C_g^2 L_f^4\sigma_f^2}{\mu^2}$$

$$+ \frac{21600\gamma_x^2\beta_x^2 p^2\eta^2 C_g^6 C_f^2 L_f^4}{\alpha\mu^2} + \frac{1800\alpha^2\eta\sigma_g^2 C_g^2 L_f^4}{K\mu^2} + \frac{300\beta_y\eta\sigma_f^2 C_g^2 L_f^2}{K\mu^2} + \frac{7200\gamma_x^2 p^2\beta_x^2\eta^2 L_f^4 C_g^6 C_f^2}{\beta_y^2\mu^2}$$

$$+ 144(C_g^4 L_f^2 + C_f^2 L_g^2)\gamma_x^2\beta_x^2 p^4\eta^4 C_g^2 C_f^2 + 41472\gamma_y^2\beta_y^2\alpha^2 p^6\eta^6 C_g^4 L_f^4\sigma_g^2 + 497664\gamma_y^2\beta_y^2\alpha^2\gamma_x^2\beta_x^2 p^{10}\eta^{10} C_g^8 C_f^2 L_f^4$$

$$+ 1152\gamma_y^2\beta_y^2 p^4\eta^4 C_g^2 L_f^2\sigma_f^2 + \frac{6(3C_g^2 L_f^2\sigma_g^2 + 3C_f^2\sigma_{g'}^2 + 3C_g^2\sigma_f^2)}{\beta_x\eta T} + 288\gamma_x^2 p^2\eta^2(C_g^4 L_f^2 + C_f^2 L_g^2)C_g^2 C_f^2 + 864\gamma_x^2\beta_x^2 p^4\eta^4 C_g^6 C_f^4$$

$$+ 1296\alpha^2 p^2\eta^2 C_g^4\sigma_g^2 + 15552\alpha^2\gamma_x^2\beta_x^2 p^6\eta^6 C_g^8 C_f^2 + \frac{12\beta_x\eta C_f^2\sigma_{g'}^2}{K} + \frac{12\beta_x\eta C_g^2\sigma_f^2}{K}$$

$$+ \frac{165888\beta_y^2\gamma_y^2\alpha^2 p^4\eta^4 C_g^4 L_f^4\sigma_g^2}{\beta_x^2} + \frac{1990656\beta_y^2\gamma_y^2\alpha^2\gamma_x^2\beta_x^2 p^8\eta^8 C_g^8 C_f^2 L_f^4}{\beta_x^2} + \frac{4716\beta_y^2\gamma_y^2 p^2\eta^2 C_g^2 L_f^2\sigma_f^2}{\beta_x^2}$$

$$+ \frac{432\gamma_x^2\beta_x^2 p^2\eta^2 C_g^6 C_f^2}{\alpha} + \frac{36\alpha^2\eta C_g^2\sigma_g^2}{K} + \frac{1800\gamma_x^2\beta_x^2 p^4\eta^4 C_g^6 C_f^2 L_f^4}{\mu^2}$$

$$+ \frac{600}{\mu^2}(1728\gamma_y^2\beta_y^2\alpha^2 p^6\eta^6 C_g^4 L_f^6\sigma_g^2 + 20736\gamma_y^2\beta_y^2\alpha^2\gamma_x^2\beta_x^2 p^{10}\eta^{10} C_g^8 C_f^2 L_f^6 + 48\gamma_y^2\beta_y^2 p^4\eta^4 C_g^2 L_f^4\sigma_f^2).$$

(84)

where the second step holds due to Lemmas 6, 7, 13, 14, 11, 15. Then, we enforce the coefficient of $\frac{1}{T}\sum_{t=0}^{T-1}\mathbb{E}[\|\bar{\mathbf{u}}_t\|^2]$ to be non-positive in the following. In particular, it can be done by solving the following

inequalities:

$$\frac{100\gamma_x^2 C_g^4 L_f^4}{\gamma_y^2 \mu^4} - \frac{1}{2} \le -\frac{1}{4},$$

$$\frac{300 C_g^2 L_f^2 (1 + 6L_f^2)}{\mu^2} \frac{2\gamma_x^2 C_g^2}{\alpha^2} + \frac{300 C_g^2 L_f^2}{\mu^2} \frac{4\gamma_x^2 L_f^2 C_g^2}{\beta_y^2} + \frac{300 C_g^2 L_f^2}{\mu^2} \frac{12 L_f^2 \gamma_x^2 C_g^2}{\alpha} + \frac{72\gamma_x^2 C_g^4}{\alpha} \tag{85}$$

$$+ \frac{48\gamma_x^2 (C_g^4 L_f^2 + C_f^2 L_g^2)}{\beta_x^2} - \frac{1}{4} \le 0.$$

Furthermore, we have the following inequalities:

$$\frac{100\gamma_x^2 C_g^4 L_f^4}{\gamma_y^2 \mu^4} - \frac{1}{2} \le -\frac{1}{4},$$

$$\frac{\gamma_x^2}{\alpha^2} \frac{600 C_g^4 L_f^2 (1 + 6L_f^2)}{\mu^2} \le \frac{1}{16},$$

$$\frac{\gamma_x^2}{\alpha} \Big( \frac{3600 C_g^4 L_f^4}{\mu^2} + 72 C_g^4 \Big) \le \frac{1}{16}, \tag{86}$$

$$\frac{\gamma_x^2}{\beta_y^2} \frac{1200 C_g^4 L_f^4}{\mu^2} \le \frac{1}{16},$$

$$\frac{\gamma_x^2}{\beta_x^2} 48 (C_g^4 L_f^2 + C_f^2 L_g^2) \le \frac{1}{16},$$

By solving these inequalities, we can get

$$\gamma_x \le \min \Big\{ \frac{\gamma_y \mu^2}{20 C_g^2 L_f^2}, \frac{\alpha\mu}{100 C_g^2 L_f \sqrt{1 + 6L_f^2}}, \frac{\sqrt{\alpha}\mu}{24 C_g \sqrt{100 C_g^2 L_f^4 + 2C_g^2 \mu^2}}, \frac{\beta_y \mu}{144 C_g^2 L_f^2}, \frac{\beta_x}{32 \sqrt{C_g^4 L_f^2 + C_f^2 L_g^2}} \Big\}. \tag{87}$$

Similarly, we enforce the coefficient of $\frac{1}{T} \sum_{t=0}^{T-1} \mathbb{E}[\|\bar{\mathbf{v}}_t\|^2]$ to be non-positive as follows:

$$\frac{300 C_g^2 L_f^2}{\mu^2} \frac{4\gamma_y^2 L_f^2}{\beta_y^2} + \frac{48\gamma_y^2 C_g^2 L_f^2}{\beta_x^2} - \frac{18\gamma_y C_g^2 L_f^2}{\mu} \le 0. \tag{88}$$

Then, it can be done by solving the following inequalities:

$$\frac{1200\gamma_y L_f^2}{\mu^2 \beta_y^2} \le \frac{9}{\mu},$$

$$\frac{48\gamma_y}{\beta_x^2} \le \frac{9}{\mu}. \tag{89}$$

Therefore, we can get

$$\gamma_y \le \min \Big\{ \frac{3\mu\beta_y^2}{400 L_f^2}, \frac{3\beta_x^2}{16\mu} \Big\}. \tag{90}$$

As a result, by setting $\alpha > 0$, $\beta_x > 0$, $\beta_y > 0$, $\eta \le \min\{\frac{1}{2\gamma_x L_\phi}, \frac{1}{\alpha}, \frac{1}{\beta_x}, \frac{1}{\beta_y}, 1\}$, and

$$\gamma_x \le \min \Big\{ \frac{\gamma_y \mu^2}{20 C_g^2 L_f^2}, \frac{\alpha\mu}{100 C_g^2 L_f \sqrt{1 + 6L_f^2}}, \frac{\beta_x}{32 \sqrt{C_g^4 L_f^2 + C_f^2 L_g^2}}, \frac{\beta_y \mu}{144 C_g^2 L_f^2}, \frac{\sqrt{\alpha}\mu}{24 C_g \sqrt{100 C_g^2 L_f^4 + 2C_g^2 \mu^2}} \Big\},$$

$$\gamma_y \le \min \Big\{ \frac{1}{6L_f}, \frac{3\mu\beta_y^2}{400 L_f^2}, \frac{3\beta_x^2}{16\mu} \Big\},$$

$$\tag{91}$$

we can get

$$\frac{1}{T}\sum_{t=0}^{T-1}\mathbb{E}[\|\nabla\Phi(\bar{\mathbf{x}}_t)\|^2] \leq \frac{2(\Phi(\bar{\mathbf{x}}_0)-\Phi(\bar{\mathbf{x}}_T)}{\gamma_x\eta T} + \frac{24C_g^2L_f^2}{\gamma_y\eta\mu T}\|\bar{\mathbf{y}}_0-\mathbf{y}^*(\bar{\mathbf{x}}_0)\|^2$$

$$+ \frac{300C_g^2L_f^2(1+6L_f^2)}{\mu^2}\frac{\sigma_g^2}{\alpha\eta TK} + \frac{300C_g^2L_f^2(1+6L_f^2)}{\mu^2}\frac{12\gamma_x^2\beta_x^2p^2\eta^2C_g^4C_f^2}{\alpha^2} + \frac{300C_g^2L_f^2(1+6L_f^2)}{\mu^2}\frac{\alpha\eta\sigma_g^2}{K}$$

$$+ \frac{600(L_f^2\sigma_g^2+\sigma_f^2)C_g^2L_f^2}{\beta_y\eta T\mu^2} + \frac{43200\gamma_x^2\beta_x^2p^4\eta^4C_g^6C_f^2L_f^4}{\mu^2} + \frac{64800\alpha^2p^2\eta^2C_g^4L_f^6\sigma_g^2}{\mu^2} + \frac{7776000\alpha^2\gamma_x^2\beta_x^2p^6\eta^6C_g^8C_f^2L_f^4}{\mu^2}$$

$$+ \frac{4147200\gamma_y^2\alpha^2p^4\eta^4C_g^4L_f^6\sigma_g^2}{\mu^2} + \frac{49766400\gamma_y^2\alpha^2\gamma_x^2\beta_x^2p^8\eta^8C_g^8C_f^2L_f^6}{\mu^2} + \frac{115200\gamma_y^2p^2\eta^2C_g^2L_f^4\sigma_f^2}{\mu^2}$$

$$+ \frac{21600\gamma_x^2\beta_x^2p^2\eta^2C_g^6C_f^2L_f^4}{\alpha\mu^2} + \frac{1800\alpha^2\eta\sigma_g^2C_g^2L_f^4}{K\mu^2} + \frac{300\beta_y\eta\sigma_f^2C_g^2L_f^2}{K\mu^2} + \frac{7200\gamma_x^2p^2\beta_x^2\eta^2L_f^4C_g^6C_f^2}{\beta_y^2\mu^2}$$

$$+ 144(C_g^4L_f^2+C_f^2L_g^2)\gamma_x^2\beta_x^2p^4\eta^4C_g^2C_f^2 + 41472\gamma_y^2\beta_y^2\alpha^2p^6\eta^6C_g^4L_f^4\sigma_g^2 + 497664\gamma_y^2\beta_y^2\alpha^2\gamma_x^2\beta_x^2p^{10}\eta^{10}C_g^8C_f^2L_f^4$$

$$+ 1152\gamma_y^2\beta_y^2p^4\eta^4C_g^2L_f^2\sigma_f^2 + \frac{6(3C_g^2L_f^2\sigma_g^2+3C_f^2\sigma_{g'}^2+3C_g^2\sigma_f^2)}{\beta_x\eta T} + 288\gamma_x^2p^2\eta^2(C_g^4L_f^2+C_f^2L_g^2)C_g^2C_f^2 + 864\gamma_x^2\beta_x^2p^4\eta^4C_g^6C_f^4$$

$$+ 1296\alpha^2p^2\eta^2C_g^4\sigma_g^2 + 15552\alpha^2\gamma_x^2\beta_x^2p^6\eta^6C_g^8C_f^2 + \frac{12\beta_x\eta C_f^2\sigma_{g'}^2}{K} + \frac{12\beta_x\eta C_g^2\sigma_f^2}{K}$$

$$+ \frac{165888\beta_y^2\gamma_y^2\alpha^2p^4\eta^4C_g^4L_f^4\sigma_g^2}{\beta_x^2} + \frac{1990656\beta_y^2\gamma_y^2\alpha^2\gamma_x^2\beta_x^2p^8\eta^8C_g^8C_f^2L_f^4}{\beta_x^2} + \frac{4716\beta_y^2\gamma_y^2p^2\eta^2C_g^2L_f^2\sigma_f^2}{\beta_x^2}$$

$$+ \frac{432\gamma_x^2\beta_x^2p^2\eta^2C_g^6C_f^2}{\alpha} + \frac{36\alpha^2\eta C_g^2\sigma_g^2}{K} + \frac{1800\gamma_x^2\beta_x^2p^4\eta^4C_g^6C_f^2L_f^4}{\mu^2}$$

$$+ \frac{600}{\mu^2}(1728\gamma_y^2\beta_y^2\alpha^2p^6\eta^6C_g^4L_f^6\sigma_g^2 + 20736\gamma_y^2\beta_y^2\alpha^2\gamma_x^2\beta_x^2p^{10}\eta^{10}C_g^8C_f^2L_f^6 + 48\gamma_y^2\beta_y^2p^4\eta^4C_g^2L_f^4\sigma_f^2) \,.$$

$$(92)$$

Since $\alpha$, $\beta_x$, $\beta_y$, $\gamma_x$, and $\gamma_y$ can be set as free hyperparameters, i.e., they are independent of the number of iterations, we can obtain

$$\frac{1}{T}\sum_{t=0}^{T-1}\mathbb{E}[\|\nabla\Phi(\bar{\mathbf{x}}_t)\|^2] \leq \frac{2(\Phi(\mathbf{x}_0)-\Phi(\mathbf{x}_*)}{\gamma_x\eta T} + \frac{24C_g^2L_f^2}{\gamma_y\eta\mu T}\|\mathbf{y}_0-\mathbf{y}^*(\mathbf{x}_0)\|^2 + O(\frac{\eta}{K}) + O(\frac{1}{\eta T})$$

$$+ O(p^2\eta^2) + O(p^4\eta^4) + O(p^6\eta^6) + O(p^8\eta^8) + O(p^{10}\eta^{10}) \,,$$

$$(93)$$

where $\mathbf{x}_*$ denotes the optimal solution. $\qquad\square$

# B Experimental Details

In Table 2, we summarize the hyperparameters for all methods. For a fair comparison, we use similar learning rates for all algorithms. For instance, the learning rate of LocalSGDAM and LocalSCGDAM is $\eta\gamma_x = 0.099$, which is very close to that of LocalSGDM and CoDA. In addition, the learning rate is decayed by 10 at 50% and 75% epochs for all methods. As for the number of epochs, we set it to 16 for Melanoma, 50 for FashionMNIST, and 100 for the others. Additionally, since CoDA is a stage-wise method, we use the same stage as that for learning rate decay.

Table 2: The hyperparameters of different methods.

| Methods | Hyperparameters | Value |
|---|---|---|
| LocalSGDM | learning rate | 0.1 |
| | momentum coefficient | 0.1 |
| CoDA | learning rate | 0.1 |
| LocalSGDAM | learning rate $\eta$ | 0.3 |
| | learning rate coefficient $\gamma_x$ and $\gamma_y$ | 0.33 |
| | momentum coefficient $\beta_x$ and $\beta_y$ | 3.3 |
| LocalSCGDAM (Ours) | learning rate $\eta$ | 0.3 |
| | learning rate coefficient $\gamma_x$ and $\gamma_y$ | 0.33 |
| | momentum coefficient $\beta_x$ and $\beta_y$ | 3.3 |
| | coefficient $\alpha$ | 3.0 |

The classifier for FashionMNIST is summarized in Table 3.

Table 3: The classifier for FashionMNIST.

| Layers | Operators | Configuration |
|---|---|---|
| Layer 1 | CNN | output channels: 32 |
| | Batchnorm | - |
| | ReLU | - |
| | Maxpooling | kernel size: 2, stride: 2 |
| Layer 2 | CNN | output channels: 64 |
| | Batchnorm | - |
| | ReLU | - |
| | Maxpooling | kernel size: 2, stride: 2 |
| Layer 3 | FC | output features: 600 |
| Layer 4 | FC | output features: 120 |
| Layer 5 | FC | output features: 1 |

Table 4: Description of benchmark datasets. Here, #pos denotes the number of positive samples, and #neg denotes the number of negative samples.

| Dataset | Training set | | Testing set | |
|---|---|---|---|---|
| | #pos | #neg | #pos | #neg |
| CIFAR10 | 2,777 | 25,000 | 5,000 | 5,000 |
| CIFAR100 | 2,777 | 25,000 | 5,000 | 5,000 |
| STL10 | 277 | 2,500 | 8,000 | 8,000 |
| FashionMNIST | 3,333 | 30,000 | 5,000 | 5,000 |
| CATvsDOG | 1,112 | 10,016 | 2,516 | 2,888 |
| Melanoma | 868 | 25,670 | 117 | 6,881 |