# OpenReview forum: "Federated Compositional Deep AUC Maximization"
_NeurIPS.cc/2023/Conference — NeurIPS 2023 poster_

### Official Review · Reviewer_5nau · 2023-07-04

**Soundness:** 2 fair
**Presentation:** 2 fair
**Contribution:** 2 fair
**Rating:** 3
**Confidence:** 3

**Summary:**

This work aims to address the challenges of imbalanced data in FL. To this end, the authors propose to optimize AUC score. Some experiments are conducted to verify the effectiveness of the proposed method.

**Strengths:**

The paper is easy to follow. The notations are well-defined. The studied problem is promising.

**Weaknesses:**

This work confuses me a lot.

The authors believe that data imbalance is a crucial issue under the FL scenario, which is consistent with my understanding. The problem has motivated the data-split fashion, i.e., Latent Dirichlet Sampling [1]. Moreover, many excellent works have demonstrated the effectiveness of their efforts in mitigating data heterogeneity [2, 3]. However, I cannot find these works in this work. This indicates that the authors may overlook some advanced methods in this field.

[1] Measuring the effects of non-identical data distribution for federated visual classification
[2] Federated optimization in heterogeneous networks
[3] SCAFFOLD: Stochastic controlled averaging for federated learning


**Questions:**

cf. Weaknesses

**Limitations:**

cf. Weaknesses

---

> ### Author Rebuttal · Authors · 2023-08-08
>
> We are grateful for the reviewer's comments and suggestions. We address the reviewer’s comments below.
>
> First, our method is significantly different from the heterogeneous federated learning approaches. Specifically, most existing heterogeneous federated learning approaches consider a setting where **the local distribution is imbalanced but the global distribution is balanced**, i.e., different clients have different data distributions but the combination of all clients' data is balanced.  On the contrary, our work considers a setting where **both the local and global distributions are imbalanced**, which is much more challenging than existing heterogeneous federated learning methods.
>
> Second, we have cited [2] and [3] in Lines 112-118. We will cite and discuss [1] as the reviewer suggested.
>
> Third, we have compared the SOTA method that is designed to address the global imbalance issue, i.e., LocalSGDM-RL [27]. Our method outperforms this strong baseline algorithm with a large margin for all datasets.
>
> **Hope we have answered all your questions. We sincerely hope the reviewer can consider our response and re-evaluate the contributions of our work. In fact, we considered a more challenging setting and provided a novel solution to address this challenge. Both theoretical and empirical results confirm the effectiveness of our method. We believe our contributions are important to this area and our work can inspire more follow-up works to handle more challenging federated learning settings.**

---

> > ### Comment · Reviewer_5nau · 2023-08-15
> > **Further comments**
> >
> > Hi,
> >
> > Thanks for the careful responses! Some of my concerns are addressed. However, it is still unclear why FedProx and Scaffold are not considered baseline methods. I suggest the authors add the corresponding results since existing methods can solve the studied imbalance problem partially.
> >
> > A following up confusion, based on the response, is that the mentioned theoretical results show the convergence rate, why it can confirm the effectiveness of the proposed method?

---

> > > ### Author Response · Authors · 2023-08-20
> > >
> > > Thanks for the reviewer's comments.
> > >
> > > **Q1**. …I suggest the authors add the corresponding results since existing methods can solve the studied imbalance problem partially.
> > >
> > > **A1**:  (1) Please note that our work focuses on optimizing AUC for federated learning. Therefore,  we compared with strong and direct baselines of federated learning for optimizing AUC, including CoDA [1] and LocalSGDAM. As prior works [2, 3] have demonstrated that optimizing traditional cross-entropy (CE) loss will yield worse results than AUC maximization for imbalanced data, we tried not to include too many baselines that optimize CE loss in FL. (2) We would like to draw reviewer's attention that CoDA leverages the epoch-wise proximal term as in FedProx and the variance-reduction as in Scaffold for solving the min-max formulation of AUC maximization. Therefore, CoDA is much a stronger baseline than FedProx and Scaffold. (3) Nevertheless, we have conducted an experiment to compare with Scaffold and FedProx for optimizing CE loss on STL10 dataset with $p=4$, and the results are shown below.  It can be observed that our algorithm can outperform those two baselines with a large margin.
> > >
> > > |     | Scaffold | FedProx | CoDA  | LocalSCGDAM(Ours)   |
> > > |-----|----------|---------|-------|-------|
> > > | AUC | 0.788    | 0.778   | 0.801 | 0.820 |
> > >
> > >
> > > **Q2**. …why it can confirm the effectiveness of the proposed method?...
> > >
> > > **A2**. The effectiveness of the proposed method can be explained below. (1) We optimize the compositional AUC formulation, which has been shown to learn much better feature representation than conventional AUC maximization [4]. (2) Our algorithm design and theoretical analysis ensure that our algorithm has a small sample complexity. This is also important as with the same number of epochs, an algorithm with a higher sample complexity may find a worse solution. Therefore, our theoretical analysis ensures that our algorithm can quickly find a good solution to compositional AUC maximization. Together, it can explain the effectiveness of our algorithm. (3) We would like to emphasize that it is not trivial to attain the $O(1/(K\epsilon^4))$ complexity for solving the compositional AUC loss as in our paper. For instance,  [5, 6] show that the stochastic gradient descent (SGD) or stochastic compositional gradient descent (SCGD) algorithm under federated learning setting has the sample complexity of $O(1/\epsilon^{8})$ or $O(1/\epsilon^{5})$, which cannot match the sample complexity $O(1/(K\epsilon^{4}))$ of SGD for non-compositional optimization problems. In our theoretical analysis, we show that our algorithm LocalSCGDAM can achieve the sample complexity of $O(1/(K\epsilon^{4}))$,  which can match that of traditional federated learning algorithms for non-compositional optimization problems.
> > >
> > >
> > >
> > > [1] Guo et al., Communication-efficient distributed stochastic auc maximization with deep neural networks. ICML 2020.
> > >
> > > [2] Yuan et al., Robust Deep AUC Maximization: A New Surrogate Loss and Empirical Studies on Medical Image Classification. ICCV 2021.
> > >
> > > [3] Liu et al., Stochastic AUC Maximization with Deep Neural Networks. ICLR 2020.
> > >
> > > [4] Yuan et al., Compositional Training for End-to-End Deep AUC Maximization. ICLR 2022.
> > >
> > > [5] Huang et al., "Compositional federated learning: Applications in distributionally robust averaging and meta learning." arXiv preprint arXiv:2106.11264 (2021).
> > >
> > > [6] Wang et al., "Memory-Based Optimization Methods for Model-Agnostic Meta-Learning and Personalized Federated Learning." JMLR 24 (2023): 1-46.

---

### Official Review · Reviewer_BTwE · 2023-07-15

**Soundness:** 3 good
**Presentation:** 3 good
**Contribution:** 3 good
**Rating:** 7
**Confidence:** 5

**Summary:**

This paper firstly studies the federated compositional AUC maximization problem, which includes both the local and global imbalanced distributions, and proposes the momentum-based algorithm LocalSCGDAM to solve this problem. The SOTA convergence rates are established and various experiments are used to evaluate the proposed algorithms.

**Strengths:**

Novelty: This paper is the first work to consider the federated compositional AUC maximization problem. Data heterogeneity is a key problem in federated learning and many works focus on it.  The Federated compositional AUC maximization problem is more challenging because it considers both the local and global imbalanced distributions.

Quality: It proposes a new algorithm to solve the problem. The structure of the algorithm is clear.  Both theoretical analysis and experimental verification are provided. The theoretical analysis is very complete.

Clarity: The paper is organized well and it is easy to follow.



**Weaknesses:**

1. More motivation should be introduced. It includes why 1) AUC is significant and 2) why FL AUC should be considered.

2. In the experiments, all samples are set to 0.1. Discussion about different imbalanced data settings is welcomed.




**Questions:**

1. In the math, $\mathcal{D}^k_{g}$ and $\mathcal{D}^k_{n}$ are clear and they are the datasets in the inner layer and out layer. But in the AUC maximization, how to define these datasets?

2. This work solves the binary imbalanced data distribution problem well. I am curious whether it is possible to extend it to the multiple-class imbalanced data distribution problem.

3. In the nonconvex optimization, we consider the convergence rate, including sample complexity and communication complexity, to reach an $\epsilon$-stationary point. In this case, what is the convergence rate of your algorithm?

---

> ### Author Rebuttal · Authors · 2023-08-08
>
> We are grateful for the reviewer's comments and suggestions. We address the reviewer’s comments below.
>
> Answers for weakness.
>
> **1**. First, when the data distribution is imbalanced, directly minimizing the cross-entropy loss function cannot learn a good classifier since it may ignore the minority class. On the contrary, the AUC loss function focuses on both majority and minority classes, which can alleviate the limitation of traditional cross-entropy loss function [31]. Thus, we leverage the AUC maximization to address the imbalance issue, which has been confirmed by our experiments, i.e., LocalSCGDAM outperforms LocalSGD.
>
> Second, the AUC loss function is a minimax function, which is more difficult to optimize than the traditional convex cross-entropy loss function. Thus, we developed the LocalSCGDAM algorithm to optimize the compositional AUC loss function, which incorporates the pertaining process as shown in Eq.(2). As such, the prediction performance is better than directly optimizing the AUC loss function, which is confirmed by our experiments, i.e., LocalSCGDAM outperforms CoDA and LocalSGDAM.
>
> Third, FL is an effective tool for real-world data analysis tasks and data distributions in those tasks are typically imbalanced, e.g., the electronic health record (EHR) data. Thus, enabling federated learning for AUC optimization is necessary and important to address real-world learning challenges.
>
> **2**. Thanks for the reviewer's suggestion. In fact, we have already conducted this experiment in Appendix. The results in Figure 4 has confirmed the superior performance of our method for different imbalanced ratios.
>
> Answers for questions.
>
> **1**. As shown in Eq. (2), the inner-level function is to pre-train the classifier by minimizing the cross-entropy loss function, and the outer-level function is to learn the classifier via optimizing the AUC loss function. Both of them use the training samples and their labels. In other words, $\mathcal{D}_f$ and $\mathcal{D}_g$ in compositional AUC maximization denote the two copies of the training dataset.
>
> **2**. It is easy to extend the binary classification case to the multi-class case. Please see Eq.(4) in [16].
>
> **3**. Based on Theorem 1, to reach the $\epsilon$-stationary point, i.e., $\frac{1}{T} \sum_{t=0}^{T-1} \mathbb{E}\left[\left\|\nabla \Phi\left(\overline{\mathbf{x}}_t\right)\right\|^2\right]\leq \epsilon^2$ the convergence rate is $O(\frac{1}{K\epsilon^4})$, the sample complexity on each client is  $O(\frac{1}{K\epsilon^4})$, the communication complexity is $O(\frac{1}{\epsilon^3})$.

---

> > ### Comment · Reviewer_BTwE · 2023-08-17
> >
> > Thanks for your response and I keep my score.

---

### Official Review · Reviewer_jnCq · 2023-07-24

**Soundness:** 3 good
**Presentation:** 3 good
**Contribution:** 3 good
**Rating:** 5
**Confidence:** 3

**Summary:**

The paper proposes a new federated learning algorithm to address the class imbalance problem. Instead of using cross-entropy loss functions, the proposed algorithm directly optimizes the AUC score by solving a federated stochastic compositional minimux optimization problem. Specifically, the paper proposes to employ the local stochastic gradient with momentum to update the local model parameters. Experiments show that the proposed algorithm achieves better model performance than the other baselines.

**Strengths:**

1. The paper considers a setting where the global distribution is also class-imbalanced, which is interesting.

2. The paper provides a theoretical analysis of the convergence of the proposed algorithm.


**Weaknesses:**

1. There are many FL studies that try to address non-IID challenge in FL. However, the baselines seem to focus on different optimization methods and lack SOTA FL studies that address the non-IID data (e.g, [1][2]).

[1] Addressing class imbalance in federated learning
[2] No fear of heterogeneity: Classifier calibration for federated learning with non-iid data

2. The theoretical analysis requires assumptions on the outer-level and inner-level functions. I’m not sure how realistic these assumptions are.

3. The algorithm estimates the inner-level function in local training. It may not be applicable in the client sampling setting, where a client may be selected after many rounds.



**Questions:**

1. Is existing FL approaches applicable with LocalSCGDAM? Existing studies usually optimize the standard cross-entropy loss. I’m curious whether they are applicable to the studied loss function and how they compare with LocalSCGDAM.

2. Is the experimental setting satisfies the assumptions in Section 3.2?

3. Is the algorithm applicable in the client sampling setting? Also, how many clients are used in the experiments? I suggest the authors to add experiments to study the scalability.

4. How does the convergence rate of LocalSCGDAM compare with other related studies?

5. Is directly optimizing AUC loss a mainstream method in class-imbalanced learning?


**Limitations:**

One potential limitation is that the algorithm may not work well in the client sampling setting.

---

> ### Author Rebuttal · Authors · 2023-08-08
>
> We are grateful for the reviewer's comments and suggestions. We address the reviewer’s comments below:
>
> Answer for weakness.
>
> **1**. Firstly, our method is significantly different from the heterogeneous federated learning methods. Specifically, our paper aims to address a more challenging issue than the traditional FL methods designed for non-IID data.
> More specifically, even though some traditional FL methods try to address the non-IID issue, they typically assume **the global data distribution (i.e., combining the data of all clients) is balanced**. For instance, the reference [2] that the reviewer provided uses Dir distribution to simulate the non-iid data, where different clients have different distributions but the global distribution is balanced. On the contrary, our method and the reference [1] that the reviewer provided assume **the global distribution is imbalanced**, which is much more challenging. To address this challenging task, we developed a federated compositional AUC maximization model to handle the global imbalance issue, which has shown superior performance according to our experimental results.
>
> Secondly, we have compared with [1] in our experiments, which is denoted by LocalSGDM-RL in Line 257. All experimental results have shown that our algorithm can outperform [1] by a large margin.
>
> **2**. Those assumptions are common in compositional optimization problems. Especially, the single-machine counterpart algorithms [4, 35] use the same assumptions. We do not have any strong assumptions compared with them. In fact, these assumptions are mild and the model used in our experiment, which is also used in [35], satisfies those assumptions.
>
>
> **3**. It is trivial to extend our algorithm to the client sampling setting. Each selected client updates the inner-level function estimator and then computes stochastic compositional gradient to update model parameters, which does not require special operations under the client sampling setting. We will provide discussions about this point in the final version.
>
>
> Answer for questions.
>
> **1**. Existing FL approaches learn the classifier via minimizing the cross-entropy loss function, which is a **minimization** problem. On the contrary, the AUC loss function used in this paper is a **minimax** loss function. For minimization problems, one should use the **stochastic gradient descent** algorithm, while the **stochastic gradient descent ascent** algorithm should be used for minimax problems. Since LocalSCGDAM is designed for a compositional minimax problem, not a minimization problem, the standard FL approaches cannot be applied to the compositional minimax problems.
>
>
> **2**. Please see the answers for weakness 2.
>
> **3**. Please see the answers for weakness 3. We use four clients currently. We will use more clients to verify the performance in the final version.
>
> **4**. As shown in Remark 1, the convergence rate is $O(1/\sqrt{KT})$ where $K$ is the number of clients, $T$ is the number of iterations. Obviously, this convergence rate matches that of standard FedAvg for nonconvex problems.
>
> **5**. Directly optimizing AUC loss is a powerful approach to address the imbalanced data classification issue. It has attracted a lot of attention in the past few years, e.g., [6,35].

---

> > ### Comment · Reviewer_jnCq · 2023-08-16
> >
> > Thanks for your response. Most concerns are addressed but my concern about client sampling remains. The estimation of inner-level function may be affected by client sampling. I suggest that the authors add experiments to it.

---

> > > ### Author Response · Authors · 2023-08-20
> > >
> > > Thanks for the reviewer's comments.
> > >
> > > **Q1**. …I suggest that the authors add experiments to it.
> > >
> > > **A1**.  We would like to emphasize that our experiments focus on the **cross-silo federated learning**, which has been widely used in many applications, such as the biomedical image classification used in our experiment. **For cross-silo federated learning, it is NOT necessary to do client sampling**. In particular, under the cross-silo federated learning setting, there are not a huge number of participants. It is preferable to include all participants to participate in the computation since there does not exist a lot of training data, e.g., biomedical data. Thus, it is not necessary to do client sampling.
> > >
> > > Nevertheless, we have conducted an experiment about client sampling. In this additional experiment, there are four workers. We use STL10 dataset and the communication period is set to 4. To simulate the client sampling scenario, we randomly select three workers to participate in the computation. The following table shows the test AUC score of our algorithm with/without client sampling. It can be observed that LocalSCGDAM-with-sampling has a similar performance to LocalSCGDAM-without-sampling.
> > >
> > > |     | LocalSCGDAM-with-sampling | LocalSCGDAM-without-sampling |
> > > |-----|---------------------------|------------------------------|
> > > | AUC | 0.813                     | 0.820                        |

---

> > > > ### Comment · Reviewer_jnCq · 2023-08-21
> > > >
> > > > Thanks for your further response! I suggest that 1) you highlight you are focusing on cross-silo FL, or 2) conduct experiments in the client-sampling setting in the revision.

---

> > > > > ### Author Response · Authors · 2023-08-22
> > > > >
> > > > > Thanks for your suggestions!

---

### Decision · Program_Chairs · 2023-09-21

**Decision:**

Accept (poster)

**Comment:**

The paper considers federated supervised learning (classification) in the case where the data belonging to different classes is (highly) imbalanced. The main idea behind the proposed method is to directly optimize the area under curve (AUC) score. This is performed by casting the AUC maximization problem as a federated compositional minimax optimization problem, for which the authors develop a local stochastic compositional gradient descent ascent with momentum algorithm, and provide bounds on the computational and communication complexities of our algorithm.

The reviewers had several concerns and the authors made a great effort to address those concerns. After going through the paper and all the responses, my assessment is that the major concerns have been resolved and the contributions are sufficiently novel to warrantee acceptance.

I recommend to the authors to incorporate all the reviewers' comments in the updated version (also, please add all the simulations results you did during the rebuttal/discussion period to the main paper). As a side note, there are approaches to federated learning with imbalanced data which are based on constrained optimization; in my view, those approaches can also viewed as some kind of a minimax method (once viewed from the angle of duality). I would recommend the authors to discuss those works as related works too (e.g. see Shen et al: "An agnostic approach to federated learning with imbalanced data")